# TOWARD MONOSEMANTIC CLINICAL EXPLANATIONS FOR ALZHEIMER'S DIAGNOSIS VIA ATTRIBUTION AND MECHANISTIC INTERPRETABILITY

## ABSTRACT

Interpretability remains a major obstacle to deploying large language models (LLMs) in high-stakes settings such as Alzheimer's disease (AD) progression diagnosis, where early and explainable predictions are essential. Traditional attribution methods suffer from inter-method variability and often produce unstable explanations due to the polysemantic nature of LLM representations, while mechanistic interpretability lacks direct alignment with model inputs/outputs and does not provide importance scores. We propose a unified interpretability framework that integrates attributional and mechanistic perspectives through monosemantic feature extraction. Our approach evaluates six attribution techniques, refines them using a learning-based explanation optimizer, and employs sparse autoencoders (SAEs) to map LLM activations into a disentangled latent space that supports clearer and more coherent attribution analysis. Comparing latent-space and native attributions, we observe substantial gains in robustness, consistency, and semantic clarity. Experiments on IID and OOD Alzheimer's cohorts across binary and three-class tasks demonstrate that our framework yields more reliable, clinically aligned explanations and reveals meaningful diagnostic patterns. This work advances the safe and trustworthy use of LLMs in cognitive health and neurodegenerative disease assessment.

## 1 INTRODUCTION

Explainable Artificial Intelligence (XAI) plays a crucial role in building trust in machine learning systems, especially in sensitive and high-stakes areas such as finance, climate, autonomous driving and healthcare (Doshi-Velez & Kim, 2017; Manifold et al., 2021). In medical settings, interpretability is essential for clinical integration and regulatory approval, especially in complex diseases such as Alzheimer's Disease (AD) progression (Super et al., 2023), where early and accurate detection can substantially alter treatment results (Jack et al., 2018).

Although machine learning has shown promise in AD diagnostics using multimodal clinical data (Bron et al., 2015), the application of large language models (LLMs) such as GPT-4 and LLaMA-2 in structured clinical settings remains limited (Brown et al., 2020; Touvron et al., 2023). A key obstacle is the *polysemanticity* of internal neural representations—individual neurons or features often encode multiple, semantically unrelated concepts (Olah et al., 2020; Cunningham et al., 2023; Elhage et al., 2022a). This entanglement undermines the interpretability of standard attribution techniques such as gradients, perturbations, and integrated paths, which typically assume one-to-one correspondence between features and meanings. Moreover, existing attribution methods assign importance scores to input features (e.g., words or tokens), yet they fall short in addressing the polysemantic nature of internal representations. This limitation often leads to ambiguous or misleading explanations—particularly problematic in clinical applications, where interpretability is critical (Samek et al., 2021; van der Velden et al., 2022; Quellec et al., 2021; Mamalakis et al., 2023). Moreover, the inter-method variability of attributional techniques is another limitation of dataset-driven explanations, as it reduces trust in the interpretability of AI models (Mamalakis et al., 2025). In contrast, mechanistic interpretability aims to uncover the internal structure of neural computation by identifying semantically coherent components within the model. Sparse Autoencoders (SAE) have played a pivotal role in advancing our understanding

of such representations in both language and vision domains (Gorton, 2024). SAEs aim to solve the superposition problem in neural feature representations by mapping the model's activations into a more monosemantic latent space, where individual features are better aligned with specific concepts in the network (Cunningham et al., 2023). However, these mechanistic tools typically lack attributional resolution, limiting their utility for explaining how specific inputs contribute to model predictions in real-world, decision-critical scenarios (Elhage et al., 2022a).

This reveals a critical gap in the current landscape: attributional techniques offer only surface-level explanations using polysemantic features, while mechanistic methods provide structural insights based on monosemantic feature alignment but lack attributional grounding. To date, no unified framework successfully integrates both paradigms—particularly in the domain of LLM-based clinical inference.

To address these challenges and produce stable, clinically meaningful explanations from large language models, we introduce a unified monosemantic attribution framework that integrates both attributional and mechanistic interpretability approaches (Figure 1). Our approach first employs sparse autoencoders (SAEs) to transform polysemantic LLM activations into a more monosemantic latent space, where individual features are encouraged to correspond to disentangled semantic factors. This bottleneck substantially reduces representational complexity and enables classical attribution methods to assign more precise and semantically coherent importance scores. For the attribution component, we apply six established techniques—Feature Ablation, Layer Activations, Layer Conductance, Layer Gradient SHAP (Lundberg & Lee, 2017a), Layer Integrated Gradients (Sundararajan et al., 2017b), and Layer Gradient×Activation—both in the native activation space (polysemantic features) and in the SAE-induced latent space (more monosemantic features). To address the inter-method variability problem (Mamalakis et al., 2025), we combine the resulting attribution vectors and introduce the *Transformer Explanation Optimizer (TEO)*, a learning-based mechanism that selects explanations with maximal alignment to model behavior and dataset-level consistency (Mamalakis et al., 2025). We experiment with encoder–decoder architectures based on 1D transformers (TEO) and diffusion networks (DEO). For meta-level assessment and visualization, we embed these optimized attributions into a 2D manifold using UMAP (McInnes et al., 2018), and impose a global coherence constraint by evaluating their linear structure along the primary UMAP components. This geometry-aware constraint acts as an additional regularizer and provides a principled way to impose global structure on the explanation space.

The central hypothesis of this work—namely, that as the embedding layer approaches a more monosemantic representation, attribution scores become more stable, less complex, and more diagnostically informative compared to those derived from polysemantic features—is evaluated through a series of experiments in which different LLMs are trained and tested on ADNI Mueller et al. (2005) (IID) and tested on BrainLAT Prado et al. (2023) (OOD) to assess resilience under demographic and protocol shifts. We study both binary (Control vs. Alzheimer's disease (AD)) and multi-class settings (Control, Early Mild Cognitive Impairment (EMCI), Late Mild Cognitive Impairment (LMCI) in ADNI; Control, Frontotemporal Dementia (FTD), and AD in BrainLAT). The data are partitioned into 80% training and 20% validation within the training portion (which itself constitutes 80% of the full dataset), with the remaining 20% held out for testing. The last-layer embeddings of the LLM that achieved the strongest baseline performance are subsequently used to evaluate the proposed interpretability framework. Importantly, both the SAEs and TEO (with and without the SAE bottleneck) are trained exclusively on IID explanation cohorts using only these training/validation splits and are then evaluated on the OOD dataset *without any additional training or adjustments*, allowing us to assess out-of-distribution robustness under strict generalization conditions.

**Contributions and Novelty:** We propose a unified interpretability framework that couples explainer optimisation with monosemantic feature extraction and an optional geometry-aware constraint. Concretely:

- **Transformer Explanation Optimizer (TEO).** We propose a learning-based optimizer that consolidates and refines the outputs of six widely used attribution methods. TEO reduces inter-method variance and enhances explanation clarity, all without requiring any retraining of the underlying LLM.

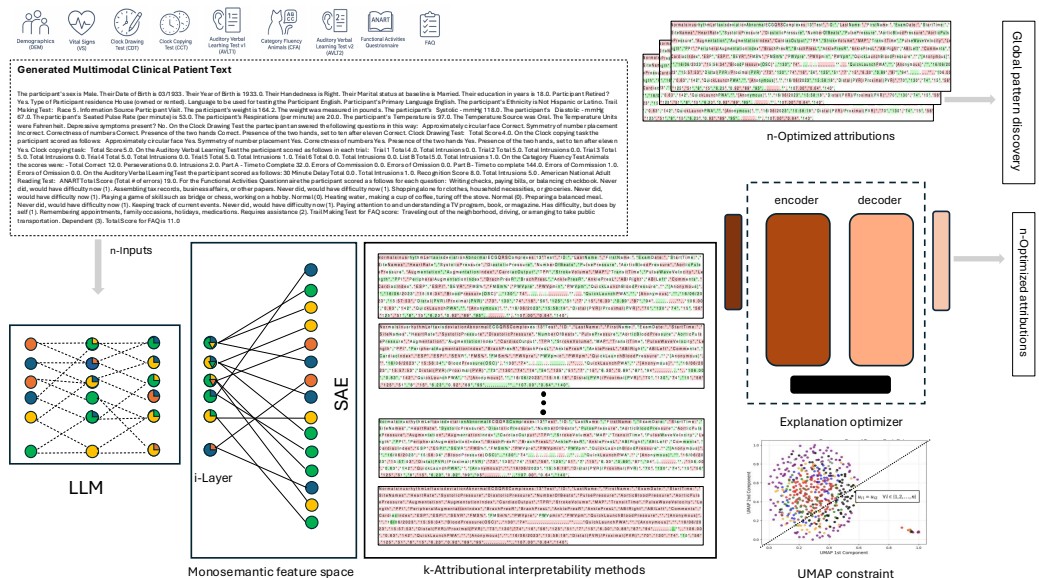

Figure 1: Proposed interpretability framework for LLM in Alzheimer's diagnosis. The model integrates k-attributional methods with a SAE to generate a monosemantic feature space. An explanation optimizer refines attribution outputs, enhancing clarity and reducing variability. Global explanation quality is visualized and assessed using UMAP and a linear meta-rule, supporting both individual prediction interpretability and cohort-level pattern discovery.

- **Attribution Through a Monosemantic Bottleneck (SAE).** We train sparse autoencoders on LLM activations to obtain a more monosemantic latent representation of the embedding layer, and we compute attributions in this disentangled space. Introducing this SAE-based bottleneck yields substantially sparser and more robust explanations with clearer semantic alignment, enabling the identification of meaningful biomarkers and pathology-related patterns when combined with the proposed TEO optimizer.

- **Latent vs. Native Attributions.** We directly compare attributions computed in the SAE-induced monosemantic latent space with those obtained in the native model space. Across all tasks, latent-space attributions exhibit improved robustness (lower RIS/ROS) and greater semantic coherence. Statistical testing confirms significant differences between the two settings, with the SAE-enabled variant consistently outperforming the no-SAE baseline.

- **Tunable Sparsity–Stability Frontier.** Across IID and OOD regimes, and for both binary and three-class tasks, TEO with monosemantic features (TEO-SAE) provides the most stable explanations, while a geometry-aware constraint (TEO-UMAP) recovers higher sparsity with only a modest reduction in stability.

- **Clinical Signal Discovery.** Our framework reveals clinically meaningful structure in multimodal Alzheimer's data and produces class-specific, human-aligned explanations suitable for integration into clinical reasoning workflows. The resulting attributions highlight the most informative neurocognitive assessments, reducing clinical burden by prioritizing the tests that most effectively support Alzheimer's diagnosis and progression monitoring.

## 2 METHODS

### 2.1 ATTRIBUTIONAL THEORY AND METHODS

Attribution explainability methods follow the framework of additive feature attribution, where the explanation model $g(f, \mathbf{x})$ is represented as a linear function of simplified input features:

$$g(f, \mathbf{x}) = \phi_0 + \sum_{i=1}^{M} \phi_i x_i \tag{1}$$

Here, $f$ is the predictive model, $\phi_i \in \mathbb{R}$ is the attribution (importance) assigned to feature $x_i$, and $M$ is the number of simplified input features (here 512).

For this study, we employed six well-established attributional interpretability methods applied to large language models (LLMs), denoted as $K = 6$: *Feature Ablation*, *Layer Activations* (which capture the embedding activation space of a specific layer of interest within the LLM), *Layer Conduction*, *Layer Gradient-SHAP* (Lundberg & Lee, 2017b), *Layer Integrated Gradients* (Sundararajan et al., 2017a), and *Layer Gradient × Activation* (For analytical mathematic formulation see Appendix A.1.). To align these layer-wise interpretability methods with the additive feature attribution framework, we reinterpret the internal activations (i.e., latent units) of a network layer $L$ as simplified input features. The objective is to estimate an attribution score $\phi_i$ for each unit, where $\phi_i \in \mathbb{R}$ quantifies the contribution of the corresponding neuron to the model's prediction.

All attribution methods were applied to the final (22nd) layer of the MODERN-BERT LLM—the model variant that achieved the highest classification accuracy in our evaluations (see Supplementary section B.2). These formulations allow us to ground various neural attribution techniques within a unified additive explanation model, facilitating their comparison and hybridization under shared theoretical assumptions.

## 2.2 ATTRIBUTIONAL EXPLANATION OPTIMIZER FRAMEWORK

Let $\mathscr{A} = \{A_1, A_2, \ldots, A_K\}$ denote the set of $K = 6$ attribution methods applied to the final layer $L$ of the model $f$. Each method $A_k$ generates an attribution vector $\boldsymbol{\phi}^{(k)} = [\phi_1^{(k)}, \phi_2^{(k)}, \ldots, \phi_M^{(k)}]$, where $M$ is the number of latent features (neurons) in layer $L$. The goal is to derive a unified attribution vector $\bar{\boldsymbol{\phi}}$ that captures the consensus explanation across methods. Each attribution vector $\boldsymbol{\phi}^{(k)}$ is evaluated using the Relative Input Stability (RIS), Relative Output Stability (ROS) Agarwal et al. (2022b), and Sparseness Chalasani et al. (2020) metrics (For analytical mathematic formulation see Appendix A.2.).

### 2.2.1 AGGREGATION OF ATTRIBUTIONS

The weighted average attribution vector $\bar{\boldsymbol{\phi}}$ serves as the target explanation for the optimization process and it is calculated as:

$$\bar{\boldsymbol{\phi}} = \sum_{k=1}^{K} w_k \cdot \boldsymbol{\phi}^{(k)} \tag{2}$$

### 2.2.2 EXPLANATION RECONSTRUCTION VIA ENCODER–DECODER MODELS

An encoder–decoder model is trained to generate a reconstructed explanation $\hat{\boldsymbol{\phi}}$ from the original input $\boldsymbol{x}$. Two architectures are considered the Diffusion UNet1D Ronneberger et al. (2015) (*Diffusion Explanation Optimizer*, DEO) and the x-transformer autoencoder Vaswani et al. (2017); Nguyen & Salazar (2019) (*Transformer Explanation Optimizer*, TEO). For the analytical mathematical formulation, see Appendix A.2.4.

### 2.2.3 THE TOTAL COST FUNCTION OF THE OPTIMIZER

As previously highlighted, the reconstruction of the optimal explanation and the associated cost function adhere to the same principles and architectural design outlined in Mamalakis et al. (2025). The cost function consists of three key components: sparseness, as defined in Chalasani et al. (2020); ROS and RIS scores Agarwal et al. (2022a); and similarity. The integration of these components ensures a robust and interpretable evaluation. The total cost function for training the reconstruction model is:

$$\begin{aligned}
\mathscr{L}_{\text{total}}(\boldsymbol{\phi}^{(k)}, \hat{\boldsymbol{\phi}}) = {} & \lambda_1 \cdot \frac{1}{M_{\text{RIS}}(f, \hat{\boldsymbol{\phi}})} + \lambda_2 \cdot \frac{1}{M_{\text{ROS}}(f, \hat{\boldsymbol{\phi}})} \\
& + \lambda_3 \cdot M_{\text{sparse}}(f, \hat{\boldsymbol{\phi}}) + \lambda_4 \cdot \mathscr{L}_{\text{similarity}}(\hat{\boldsymbol{\phi}}, \bar{\boldsymbol{\phi}})
\end{aligned} \tag{3}$$

where:$\lambda_1, \lambda_2, \lambda_3, \lambda_4$ are hyperparameters controlling the influence of each loss term. This formulation enables a principled and quantitative integration of multiple attribution methods, optimizing toward a robust and interpretable explanation.

## 2.3 UMAP PROJECTION AND LINEAR CONSTRAINT

To obtain a unified and comparable low-dimensional representation of attribution scores across all tokenizer features, we apply a feature-wise UMAP projection to the normalized attribution tensor of shape $\mathbb{R}^{M \times T}$, where $M$ is the number of test samples and $T$ denotes the dimensionality of the tokenizer embedding space. For each feature $j$, the attribution vector $\mathbf{x}^{(j)} \in \mathbb{R}^M$ is min–max normalized and embedded using a one-dimensional UMAP transformation into a two-dimensional space $\mathbf{y}^{(j)} \in \mathbb{R}^{M \times 2}$. The resulting coordinates are normalized to $[0, 1]$ so that all feature-wise embeddings share a common bounded range. This nonlinear projection preserves the local neighborhood structure of each attribution distribution while mapping all $T$ tokenizer features into an aligned and directly comparable representation space across attribution methods (further mathematical formulation please see Appendix A.4).

**Linear Constraint on UMAP Embeddings:** To encourage structural consistency in the UMAP McInnes et al. (2018) embeddings, we introduce a linear constraint requiring equality between the first and second embedding components for each point, expressed as $u_{i1} = u_{i2}$. This can be written equivalently as $u_{i1} - u_{i2} = 0$ for all $i$. We incorporate this constraint into the overall optimization objective via a penalty term,

$$\lambda_5 \sum_{i=1}^{n} (u_{i1} - u_{i2})^2,$$

where $\lambda_5$ controls the strength of the constraint. This formulation enforces consistency of the reconstructed embeddings while preserving flexibility in the learned attribution representation.

## 2.4 THE SAE APPROACH AND ARCHITECTURES

The mathematical formulation situates SAE architectures within the theoretical framework of superposition and semantic disentanglement (for an analytical mathematical formulation, see Appendix A.5). By expressing hidden states as sparse linear combinations of interpretable features, SAEs bridge the gap between low-level activations and human-understandable concepts. Let $\mathbf{x} \in \mathbb{R}^d$ denote a layer's neuron activation vector in a pretrained model. A Sparse Autoencoder learns a sparse feature representation $\mathbf{a} \in \mathbb{R}^F$ such that:

$$\hat{\mathbf{x}} = W\mathbf{a} + \mathbf{b}, \tag{4}$$

where $W \in \mathbb{R}^{d \times F}$ is the decoder (dictionary) matrix and $\mathbf{b} \in \mathbb{R}^d$ is a learned bias term. Each column $W_{\cdot,i}$ represents the direction of feature $i$ in neuron space, and $a_i$ is its activation. This linear mapping enables complex activations to be expressed as combinations of more interpretable features. If $F > d$, then the feature space is overcomplete, and $W$ cannot be full-rank. This leads to superposition, where multiple features overlap in the same subspace, and individual neurons encode multiple unrelated concepts Elhage et al. (2022b). If $W$ is invertible and aligned to a basis, each neuron corresponds to a single feature. The representation is monosemantic and disentangled Olah et al. (2020). When $W$ has overlapping columns, neurons can respond to multiple features, yielding polysemantic behavior. That is, for some $j$, $x_j = \sum_i W_{j,i} a_i$ involves multiple nonzero terms Bills et al. (2023). Variants of SAEs like TopK, JumpReLU, and Gated-SAEs offer increasingly precise control over the mapping between low-level activations and human-understandable concepts, enabling fine-grained analysis and intervention (analytical mathematical formulation, see Appendix A.6.).

## 2.5 ATTRIBUTION FROM SPARSE FEATURE SPACE TO INPUT TOKENS

Let $\mathbf{x}_{\text{input}} \in \mathbb{R}^{d_{\text{input}}}$ denote the input embedding vector (e.g., LLM token embeddings), $\mathbf{x} = f(\mathbf{x}_{\text{input}}) \in \mathbb{R}^d$ the hidden layer activation of the LLM, $\mathbf{a} = \text{Encoder}(\mathbf{x}) \in \mathbb{R}^F$ the SAE sparse feature vector, and $\hat{\mathbf{x}} = W\mathbf{a} + \mathbf{b}$ the reconstructed activation from the SAE decoder. Now suppose we have a sparse attribution vector $\psi_i$ over features $\mathbf{a}$, i.e., $\psi \in \mathbb{R}^F$, where each $\psi_i$ reflects the importance of SAE feature $a_i$. We aim to assign importance $\Phi_k$ to each input token dimension $x_{\text{input},k}$.

We propagate the feature attributions backward through the encoder to the input. Using the chain rule:

$$\Phi_k = \sum_{i=1}^{F} \psi_i \cdot \frac{\partial a_i}{\partial x_{\text{input},k}} = \sum_{i=1}^{F} \psi_i \cdot \frac{\partial a_i}{\partial \mathbf{x}} \cdot \frac{\partial \mathbf{x}}{\partial x_{\text{input},k}} \qquad (5)$$

where $\frac{\partial a_i}{\partial \mathbf{x}}$ is the encoder Jacobian (SAE layer), and $\frac{\partial \mathbf{x}}{\partial x_{\text{input},k}}$ is the LLM gradient from input token to hidden layer. This gives us a scalar attribution $\Phi_k \in \mathbb{R}$ for each token/input embedding dimension $k$. This represents how much each input token contributes to the sparse SAE features identified as important. In doing so, we assess the contribution of input features based on the monosemantic behavior of the trained network's internal mechanisms. Building on our study, we apply the six previously discussed attribution methods at two levels: from the SAE feature space to the encoder layer, and from the encoder layer to the input embedding space. This dual-level attribution analysis enables us to investigate how interpretable sparse features relate to model internals and ultimately shape the input-level representations. Attribution methods (e.g., Integrated Gradients, SHAP) can directly estimate:

$$\boldsymbol{\phi}^{\text{input}} = \text{AttributionMethod}(f, \mathbf{x}_{\text{input}}, \boldsymbol{\phi}^{\text{SAE}}) \qquad (6)$$

where $\boldsymbol{\phi}^{\text{SAE}}$ denotes the monosemantic feature space of the SAE network. Thus, the dual-level approach allows us to connect semantically meaningful sparse features to the raw input representation space.

## 2.6 LLM NETWORKS AND HYPERPARAMETER TUNING

We evaluate encoder-based LLMs (BERT, RoBERTa, DistilBERT, ALBERT, BioBERT, ModernBERT) on ADNI (IID) and BRAINLAT (OOD) under a unified protocol spanning full fine-tuning, zero-shot, few-shot with temperature control, and parameter-efficient LoRA. *ModernBERT* outperformed all other networks on ADNI: in the binary task it achieved the highest F1 (75.89%), AUC-PR (86.41%), ROC-AUC (83.95%), and Accuracy (72.37%), and it remained strongest in the three-class setting (F1 68.80%, AUC-PR 78.48%, ROC-AUC 78.67%, Accuracy 65.05%). For OOD model adaptation, *ModernBERT* zero-shot yielded 55% Accuracy, few-shot/LoRA provided modest gains (62%), while full fine-tuning peaked at 84% Accuracy but lies outside our scope. Accordingly, we use *ModernBERT* fine-tuned on ADNI for IID and zero-shot on BRAINLAT for OOD, and all interpretability analyses are conducted on the 22nd layer of *ModernBERT*. We conducted extensive hyperparameter tuning for all components. The explanation optimizer performed best at a learning rate of 2e-4, with the optimal weight configuration $(\lambda_1, \lambda_2, \lambda_3, \lambda_4) = (0.1, 0.3, 0.1, 0.5)$. UMAP constraints were most effective at the 4× batch size level (4×64). Among the four SAE variants, TopK produced the strongest overall performance (Supplementary B.3; Figures 2, 4, 5), using a 32× feature depth. All models were trained using the AdamW optimizer with early stopping and standard evaluation metrics. Further hyperparameter tuning and evaluation details are provided in Supplementary section B.3 and B.2, B.4.

## 2.7 DATASET AND CODE AVAILABILITY

The dataset used in this study originates from the ADNI cohort Mueller et al. (2005) and is represented as text generated from multiple modalities, serving as the in-distribution (IID) dataset. We further split the generated text into nine subgroups based on input modality, as detailed in Supplementary Material B.1.5, for pattern analysis and biomarker research purposes. For the out-of-distribution (OOD) cohort, we used text generated from multimodal sources (MRI and clinical files) in the Latin American Brain Health Institute (BrainLat) dataset, a multi-site initiative providing neuroimaging, cognitive, and clinical data across several Latin American countries Prado et al. (2023). Additional demographic and preprocessing details are provided in Supplementary Sections B.1.1–B.1.5. The code is implemented in Python using PyTorch and runs on an NVIDIA cluster in one A100-SXM-80GB GPU. It leverages `SAE_LENs` Bloom et al. (2024) for SAE training, `quantus` Hedström et al. (2023) for evaluation, and `captum` for attribution analysis. The codebase `https://anonymous.4open.science/r/TEO_SAE-6D24/README.md`.

Table 1: Unified attribution summary split vertically by task and setting. Values are mean±std per class. Binary columns use (A/C) = Alzheimer/Control; Three-class columns use (L/M/C) = LMCI/MCI/Control, (A/F/C)= Alzheimer, Frontotemporal Dementia, Control;

| Task & Setting | Method (row = variant) | Sparseness | RIS | ROS |
|---|---|---|---|---|
| **Binary — IID (ADNI)**   Columns: (A/C) | | | | |
| | Gradient Activation | 0.3277/0.2500 ± 0.0384/0.0230 | 5.6149/5.6170 ± 0.0193/0.0221 | 16.9303/16.9347 ± 0.0034/0.0 |
| | Gradient Activation-SAE | 0.2035/0.1668 ± 0.0117/0.0072 | 5.6252/5.6173 ± 0.0213/0.0221 | 16.9343/16.9347 ± 6.8e−5/4.0e−5 |
| | DEO | 0.3383/0.3377 ± 0.0033/0.0017 | 9.2839/9.3131 ± 0.0800/0.1427 | 20.6342/20.6159 ± 0.0866/0.2026 |
| | DEO-SAE | 0.3374/0.3140 ± 0.0029/0.0010 | 9.2790/9.1750 ± 0.0646/0.1088 | 20.6150/20.5150 ± 0.0880/0.1299 |
| | TEO | **0.4220/0.4199 ± 0.0003/0.0005** | 5.0520/5.0688 ± 0.0192/0.0184 | 16.3529/16.3777 ± 0.0056/0.0011 |
| | TEO-SAE | 0.2672/0.2682 ± 0.0010/0.0007 | **1.6227/0.9964 ± 0.1708/0.2639** | **12.9250/12.2983 ± 0.1703/0.2613** |
| | TEO-UMAP *(SAE)* | 0.3989/0.4057 ± 0.0004/0.0003 | 5.4394/5.4709 ± 0.0332/0.1746 | 16.3037/16.2102 ± 0.0033/0.0079 |
| **Binary — OOD (BrainLat)**   Columns: (A/C) | | | | |
| | Gradient Activation-SAE | 0.1140/0.0630 ± 0.0177/0.0069 | 6.0328/6.0339 ± 0.0277/0.0398 | 16.9347/16.9348 ± 3.6e−6/3.7e−5 |
| | TEO-SAE | 0.2691/0.2725 ± 0.0016/0.0004 | **0.6835/0.4734 ± 0.6676/0.2801** | **11.5236/11.2130 ± 0.6591/0.5150** |
| | TEO-UMAP *(SAE)* | **0.3989/0.4043 ± 0.0005/0.0029** | 5.4394/5.4282 ± 0.0383/0.1944 | 16.3037/16.1577 ± 0.0039/0.1054 |
| **Three-class — IID (ADNI)**   Columns: (L/M/C) | | | | |
| | Gradient Activation | 0.3839/0.2917/0.2697 ± 0.0177/0.0200/0.0061 | 5.6272/5.6269/5.6290 ± 0.0212/0.0193/0.0225 | 16.9339/16.9340/16.9347 ± 0.0008/0.0006/0 |
| | Gradient Activation-SAE | **0.4310/0.2579/0.2296 ± 0.1156/0.1095/0.0036** | 5.6297/5.6172/5.6210 ± 0.9478/1.1684/0.0194 | 16.9347/16.9347/16.9348 ± 2.8625/3.5241/1.22e−5 |
| | TEO | 0.4131/0.3909/0.3918 ± 0.0003/0.0047/0.0008 | 5.0938/4.8283/4.8080 ± 0.0188/0.0377/0.0184 | 16.4043/16.1354/16.1172 ± 0.0024/0.0324/0.0090 |
| | TEO-SAE | 0.2860/0.2838/0.2682 ± 0.0374/0.0523/0.0649 | **2.2642/2.1617/1.5468 ± 0.4877/0.4547/0.1171** | **13.5646/13.4676/12.8570 ± 2.2745/2.7641/0.1179** |
| | TEO-UMAP *(SAE)* | 0.4161/0.4172/0.3973 ± 0.0870/0.2372/0.0749 | 5.1017/5.1116/5.1086 ± 0.1697/0.1072/0.2083 | 16.4031/16.4088/16.4123 ± 3.8616/0.4924/6.8439 |
| **Three-class — OOD (BrainLat)**   Columns: (A/F/C) | | | | |
| | Gradient Activation-SAE | 0.1836/0.4303/0.1772 ± 0.0016/0.0022/0.0112 | 6.0269/6.1450/6.1234 ± 0.0302/0.1210/0.1912 | 16.9348/16.9345/16.9346 ± 2.6e−6/2.8e−5/1.5e−5 |
| | TEO-SAE | 0.3716/0.4224/0.4162 ± 0.0009/0.0002/0.0029 | **4.9396/5.5421/5.7520 ± 0.0148/0.0611/0.3645** | **15.8121/16.2773/16.3792 ± 0.0099/0.0010/0.0034** |
| | TEO-UMAP *(SAE)* | **0.4239/0.4246/0.4238 ± 5.0e−5/2.4e−4/5.6e−4** | 5.4583/5.5576/5.5525 ± 0.0297/0.0954/0.1862 | 16.3726/16.3572/16.3661 ± 0.0017/0.0035/0.0073 |

# 3 RESULTS

## 3.1 ABLATION AND EVALUATION SCORES WITH/WITHOUT UMAP CONSTRAINT AND SAE LAYER.

In this study, sparseness is defined such that higher values correspond to more selective and concentrated attributions across input features—that is, greater sparseness. However, sparseness alone is insufficient to assess explanation quality, as it does not account for robustness or stability. Therefore, the most effective explanation method is one that simultaneously achieves high sparseness and low RIS and ROS values.

Across IID, OOD datasets and for both binary and three-class classification tasks, Table 1 shows a consistent stability–sparsity frontier governed by the proposed optimizers and a monosemantic bottleneck (SAE). In the binary IID case, SAE substantially improves stability for feature-learning explainers, most notably: Layer Conductance and especially TEO, with large drops in RIS/ROS for both Alzheimer's and Control, whereas Activation–SAE increases RIS/ROS relative to its no-SAE variant and is therefore less robust. In the binary OOD case, this pattern persists and strengthens: TEO–SAE achieves the lowest RIS/ROS overall (strong cross-dataset stability), while TEO–UMAP recovers higher sparsity (> 0.40) at a modest stability cost versus TEO–SAE, offering a tunable sparsity–stability trade-off. In the three-class IID setting, Feature Ablation is the sparsity leader across Control/LMCI/MCI (~ 0.52–0.53) with moderate, steady yet still high RIS/ROS values; Layer Conductance–SAE markedly reduces RIS/ROS for LMCI/MCI; and TEO–SAE again delivers the most stable attributions across all classes, albeit with reduced sparsity compared with no-SAE variants. The same rank ordering holds OOD. Across all blocks, gradient-formulaic methods (Grad-SHAP, Guided Backprop, Integrated Gradients) show near-invariant RIS/ROS (~ 5.6/ ~ 16.93) regardless of SAE, class, or domain, indicating that SAE chiefly benefits learned-attribution methods. Further analyses are provided in Supplementary §B.4, Tables 2–5.

## 3.2 INDIVIDUAL-LEVEL AND COHORT-LEVEL EXPLANATIONS AND PATTERNS

Figure 2 presents qualitative local-level and cohort-level attributions for the LMCI class in the three-class classification task on both IID and OOD settings using our proposed optimisers TEO, TEO-SAE, and TEO-UMAP (SAE). At the local level, the heatmaps illustrate feature importance, with tokens colour-coded to indicate relevance (green = positive relevance; red = negative relevance). Visually, TEO-SAE produces the tightest, least noisy explanations—fewer spurious highlights and clearer token groupings—consistent with its lowest RIS/ROS in Table 1. Adding the UMAP constraint restores higher sparseness while preserving much of TEO-SAE stability: TEO-UMAP (SAE) yields compact, well-separated patterns that remain clinically interpretable across IID

and OOD. Across classes and datasets, higher Sparseness corresponds to less diffuse maps with balanced positive/negative highlights, whereas low Sparseness with high RIS/ROS manifests as saturated red/green patches and unstable saliency (see Supplementary Figures 7–16). Among the six classical attribution methods (Activation, Layer Conductance, Feature Ablation, Gradient SHAP, Gradient Activation, Integrated Gradient), Feature Ablation attains the highest Sparseness but exhibits poorer stability (elevated RIS/ROS), a gap that worsens with SAE due to decoder-driven "decompression" (Supplementary Figures 7–8). Layer Conductance shows the opposite trade-off: SAE reduces Sparseness but improves stability (lower RIS/ROS), with similar stability gains observed for Gradient Activation, Integrated Gradient, and Gradient SHAP (Supplementary Figures 9–10). Overall, none of these classical methods match the proposed framework as TEO-SAE is consistently most stable, and TEO-UMAP (SAE) offers a tunable sparsity–stability compromise that generalises from ADNI to BrainLat (for extended analyses see Supplementary §B.6).

At the cohort level, we extracted UMAP embeddings to visualise patterns and text–category clusters in 2D space, observing any spreading effects or homogeneous clustering. We also applied PCA to identify high-contributing features (threshold 0.6 on the first component; Figure 2). Moving from TEO to TEO–SAE produces tighter, more homogeneous low-to-high attribution and the lowest RIS/ROS (highest stability), but also a marked reduction in sparseness, evident as broader token spread in the 2D manifold (see Figure 2). In some cases, this stabilisation concentrates signals so strongly that few features exceed the significance threshold (square box in the 1D scatter plot where PCA first component ≥ 0.6), and not all subgroups are represented (Figure 2, Supplementary Figure 32). Imposing a linear UMAP constraint (TEO–UMAP) mitigates this effect by restoring sparsity in significant attributions while retaining stability, yielding compact, clinically interpretable maps with more uniform subgroup coverage (Figure 2; Supplementary Figures 33–35). The behaviour of the proposed framework (TEO, TEO–SAE, TEO–UMAP) shows that higher sparseness corresponds to less diffuse, more balanced highlights, whereas lower sparseness with higher RIS/ROS results in saturated red/green patches. This mirrors the patterns observed across the six classical methods (Activation, Layer Conduction, Feature Ablation, Gradient SHAP, Gradient Activation, Integrated Gradient; Supplementary §B.10). With SAE, feature-learning explainers such as Layer Conduction generally gain stability (lower RIS/ROS) at some sparsity cost, while Feature Ablation maintains high sparsity but remains unstable. None, however, match the stability–sparsity trade-off achieved by TEO–SAE and TEO–UMAP (box plots in Figure 2; Table 1). Supplementary §B.7 (Figures 22–31) provides more details about cohort-level attributions.

### 3.3 Are monosemantic representation–based attribution methods statistically distinct from standard attribution techniques?

A statistical evaluation of interpretability metrics (sparseness, RIS, and ROS, see Supplementary §B5) across methods with and without the SAE layer was computed. In the binary ADNI task, *paired t-tests* with FDR correction showed that adding an SAE bottleneck significantly reduced Complexity ($p < 10^{-10}$) and RIS ($p < 10^{-4}$) in both groups, while ROS changes were small and inconsistent (marginal for Control, non-significant for Alzheimer's). The strongest SAE effects appeared in attribution metrics, with Gradient SHAP ($p < 10^{-45}$), Layer Conduction ($p = 3.2 \times 10^{-7}$), Integrated Gradients ($p < 10^{-55}$), and the TEO ($p < 10^{-95}$) all showing decisive reductions, confirming robust stability gains under SAE. In the three-class ADNI task, *paired t-tests and Wilcoxon signed-rank tests* (BH-FDR) indicated that the MCI group showed the clearest improvement: ROS decreased strongly ($t(17) = -10.12$, $p = 1.30 \times 10^{-8}$; $W = 0$, $p = 8.0 \times 10^{-6}$, $q = 2.3 \times 10^{-5}$), RIS showed a smaller reduction detected non-parametrically ($W = 19$, $p = 0.00117$), and Complexity increased modestly by Wilcoxon ($W = 17$, $p = 7.9 \times 10^{-4}$) while paired $t$-tests were non-significant. Control and LMCI had incomplete pairs, preventing matched testing with correction. Overall, SAE reliably improves attribution stability (lower RIS/ROS) and increase sparseness in binary tasks, with the three-class MCI group showing the most consistent ROS gains.

### 3.4 Clinical relevance in Alzheimer's disease progression: SAE-guided attribution produces more reliable and clinically coherent explanations

To assess the clinical relevance of our attribution framework and, in particular, the contribution of the SAE layer in producing monosemantic and diagnostically coherent explanations, we performed an auxiliary evaluation across three cohorts (Binary ADNI, Binary BrainLAT, and three-class

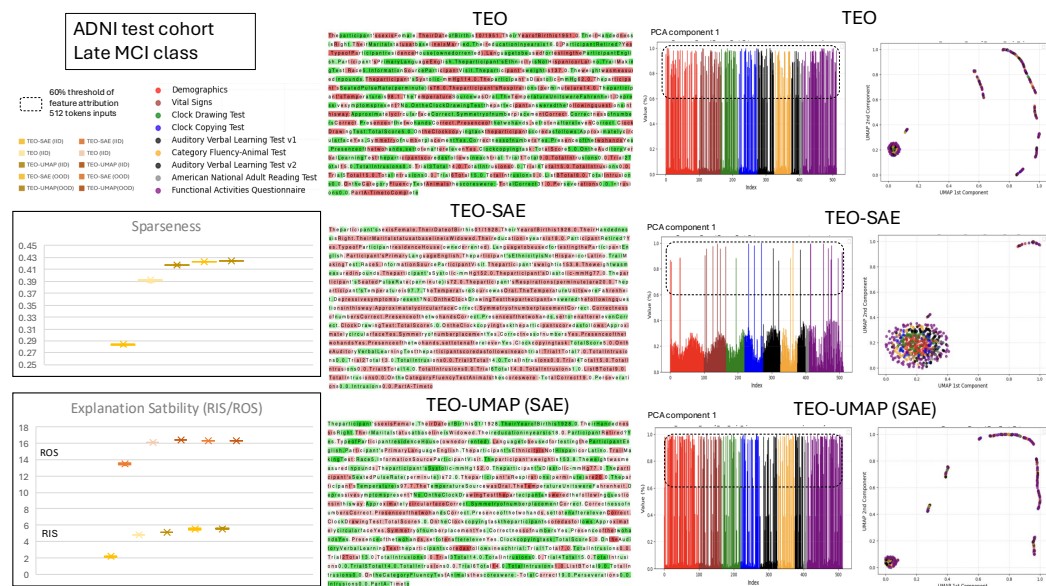

Figure 2: **Left:** Stability–sparsity frontier for explanation optimizers on ADNI (Late MCI) in the testing cohort. Scatter points show TEO, TEO–SAE, and TEO–UMAP (SAE) on ADNI (IID) and BrainLat (OOD). Metrics: Sparseness (higher is better) vs. RIS/ROS stability (lower is better). **Middle:** Token-level heatmap produced by the proposed framework, with feature-attribution scale (green: positive relevance; red: negative relevance; white: neutral). **Right:** 1st PCA and 2D UMAP projections of the full testing cohort. Thresholding uses 60% feature attribution over 512 tokens. The generated input text is split into nine subgroups (different colours) based on input modality, as detailed in Supplementary Material B.1.5, for pattern analysis and biomarker identification.

ADNI). For each test sample, we extracted the top 50% most influential token attributions from TEO without SAE, TEO with SAE, and TEO-UMAP, and constructed class-specific CSV files in which highlighted characters for each attribution method were arranged column-wise, with the full character sequence (including the CLS token) provided in the first column to ensure clear sample-level distinction. These CSVs were then presented to a large language model (ChatGPT-5.1 OpenAI (2024)) under a fixed prompting protocol to yield an external, model-agnostic assessment of the interpretability structure encoded by each explanation space. Across both binary tasks, all three attribution methods correctly identified the pathological Alzheimer's class; however, TEO without SAE consistently fixated on task artefacts (e.g., instruction counts or label-like patterns) rather than clinically meaningful biomarkers, whereas TEO-SAE and TEO-UMAP surfaced coherent neurocognitive indicators, demographic risk factors, and processing-speed impairments. In the more challenging three-class ADNI setting, the SAE-enabled variants again produced clearer diagnostic separation and more structured biomarker profiles, while TEO without SAE failed to identify pathology or highlight relevant clinical features. These results collectively demonstrate that imposing an SAE-driven monosemantic representation substantially strengthens the reliability, diagnostic validity, and clinical interpretability of the resulting attribution signals. Additional methodological details and full GPT prompts are provided in Supplementary Material B.8.

### 3.4.1 THE CLINICAL IMPACT AND OUTCOME IN THE DIAGNOSIS OF ALZHEIMER

**Biomarker Identification:** Our findings demonstrate that both TEO-SAE and TEO-UMAP yield the most reliable and consistent identification of informative sources across the nine multimodal subgroups. Using a threshold of 0.6 on the first principal component (PCA; Figure 2), we observe systematic and class-specific biomarker patterns in the IID binary task (ADNI). For the Control group, TEO-SAE is primarily driven by FAQ, whereas TEO-UMAP shifts emphasis toward DEM, AVLT2, and FAQ. For the Alzheimer's group, TEO prioritises FAQ, AVLT1, and CFA, while TEO-UMAP highlights ANART, FAQ, and DEM. In the three-class setting, TEO identifies AVLT1, CDT, and ANART as most influential for Controls, while TEO-UMAP selects AVLT1, CDT, and CFA. For

MCI, TEO assigns highest relevance to CCT, AVLT2, and FAQ, whereas TEO-UMAP favours AVLT2, ANART, and CFA. For LMCI, TEO elevates AVLT1, FAQ, and CDT, while TEO-UMAP elevates FAQ, ANART, and AVLT2. These trends are summarised in Supplementary Table 6 (Section B.9), with acronym definitions provided in Sections B.1.5 (Table 1).

Importantly, these results illustrate how our proposed mechanistic attribution framework can disentangle which demographic/vital features and, crucially, which cognitive assessments contribute most strongly to each diagnostic category. This is particularly relevant in clinical neuroscience, where cognitive tests are often time-consuming, resource-intensive, and susceptible to demographic or cultural bias. Our framework provides a principled approach for identifying the most informative cognitive biomarkers-such as AVLT, FAQ, CDT, and ANART-which have been repeatedly linked to early Alzheimer's progression in the literature (Petersen et al., 1999), (Bondi et al., 2019). By pinpointing the minimal set of high-yield assessments for each classification task, the method supports more efficient screening pipelines, reduces clinical burden, and enables scalable application in larger cohorts where repeated or comprehensive testing is impractical.

### 3.5 Limitation and Future Work

Although our framework demonstrates substantial improvements in attribution clarity and robustness, some limitations remain. A generalized outcome about clinical LLMs is not feasible at the level of this study, as the analysis was restricted to the neurodegenerative domain, limiting generalisability to other areas such as oncology. Constraining the manifold space of explanations with explicit guidance from clinical experts could further improve explanation quality and enhance pattern discovery within the proposed framework. In addition, while stability–sparsity assessment focused on RIS/ROS and sparsity indices as important first-level evaluation metrics, additional measures such as uncertainty quantification and fairness auditing should be incorporated in future work. Future work will further strengthen these results by integrating clinical experts into the loop to refine a more analytical vocabulary and to further characterize the enriched patterns identified by the proposed framework. Lastly, we aim to prospectively validate the approach, extend it to additional centres, modalities, and clinical domains (e.g., oncology), explore alternative constraints, and incorporate uncertainty and fairness auditing.

## 4 Conclusion

We presented a unified interpretability framework that combines monosemantic feature extraction with learning-based explainer optimisation (TEO–SAE) and an optional geometry-aware constraint (TEO–UMAP). Across IID (ADNI) and OOD (BrainLat) cohorts, and in both binary and three-class tasks, TEO–SAE consistently achieved the most stable explanations (lowest RIS/ROS), while TEO–UMAP offered a complementary high-sparsity regime with only a modest reduction in stability—together defining a tunable sparsity–stability frontier that generalises across distribution shifts. Classical attribution methods benefited unevenly from the monosemantic bottleneck: gradient-based techniques changed little, whereas feature-learning explainers such as Layer Conductance showed substantial gains, though none matched the robustness of our optimisers. Clinically, the framework reliably identified meaningful diagnostic markers, with stable contributions centred on functional status, memory performance, and visuospatial abilities. TEO–SAE highlighted core neuropsychological indicators, while TEO–UMAP revealed complementary demographic and language-related signals. At the cohort level, a simple thresholding rule on the 2D UMAP embeddings produced actionable maps that prioritise high-yield assessments, streamline diagnostic workflows, reduce clinical burden, and enable scalable deployment in large or resource-limited settings. Overall, integrating monosemantic encoding with geometry-aware explanation optimisation offers substantially more robust, coherent, and clinically aligned explanations than standard attribution methods, charting a principled path toward trustworthy, human-aligned interpretability for LLMs in Alzheimer's disease progression assessment.

### Reproducibility Statement

For reproducibility during review, anonymised source codes, results and datasets are included in the supplementary material and in the manuscript subsection 2.7 'Dataset and code availability' of the main manuscript.

ETHICS STATEMENT

The paper discusses several potential positive societal impacts, particularly emphasizing its relevance to clinical applications such as the early diagnosis and treatment planning of Alzheimer's Disease. By proposing a unified interpretability framework that combines attributional and mechanistic techniques, the authors aim to enhance the trustworthiness, consistency, and human alignment of large language model (LLM) outputs. This improved interpretability is presented as a means to support safer and more effective integration of LLMs into cognitive health and clinical decision-making, with the potential to uncover clinically meaningful patterns and ultimately improve patient outcomes. However, the paper does not explicitly address possible negative societal impacts of the work. It does not discuss risks such as the misinterpretation of model ex-677 planations, over-reliance on machine-generated insights in high-stakes medical contexts, or the potential for the framework to inadvertently reinforce biases embedded in training data. Societal impacts can be better established through future work, in which we plan to incorporate clinician-in-the-loop evaluation and patients.

ACKNOWLEDGMENTS

Data collection and sharing for the Alzheimer's Disease Neuroimaging Initiative (ADNI) is funded by the National Institute on Aging (National Institutes of Health Grant U19AG024904). The grantee organization is the àNorthern California Institute for Research and Education. Past funding was obtained from: the National Institute of Biomedical Imaging and Bioengineering, the Canadian Institutes of Health Research, and private sector contributions through the Foundation for the National Institutes of Health (FNIH) including generous contributions from the following: AbbVie, Alzheimer's Association; Alzheimer's Drug Discovery Foundation; Araclon Biotech; BioClinica, Inc.; Biogen; BristolMyers Squibb Company; CereSpir, Inc.; Cogstate; Eisai Inc.; Elan Pharmaceuticals, Inc.; Eli Lilly and Company; EuroImmun; F. Hoffmann-La Roche Ltd and its affiliated company Genentech, Inc.; Fujirebio; GE Healthcare; IXICO Ltd.; Janssen Alzheimer Immunotherapy Research Development, LLC.; Johnson Johnson Pharmaceutical Research Development LLC.; Lumosity; Lundbeck; Merck Co., Inc.; Meso Scale Diagnostics, LLC.; NeuroRx Research; Neurotrack Technologies; Novartis Pharmaceuticals Corporation; Pfizer Inc.; Piramal Imaging; Servier; Takeda Pharmaceutical Company; and Transition Therapeutics.

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
