# Toward Monosemantic Clinical Explanations for Alzheimer's Diagnosis via Attribution and Mechanistic Interpretability

## A    Technical Appendices

### A.1    Attributional theory and methods

Attribution explainability methods follow the framework of additive feature attribution, where the explanation model $g(f, \mathbf{x})$ is represented as a linear function of simplified input features:

$$g(f, \mathbf{x}) = \phi_0 + \sum_{i=1}^{M} \phi_i x_i \qquad (1)$$

Here, $f$ is the predictive model, $\phi_i \in \mathbb{R}$ is the attribution (importance) assigned to feature $x_i$, and $M$ is the number of simplified input features.

For this study, we employed six well-established attributional interpretability methods applied to large language models (LLMs), denoted as $K = 6$: *Feature Ablation*, *Layer Activations* (which capture the embedding activation space of a specific layer of interest within the LLM), *Layer DeepLIFT SHAP*, *Layer Gradient SHAP* (Lundberg & Lee, 2017), *Layer Integrated Gradients* (Sundararajan et al., 2017), and *Layer Gradient × Activation.*

To align these layer-wise interpretability methods with the additive feature attribution framework, we reinterpret the internal activations (i.e., latent units) of a network layer $L$ as simplified input features. The objective is to estimate an attribution score $\phi_i$ for each unit, where $\phi_i \in \mathbb{R}$ quantifies the contribution of the corresponding neuron to the model's prediction.

**Layer SHAP implementations:** This directly corresponds to the Shapley formulation:

$$\phi_i = \sum_{S \subseteq F \setminus \{i\}} \frac{|S|!(|F| - |S| - 1)!}{|F|!} \left[ f_{S \cup \{i\}}(x_{S \cup \{i\}}) - f_S(x_S) \right] \qquad (2)$$

In practice, Deep SHAP approximates this using sampling and a chain-rule based linearization over network layers Lundberg & Lee (2017). *Gradient SHAP* assumes that input features are independent and that the explanation model is linear, allowing explanations to be expressed as an additive composition of feature contributions. Under these assumptions, SHAP values (Lundberg & Lee, 2017) can be approximated by computing the expected gradients over a distribution of perturbed inputs. Specifically, Gaussian noise is added to each input feature to generate multiple baseline samples, and the resulting gradients are averaged to approximate SHAP attributions.

**Activation Attribution:** This method treats the raw activation $a_i^L(\mathbf{x})$ as proportional to its importance in the output. In the additive form:

$$\phi_i = a_i^L(\mathbf{x}) \qquad (3)$$

Assuming linearity between layer $L$ and the output, activations themselves serve as proxy contributions.

**Gradient × Activation Attribution:** This method computes the element-wise product between the activation values and the gradients of the model output with respect to those activations, thereby

capturing the first-order sensitivity of the output to the neurons in the layer. To this end, the method estimates the first-order sensitivity of the output with respect to the activation:

$$\phi_i = a_i^L(\mathbf{x}) \cdot \frac{\partial f}{\partial a_i^L}(\mathbf{x}) \tag{4}$$

This corresponds to a local linear approximation (first-order Taylor expansion) of the model at $\mathbf{x}$, akin to DeepLIFT and the SHAP linearization used in DeepLift SHAP (Lundberg & Lee, 2017).

**Feature Ablation Attribution:** This attributional interpretability technique is a perturbation-based approach to estimating attributions. It involves replacing the input or output values of a selected layer with a given baseline or reference value and computing the resulting change in the model's output. By default, each neuron (i.e., scalar input or output value) within the layer is ablated independently. For neuron group $S \subseteq \{1, \ldots, d_L\}$, the perturbed activation is:

$$\tilde{a}_i^L = \begin{cases} b_i^L & \text{if } i \in S, \\ a_i^L(\mathbf{x}) & \text{otherwise,} \end{cases} \tag{5}$$

and the attribution is the marginal effect:

$$\phi_S = f\left(\mathbf{x}; \tilde{\mathbf{a}}_S^L\right) - f(\mathbf{x}) \tag{6}$$

All attribution methods were applied to the final (22nd) layer of the MODERN-BERT LLM—the model variant that achieved the highest classification accuracy in our evaluations (see Suplementary material section 1.1). These formulations allow us to ground various neural attribution techniques within a unified additive explanation model, facilitating their comparison and hybridization under shared theoretical assumptions.

## A.2 ATTRIBUTIONAL EXPLANATION OPTIMIZER FRAMEWORK

Let $\mathcal{A} = \{A_1, A_2, \ldots, A_K\}$ denote the set of $K = 6$ attribution methods applied to the final layer $L$ of the model $f$. Each method $A_k$ generates an attribution vector $\boldsymbol{\phi}^{(k)} = [\phi_1^{(k)}, \phi_2^{(k)}, \ldots, \phi_M^{(k)}]$, where $M$ is the number of latent features (neurons) in layer $L$. The goal is to derive a unified attribution vector $\bar{\boldsymbol{\phi}}$ that captures the consensus explanation across methods.

### A.2.1 SCORING AND WEIGHTING ATTRIBUTION METHODS

Each attribution vector $\boldsymbol{\phi}^{(k)}$ is evaluated using the following quality metrics:

### A.2.2 EVALUATION INTERPRETABILITY METRICS

We evaluate the robustness of each attribution method $A_k$ using the following stability metrics:

**Relative Input Stability (RIS):**

$$M_{\text{RIS}}^{(k)} = \text{RIS}(f, \boldsymbol{\phi}^{(k)}; \mathbf{x}) = \frac{\|\mathbf{x}\|_p}{\|\boldsymbol{\phi}^{(k)}(\mathbf{x})\|_p} \max_{\mathbf{x}' \in \mathcal{N}_\mathbf{x}, \hat{y}_{\mathbf{x}'} = \hat{y}_\mathbf{x}} \frac{\|\boldsymbol{\phi}^{(k)}(\mathbf{x}) - \boldsymbol{\phi}^{(k)}(\mathbf{x}')\|_p}{\|\mathbf{x} - \mathbf{x}'\|_p} \tag{7}$$

**Relative Output Stability (ROS):**

$$M_{\text{ROS}}^{(k)} = \text{ROS}(f, \boldsymbol{\phi}^{(k)}; \mathbf{x}) = \frac{\|f(\mathbf{x})\|_p}{\|\boldsymbol{\phi}^{(k)}(\mathbf{x})\|_p} \max_{\mathbf{x}' \in \mathcal{N}_\mathbf{x}, \hat{y}_{\mathbf{x}'} = \hat{y}_\mathbf{x}} \frac{\|\boldsymbol{\phi}^{(k)}(\mathbf{x}) - \boldsymbol{\phi}^{(k)}(\mathbf{x}')\|_p}{\|f(\mathbf{x}) - f(\mathbf{x}')\|_p} \tag{8}$$

Here, $\mathcal{N}_\mathbf{x}$ denotes a neighborhood of perturbed inputs $\mathbf{x}'$ around $\mathbf{x}$, and $\hat{y}_\mathbf{x}$ is the predicted class label. Both metrics measure the relative sensitivity of the attribution vector $\boldsymbol{\phi}^{(k)}$ to perturbations in the input or output space.

**Sparseness Metric:** We quantify the **sparseness** of the attribution vector $\boldsymbol{\phi}^{(k)} \in \mathbb{R}^d$ using the *Gini Index*, a measure of inequality that has been shown to satisfy several desirable properties for evaluating sparseness Chalasani et al. (2020). This formulation is adopted in the context of explaining neural network predictions Chalasani et al. (2020).

Let $v \in \mathbb{R}_{\geq 0}^d$ be a non-negative vector. Denote by $v_{(k)}$ the $k$-th smallest element in $v$ after sorting it in non-decreasing order. Then, the **Gini Index** $G(v) \in [0,1]$ is defined as:

$$G(v) = 1 - 2 \sum_{k=1}^{d} \frac{v_{(k)}}{\|v\|_1} \cdot \left( \frac{d - k + 0.5}{d} \right), \tag{9}$$

where $\|v\|_1 = \sum_{i=1}^{d} v_i$ is the $\ell_1$-norm of $v$. To evaluate the sparseness of an attribution vector $\boldsymbol{\phi}^{(k)}$, we apply the Gini Index to the vector of its absolute values:

$$\text{Sparseness}\left(\boldsymbol{\phi}^{(k)}\right) = G\left(\left|\boldsymbol{\phi}^{(k)}\right|\right),$$

where $\left|\boldsymbol{\phi}^{(k)}\right| = \left(|\phi_1^{(k)}|, |\phi_2^{(k)}|, \ldots, |\phi_d^{(k)}|\right)$.

Higher values of $G\left(\left|\boldsymbol{\phi}^{(k)}\right|\right)$ indicate greater sparseness. In the extreme case, if only one component is non-zero, the Gini Index reaches its maximum value of 1, indicating perfect sparseness. If all components are equal, the Gini Index is 0.

### A.2.3 AGGREGATION OF ATTRIBUTIONS

The weighted average attribution vector $\bar{\boldsymbol{\phi}}$ is calculated as:

$$\bar{\boldsymbol{\phi}} = \sum_{k=1}^{K} w_k \cdot \boldsymbol{\phi}^{(k)} \tag{10}$$

This vector serves as the target explanation for the optimization process.

### A.2.4 EXPLANATION RECONSTRUCTION VIA ENCODER–DECODER MODELS

An encoder–decoder model is trained to generate a reconstructed explanation $\hat{\boldsymbol{\phi}}$ from the original input $\boldsymbol{x}$. Two architectures are considered the Diffusion UNet1D Ronneberger et al. (2015) and the x-transformer autoencoder Vaswani et al. (2017); Nguyen & Salazar (2019).

**Diffusion model:** The diffusion model follows the basic structure of a 1-dimensional U-Net and is trained using diffusion principles. In this framework, diffusion models Ho et al. (2020) are latent variable models in which the observed data $\phi_0^{(k)}$ is gradually corrupted through a forward noising process, producing a sequence of latent variables $\phi_{1:T}^{(k)}$. A corresponding reverse process is then learned to recover the original data from noise. The mathematical formulation is as follows:

FORWARD PROCESS: A fixed Markov chain progressively adds Gaussian noise to the data:

$$q(\phi_{1:T}^{(k)}|\phi_0^{(k)}) := \prod_{t=1}^{T} q(\phi_t^{(k)}|\phi_{t-1}^{(k)}), \quad q(\phi_t^{(k)}|\phi_{t-1}^{(k)}) := \mathcal{N}(\phi_t^{(k)}; \sqrt{1 - \beta_t}\phi_{t-1}^{(k)}, \beta_t \mathbf{I}) \tag{11}$$

Alternatively, sampling from the forward process at an arbitrary timestep $t$ is possible in closed form:

$$q(\phi_t^{(k)}|\phi_0^{(k)}) = \mathcal{N}(\phi_t^{(k)}; \sqrt{\bar{\alpha}_t}\phi_0^{(k)}, (1 - \bar{\alpha}_t)\mathbf{I}), \tag{12}$$

where $\alpha_t := 1 - \beta_t$ and $\bar{\alpha}_t := \prod_{s=1}^{t} \alpha_s$.

REVERSE PROCESS: A learned time-reversal model with Gaussian transitions:

$$p_\theta(\phi_{0:T}^{(k)}) := p(\phi_T^{(k)}) \prod_{t=1}^{T} p_\theta(\phi_{t-1}^{(k)}|\phi_t^{(k)}), \quad p_\theta(\phi_{t-1}^{(k)}|\phi_t^{(k)}) := \mathcal{N}(\phi_{t-1}^{(k)}; \boldsymbol{\mu}_\theta(\phi_t^{(k)}, t), \Sigma_\theta(\phi_t^{(k)}, t)), \tag{13}$$

where $p(\phi_T^{(k)}) := \mathcal{N}(\phi_T^{(k)}; \mathbf{0}, \mathbf{I})$.

TRAINING OBJECTIVE: The training objective of diffusion models is based on a variational bound, which includes Kullback–Leibler (KL) divergence terms. The KL term comparing the true posterior from the forward process and the model's learned reverse process is written as:

$$\mathrm{KL}\left(q(\phi_{t-1}^{(k)} \mid \phi_t^{(k)}, \phi_0^{(k)}) \,\|\, p_\theta(\phi_{t-1}^{(k)} \mid \phi_t^{(k)})\right) \tag{14}$$

Both distributions are Gaussian:

$$q(\phi_{t-1}^{(k)} \mid \phi_t^{(k)}, \phi_0^{(k)}) = \mathcal{N}(\phi_{t-1}^{(k)}; \tilde{\mu}_t(\phi_t^{(k)}, \phi_0^{(k)}), \tilde{\beta}_t \mathbf{I}) \tag{15}$$

$$p_\theta(\phi_{t-1}^{(k)} \mid \phi_t^{(k)}) = \mathcal{N}(\phi_{t-1}^{(k)}; \mu_\theta(\phi_t^{(k)}, t), \sigma_t^2 \mathbf{I}) \tag{16}$$

The closed-form KL divergence between two Gaussians $\mathcal{N}(\mu_1, \sigma_1^2 \mathbf{I})$ and $\mathcal{N}(\mu_2, \sigma_2^2 \mathbf{I})$ in $d$-dimensions is:

$$\mathrm{KL} = \frac{1}{2}\left[\log\left(\frac{\sigma_2^2}{\sigma_1^2}\right) + \frac{\sigma_1^2 + \|\mu_1 - \mu_2\|^2}{\sigma_2^2} - d\right] \tag{17}$$

In our setting, this term is computed for each timestep $t$ and summed across all steps:

$$\mathcal{L}_{1:T-1} = \sum_{t=2}^{T} \mathbb{E}_{q(\phi_0^{(k)}, \phi_t^{(k)})}\left[\mathrm{KL}\left(q(\phi_{t-1}^{(k)} \mid \phi_t^{(k)}, \phi_0^{(k)}) \,\|\, p_\theta(\phi_{t-1}^{(k)} \mid \phi_t^{(k)})\right)\right] \tag{18}$$

This forms a core part of the evidence lower bound (ELBO) optimized during training. Using variational inference, we minimize the negative ELBO:

$$\mathcal{L} = \mathbb{E}_q\left[-\log p(\phi_T^{(k)}) + \sum_{t=1}^{T} \mathrm{KL}\left(q(\phi_{t-1}^{(k)} | \phi_t^{(k)}, \phi_0^{(k)}) \,\|\, p_\theta(\phi_{t-1}^{(k)} | \phi_t^{(k)})\right) - \log p_\theta(\phi_0^{(k)} | \phi_1^{(k)})\right]. \tag{19}$$

Each KL term compares Gaussian distributions and can be computed in closed form. The posterior $q(\phi_{t-1}^{(k)} | \phi_t^{(k)}, \phi_0^{(k)})$ is also Gaussian:

$$q(\phi_{t-1}^{(k)} | \phi_t^{(k)}, \phi_0^{(k)}) = \mathcal{N}(\phi_{t-1}^{(k)}; \tilde{\boldsymbol{\mu}}_t(\phi_t^{(k)}, \phi_0^{(k)}), \tilde{\beta}_t \mathbf{I}), \tag{20}$$

with:

$$\tilde{\boldsymbol{\mu}}_t(\phi_t^{(k)}, \phi_0^{(k)}) = \frac{\sqrt{\bar{\alpha}_{t-1}}\beta_t}{1 - \bar{\alpha}_t}\phi_0^{(k)} + \frac{\sqrt{\alpha_t}(1 - \bar{\alpha}_{t-1})}{1 - \bar{\alpha}_t}\phi_t^{(k)}, \tag{21}$$

$$\tilde{\beta}_t = \frac{1 - \bar{\alpha}_{t-1}}{1 - \bar{\alpha}_t}\beta_t. \tag{22}$$

SIMPLIFIED TRAINING LOSS: The common parameterization rewrites the objective as denoising score matching:

$$\mathcal{L}_{\mathrm{simple}}(\theta) := \mathbb{E}_{t, \phi_0^{(k)}, \boldsymbol{\epsilon}}\left[\left\|\boldsymbol{\epsilon} - \boldsymbol{\epsilon}_\theta(\sqrt{\bar{\alpha}_t}\phi_0^{(k)} + \sqrt{1 - \bar{\alpha}_t}\boldsymbol{\epsilon}, t)\right\|^2\right], \tag{23}$$

where $\boldsymbol{\epsilon} \sim \mathcal{N}(0, \mathbf{I})$ and $\boldsymbol{\epsilon}_\theta$ is the neural network trained to predict noise.

In our implementation we compute the total loss for the diffusion model as:

$$\mathcal{L}_{\mathrm{similarity}}(\hat{\boldsymbol{\phi}}, \bar{\boldsymbol{\phi}}) = \mathcal{L}_{\mathrm{similarity}}(\theta) = \frac{1}{K+1}\sum_{l=0}^{K} \mathcal{L}_{\mathrm{simple}}^{(l)}(\theta) \tag{24}$$

**x-Transformer:** Let the input sequence be:

$$\boldsymbol{\phi}^{(k)} = [\boldsymbol{\phi}_1^{(k)}, \boldsymbol{\phi}_2^{(k)}, \ldots, \boldsymbol{\phi}_T^{(k)}] \in \mathbb{R}^{T \times d_{\mathrm{in}}}$$

where $d_{\mathrm{in}} = 7$ is the input dimensionality and $T = 512$ is the sequence length. We consider a Transformer-based encoder-decoder architecture operating on input sequences $\Phi^{(k)} \in \mathbb{R}^{B \times T \times d_{\mathrm{in}}}$

at diffusion step $k$, where: $B$ is the batch size, $T$ is the sequence length, $d_{\text{in}}$ is the input feature dimension, and $\Phi^{(k)}$ is the input sequence at step $k$.

The processing pipeline is mathematically formulated as follows:

INPUT PROJECTION AND POSITIONAL ENCODING: We first project the input to the model dimension $d$ and add positional encodings:

$$\mathbf{X}_0 = \mathbf{W}_{\text{in}}\Phi^{(k)} + \mathbf{P}, \quad \mathbf{X}_0 \in \mathbb{R}^{B \times T \times d} \tag{25}$$

where: $\mathbf{W}_{\text{in}} \in \mathbb{R}^{d_{\text{in}} \times d}$ is a learnable linear projection matrix, and $\mathbf{P} \in \mathbb{R}^{1 \times T \times d}$ is a learnable positional embedding matrix.

ENCODER: MULTI-HEAD SELF-ATTENTION LAYERS: The encoder consists of $L_e$ stacked multi-head self-attention (MHSA) layers:

$$\mathbf{H}_{\text{enc}} = \text{MHSA}_{L_e} \circ \cdots \circ \text{MHSA}_1(\mathbf{X}_0) \tag{26}$$

where each MHSA layer performs:

$$\text{MHSA}(\mathbf{X}) = \text{Softmax}\left(\frac{\mathbf{Q}\mathbf{K}^\top}{\sqrt{d_h}}\right)\mathbf{V} \tag{27}$$

with: $\mathbf{Q}, \mathbf{K}, \mathbf{V}$: Query, Key, and Value matrices obtained via learned linear projections, and $d_h$: the dimensionality of each attention head.

DECODER INPUT PROJECTION: During training, the decoder may receive the ground-truth output $\Phi_{\text{target}}^{(k)} \in \mathbb{R}^{B \times T \times 1}$:

$$\mathbf{Y}_0 = \mathbf{W}_{\text{dec}}\Phi_{\text{target}}^{(k)} + \mathbf{P} \tag{28}$$

where $\mathbf{W}_{\text{dec}} \in \mathbb{R}^{1 \times d}$ is a projection matrix.

If no decoder input is available (e.g., during inference), $\Phi_{\text{target}}^{(k)}$ is initialized to a zero tensor.

DECODER MHSA + CROSS-ATTENTION LAYERS: The decoder consists of $L_d$ layers of MHSA followed by cross-attention (CA) using the encoder context:

$$\mathbf{H}_{\text{dec}} = \text{CA}_{L_d} \circ \cdots \circ \text{CA}_1\left(\text{MHSA}_{L_d} \circ \cdots \circ \text{MHSA}_1(\mathbf{Y}_0) \,\middle|\, \mathbf{H}_{\text{enc}}\right) \tag{29}$$

Each cross-attention (CA) layer uses the decoder hidden state as the query and encoder output as the key and value:

$$\text{CA}(\mathbf{Y}, \mathbf{H}_{\text{enc}}) = \text{Softmax}\left(\frac{\mathbf{Q}_{\text{dec}}\mathbf{K}_{\text{enc}}^\top}{\sqrt{d_h}}\right)\mathbf{V}_{\text{enc}} \tag{30}$$

OUTPUT PROJECTION: Finally, the decoder output is projected back to the target dimension:

$$\hat{\Phi}^{(k)} = \mathbf{W}_{\text{out}}\mathbf{H}_{\text{dec}}, \quad \hat{\Phi}^{(k)} \in \mathbb{R}^{B \times T \times 1} \tag{31}$$

where $\mathbf{W}_{\text{out}} \in \mathbb{R}^{d \times 1}$ is a linear projection matrix.

The similarity cost function is given by the Mean Squared Error (MSE) loss between the predicted output of the x-Transformer and the target weighted attribution vector as follow:

$$\mathscr{L}_{\text{similarity}}(\hat{\boldsymbol{\phi}}, \bar{\boldsymbol{\phi}}) = \mathscr{L}_{\text{MSE}} = \frac{1}{T}\sum_{t=1}^{T}\left\|\hat{\boldsymbol{\phi}}_t - \bar{\boldsymbol{\phi}}_t\right\|^2, \tag{32}$$

### A.3 THE TOTAL COST FUNCTION OF THE OPTIMIZER

As previously highlighted, the reconstruction of the optimal explanation and the associated cost function adhere to the same principles and architectural design outlined in Mamalakis et al. (2025). The cost function consists of three key components: sparseness, as defined in **?**; ROS and RIS scores Agarwal et al. (2022); and similarity. The integration of these components ensures a robust and interpretable evaluation. The total cost function for training the reconstruction model is:

$$\begin{aligned}
\mathscr{L}_{\text{total}}(\boldsymbol{\phi}^{(k)}, \hat{\boldsymbol{\phi}}) = &\lambda_1 \cdot \frac{1}{M_{\text{RIS}}(f, \hat{\boldsymbol{\phi}})} + \lambda_2 \cdot \frac{1}{M_{\text{ROS}}(f, \hat{\boldsymbol{\phi}})} \\
&+ \lambda_3 \cdot M_{\text{sparse}}(f, \hat{\boldsymbol{\phi}}) + \lambda_4 \cdot \mathscr{L}_{\text{similarity}}(\hat{\boldsymbol{\phi}}, \bar{\boldsymbol{\phi}})
\end{aligned} \tag{33}$$

where:$\lambda_1, \lambda_2, \lambda_3, \lambda_4$ are hyperparameters controlling the influence of each loss term. This formulation enables a principled and quantitative integration of multiple attribution methods, optimizing toward a robust and interpretable explanation.

### A.4 THE UMAP EXTRACTION AND THE LINEAR CONSTRAIN

Given a dataset $\hat{\mathbf{\Phi}} = \{\hat{\boldsymbol{\phi}}_1, \hat{\boldsymbol{\phi}}_2, \ldots, \hat{\boldsymbol{\phi}}_n\} \subset \mathbb{R}^D$, UMAP aims to find a low-dimensional embedding $U = \{u_1, u_2, \ldots, u_n\} \subset \mathbb{R}^d$ where typically $d = 2$ or $d = 3$, such that the local topological structure of the data in $\hat{\mathbf{\Phi}}$ is preserved in $U$.

HIGH-DIMENSIONAL GRAPH CONSTRUCTION: First, the algorithm constructs a k-nearest neighbors graph in the high-dimensional space $\hat{\mathbf{\Phi}}$. The distance metric used to calculate the pairwise distances is typically Euclidean:

$$d(\hat{\boldsymbol{\phi}}_i, \hat{\boldsymbol{\phi}}_j) = \|\hat{\boldsymbol{\phi}}_i - \hat{\boldsymbol{\phi}}_j\|_2$$

Next, a conditional probability is defined between points $\hat{\boldsymbol{\phi}}_i$ and $\hat{\boldsymbol{\phi}}_j$ using a Gaussian distribution:

$$p_{ij} = \exp\left(-\frac{\|\hat{\boldsymbol{\phi}}_i - \hat{\boldsymbol{\phi}}_j\|^2}{\sigma_i^2}\right)$$

where $\sigma_i$ is the bandwidth for the Gaussian distribution, determined through a binary search to match a fixed perplexity.

The graph is symmetrized:

$$P_{ij} = \frac{p_{ij} + p_{ji}}{2}$$

LOW-DIMENSIONAL EMBEDDING GRAPH: In the low-dimensional space, a similar probability is defined between points $u_i$ and $u_j$:

$$q_{ij} = \frac{1}{1 + a\|u_i - u_j\|^{2b}}$$

where $a$ and $b$ are hyperparameters that control the shape of the distribution, and $\|u_i - u_j\|_2$ is the Euclidean distance between points in the low-dimensional embedding.

OBJECTIVE FUNCTION: The optimization process involves minimizing the cross-entropy between the high-dimensional and low-dimensional probability distributions:

$$\mathcal{L} = \sum_{i<j} \left[ P_{ij} \log(Q_{ij}) + (1 - P_{ij}) \log(1 - Q_{ij}) \right]$$

This loss function encourages points that are close in the high-dimensional space to be close in the low-dimensional space, and points that are distant to remain distant.

OPTIMIZATION PROCESS: The optimization is carried out using stochastic gradient descent (SGD), updating the embedding points $\{u_i\}$ iteratively based on the gradient of the loss function $\mathcal{L}$. The gradient updates for the low-dimensional embedding $u_i$ are computed as follows:

$$\frac{\partial \mathcal{L}}{\partial u_i} = -\sum_{j \neq i} \left( P_{ij} - Q_{ij} \right) \frac{u_i - u_j}{\|u_i - u_j\|_2^2}$$

REGULARIZATION CONSTRAINT: To prevent the embedding from collapsing to a single point, we introduce a variance constraint to ensure that the variance of the embedding does not approach zero:

$$\mathrm{Var}(U) = \frac{1}{n} \sum_{i=1}^{n} \| u_i - \bar{u} \|_2^2 \geq \epsilon$$

where $\bar{U} = \frac{1}{n} \sum_{i=1}^{n} u_i$ is the mean of the embeddings, and $\epsilon > 0$ is a small constant that enforces a lower bound on the variance.

APPLICATION OF UMAP IN OUR PROBLEM: To obtain a comparable low-dimensional representation of the attribution scores across all tokenizer features, we applied a feature-wise UMAP projection procedure to the normalized attribution matrix. For each attribution method, the attribution tensor has shape $\mathbb{R}^{M \times T}$, where $M$ denotes the number of test samples in the evaluation cohort and $T$ corresponds to the dimensionality of the tokenizer embedding space of the input text. For each feature $j \in \{1, \ldots, T\}$, we first applied min–max normalization to the feature-specific attribution vector

$$\mathbf{x}^{(j)} \in \mathbb{R}^{M},$$

and subsequently performed a one-dimensional UMAP projection to obtain a two-dimensional embedding

$$\mathbf{y}^{(j)} \in \mathbb{R}^{M \times 2}.$$

The resulting coordinates were then normalized to the interval $[0,1]$ to ensure that all feature-wise embeddings share a common bounded range. This procedure preserves the relative neighborhood structure of the $M$-sample attribution distribution for each feature while mapping all $T$ features into a comparable two-dimensional representation space.

The motivation for applying UMAP independently to each of the $T$ tokenizer features is to ensure that all attribution methods are projected into an aligned and comparable representation space. Since each attribution method produces values defined over the same token embedding dimensions, a feature-wise nonlinear projection enables consistent cross-method comparison of attribution patterns within the shared tokenizer feature space.

LINEAR CONSTRAINT FOR EQUAL COMPONENTS IN UMAP: Let $u_i = (u_{i1}, u_{i2}, \ldots, u_{id})$ denote the embedding of the $i$-th data point in a $d$-dimensional space. The requirement that the first and second embedding components are equal can be written as:

$$u_{i1} = u_{i2} \quad \forall i \in \{1, 2, \ldots, n\}.$$

Equivalently, this can be expressed as the linear equality constraint:

$$u_{i1} - u_{i2} = 0 \quad \forall i \in \{1, 2, \ldots, n\}.$$

This constraint enforces that, for each data point $i$, the first and second coordinates of the embedding vector $u_i$ are identical.

Within the total objective $\mathcal{L}_{\mathrm{total}}(\boldsymbol{\phi}^{(k)}, \hat{\boldsymbol{\phi}})$ of Eq. 35, an additional penalty term may be introduced to enforce this constraint. The penalty can be written as:

$$\lambda_5 \sum_{i=1}^{n} (u_{i1} - u_{i2})^2,$$

where $\lambda_5$ is a regularization parameter controlling the strength of the constraint. This term encourages the first and second components of each reconstructed embedding point from the optimizer ($\hat{\boldsymbol{\phi}}$) to be equal, while still allowing flexibility depending on the value of $\lambda_5$.

A.5   THE SUPERPOSITION AND THE MONOSEMANTIC REPRESENTATIONS

We model an embedding space as a real vector space $\mathbb{R}^d$, where a hidden activation vector $\mathbf{h} \in \mathbb{R}^d$ represents a combination of underlying semantic features. By the linear representation hypothesis, each interpretable feature corresponds to a fixed direction in $\mathbb{R}^d$ Olah et al. (2020b); Elhage et al. (2022b).

Let $\mathbf{a} \in \mathbb{R}^F$ be a sparse feature activation vector and $W \in \mathbb{R}^{d \times F}$ be a linear transformation such that:

$$\mathbf{h} = W\mathbf{a} = \sum_{i=1}^{F} a_i \mathbf{w}_i,$$

where $\mathbf{w}_i$ denotes the $i$-th column of $W$, corresponding to the direction of the feature $i$.

If $F > d$, the map $W$ cannot be invertible, and thus different combination of characteristics can map to the same embedding. This gives rise to superposition, where multiple semantic features are embedded into shared subspaces or overlapping neuron activations Elhage et al. (2022b).

MONOSEMANTIC REPRESENTATIONS: A representation is called monosemantic when each neuron corresponds to a single interpretable feature Olah et al. (2020b). Mathematically, this corresponds to the case where $W$ is full-rank and aligned with the identity matrix (or a rotation of it):

$$W = I \Rightarrow \mathbf{h} = \mathbf{a}.$$

This implies that each feature $a_i$ is represented by a unique dimension $h_i$, with no overlap. Each neuron responds to a single, isolated concept, akin to "grandmother cells" in neuroscience Quiroga et al. (2005).

POLYSEMANTIC REPRESENTATIONS: In contrast, polysemantic neurons represent multiple, distinct concepts. Formally, if neuron $h_j$ computes:

$$h_j = \sum_{i=1}^{F} W_{j,i} a_i,$$

and two or more $W_{j,i} \neq 0$, then neuron $j$ encodes multiple features simultaneously, exhibiting polysemanticity Elhage et al. (2022b); Bills et al. (2023a).

More generally, a polysemantic embedding may be viewed as a mixture:

$$\mathbf{h} = \sum_{k=1}^{K} \alpha_k \mathbf{c}_k, \quad K > 1,$$

where $\mathbf{c}_k$ are concept vectors and $\alpha_k$ are scalar weights.

This behavior is prevalent in both neural network activations and in biological neurons that exhibit mixed selectivity Rigotti et al. (2013).

Monosemantic representations arise from disentangled bases, where neurons correspond to isolated features. Superposition emerges from dimensionality compression and necessarily leads to polysemantic neurons, each encoding a combination of features. Spare auto-encoder is a way to try to solve the polysemantic neurons—each encoding problem.

## A.6 THE SAE APPROACH AND ARCHITECTURES

Sparse Autoencoder (SAE) architectures have advanced our understanding of how language and vision models represent features Gorton (2024). Neural network behavior is often explained via *computational circuits*—collections of neurons that together compute meaningful functions. Classical circuit analysis has identified key components such as edge detectors Olah et al. (2020a) or word-copying units Olsson et al. (2022). By using features derived from SAEs rather than raw neurons, researchers have improved the interpretability of circuits related to complex behaviors Marks et al. (2024).

Feature discovery can involve visual analysis McDougall (2024), manual inspection Bricken et al. (2023), and even assistance from large language models Bills et al. (2023b). Their causal role is often validated via activation interventions: modifying a feature activation vector $\mathbf{a}$ and observing predictable changes in model output Templeton et al. (2024).

The mathematical formulation situates SAE architectures within the theoretical framework of superposition and semantic disentanglement. By expressing hidden states as sparse linear combinations of interpretable features, SAEs bridge the gap between low-level activations and human-understandable concepts.

LINEAR FORMULATION OF SAEs: Let $\mathbf{x} \in \mathbb{R}^d$ denote a layer's neuron activation vector in a pretrained model. A Sparse Autoencoder learns a sparse feature representation $\mathbf{a} \in \mathbb{R}^F$ such that:

$$\hat{\mathbf{x}} = W\mathbf{a} + \mathbf{b}, \tag{34}$$

where $W \in \mathbb{R}^{d \times F}$ is the decoder (dictionary) matrix and $\mathbf{b} \in \mathbb{R}^d$ is a learned bias term. Each column $W_{\cdot,i}$ represents the direction of feature $i$ in neuron space, and $a_i$ is its activation. This linear mapping enables complex activations to be expressed as combinations of more interpretable features.

If $F > d$, then the feature space is overcomplete, and $W$ cannot be full-rank. This leads to superposition, where multiple features overlap in the same subspace, and individual neurons encode multiple unrelated concepts Elhage et al. (2022b). If $W$ is invertible and aligned to a basis, each neuron corresponds to a single feature. The representation is monosemantic and disentangled Olah et al. (2020b). When $W$ has overlapping columns, neurons can respond to multiple features, yielding polysemantic behavior. That is, for some $j$, $x_j = \sum_i W_{j,i} a_i$ involves multiple nonzero terms Bills et al. (2023a).

VARIANTS OF SAEs: Variants of SAEs like TopK, JumpReLU, and Gated-SAEs offer increasingly precise control over the mapping between low-level activations and human-understandable concepts, enabling fine-grained analysis and intervention.

**TopK-SAEs:** Instead of using a soft sparsityconstraint (e.g., L1 regularization), TopK-SAEs enforce hard sparsity using a top-$K$ activation function:

$$\mathbf{a} = \text{TopK}(W_{\text{enc}}(\mathbf{x} - \mathbf{b}_{\text{dec}})), \tag{35}$$

which retains only the $K$ largest entries of the preactivation and zeros out the rest. This promotes discrete sparsity and avoids complex hyperparameter tuning.

**JumpReLU-SAEs:** JumpReLU replaces ReLU with a thresholded step function:

$$\text{JumpReLU}_\theta(x) = x \cdot H(x - \theta), \tag{36}$$

where $H(\cdot)$ is the Heaviside step function and $\theta$ is a learnable threshold. This allows neurons to activate only above a semantic threshold, aligning with binary behavior observed in some interpretable features. However, the discontinuity makes training difficult due to non-differentiability.

**Gated-SAEs:** Gated-SAEs introduce a gating mechanism that decouples activation magnitude and presence. Let $W_{\text{mag}}$ and $W_{\text{gate}}$ be two encoders. Then the feature activation is computed as:

$$\mathbf{a} = \big(W_{\text{mag}}(\mathbf{x})\big) \odot H(W_{\text{gate}}(\mathbf{x}) - \theta), \tag{37}$$

where $\odot$ denotes elementwise multiplication. This enables better control over when and how strongly a feature activates, making them easier to train than JumpReLU-SAEs Rajamanoharan et al. (2024).

In this study we utilize two different architectures of SAEs the standard SAE and TopK-SAE.

## A.7    ATTRIBUTION FROM SPARSE FEATURE SPACE TO INPUT TOKENS

Let $\mathbf{x}_{\text{input}} \in \mathbb{R}^{d_{\text{input}}}$ denote the input embedding vector (e.g., LLM token embeddings), $\mathbf{x} = f(\mathbf{x}_{\text{input}}) \in \mathbb{R}^d$ the hidden layer activation of the LLM, $\mathbf{a} = \text{Encoder}(\mathbf{x}) \in \mathbb{R}^F$ the SAE sparse feature vector, and $\hat{\mathbf{x}} = W\mathbf{a} + \mathbf{b}$ the reconstructed activation from the SAE decoder. Now suppose we have a sparse attribution vector $\psi_i$ over features $\mathbf{a}$, i.e., $\psi \in \mathbb{R}^F$, where each $\psi_i$ reflects the importance of SAE feature $a_i$. We aim to assign importance $\Phi_k$ to each input token dimension $x_{\text{input},k}$.

ATTRIBUTION FLOW THROUGH THE ENCODER: We propagate the feature attributions backward through the encoder to the input. Using the chain rule:

$$\Phi_k = \sum_{i=1}^{F} \psi_i \cdot \frac{\partial a_i}{\partial x_{\text{input},k}} = \sum_{i=1}^{F} \psi_i \cdot \frac{\partial a_i}{\partial \mathbf{x}} \cdot \frac{\partial \mathbf{x}}{\partial x_{\text{input},k}} \tag{38}$$

where $\frac{\partial a_i}{\partial \mathbf{x}}$ is the encoder Jacobian (SAE layer), and $\frac{\partial \mathbf{x}}{\partial x_{\text{input},k}}$ is the LLM gradient from input token to hidden layer.

This gives us a scalar attribution $\Phi_k \in \mathbb{R}$ for each token/input embedding dimension $k$.

This represents how much each input token contributes to the sparse SAE features that have been identified as important. In this way, we evaluate the contribution of input features based on the monosemantic behavior of the trained network's mechanism. Based on our study thus far, we will apply the six attribution methods previously discussed at two levels: from the SAE feature space to the encoder layer, and from the encoder layer to the input embedding space. This dual-level attribution analysis enables us to investigate how interpretable sparse features relate to model internals and ultimately influence the input-level representations.

To this end, we define a two-step attribution mechanism:

STEP 1: ATTRIBUTION FROM SPARSE FEATURES TO ENCODER LAYER

Let $\boldsymbol{\psi} \in \mathbb{R}^F$ represent the importance scores of sparse features (obtained via attribution methods). We propagate these to the encoder layer as:

$$\boldsymbol{\phi}^{\text{enc}} = W\boldsymbol{\psi} \in \mathbb{R}^d, \tag{39}$$

where $\boldsymbol{\phi}^{\text{enc}}$ quantifies the contribution of each encoder neuron to the important SAE features.

STEP 2: ATTRIBUTION FROM ENCODER LAYER TO INPUT

To assign attribution scores to input dimensions, we propagate $\boldsymbol{\phi}^{\text{enc}}$ to the input embedding via the gradient of the encoder:

$$\boldsymbol{\phi}^{\text{input}} = \left(\frac{\partial \mathbf{x}}{\partial \mathbf{x}_{\text{input}}}\right)^\top \boldsymbol{\phi}^{\text{enc}} \in \mathbb{R}^{d_{\text{input}}}. \tag{40}$$

Alternatively, attribution methods (e.g., Integrated Gradients, SHAP) can directly estimate:

$$\boldsymbol{\phi}^{\text{input}} = \text{AttributionMethod}(f, \mathbf{x}_{\text{input}}, \boldsymbol{\phi}^{\text{enc}})$$

This dual-level attribution analysis allows us to connect semantically meaningful sparse features to the raw input representation space.

## B  SUPPLEMENTARY MATERIAL

### B.0  RELATED WORK

#### B.0.1  ATTRIBUTIONAL INTERPRETABILLITY

Attributional interpretability (AtI), a branch of explainable AI (XAI), focuses on explaining model outputs by tracing predictions back to individual input contributions, often using gradient-based methods Bereska & Gavves (2024). While gradients provide insights into the relationship between inputs and outputs, they can be sensitive to perturbations or discontinuities, posing challenges for reliable interpretation.

AtI encompasses various methods for interpreting complex, nonlinear models, including techniques like Local Interpretable Model-agnostic Explanations (LIME; Ribeiro et al. (2016)) and SHapley Additive exPlanations (SHAP; Lundberg & Lee (2017)). In medical imaging, popular attribution techniques include SHAP, Layer-wise Relevance Propagation (LRP; Bach et al. (2015)), and gradient-based methods like GRAD-CAM (Singh et al. (2020)). These methods aim to enhance trust in models and provide valuable insights into decision-making processes. However, they face limitations. For instance, LRP emphasizes positive preactivations, often yielding less precise explanations, while SHAP is computationally intensive due to the complexity of calculating Shapley values Lundberg & Lee (2017). Adaptations like Monte Carlo methods and stratified sampling (e.g., SVARM) have improved the efficiency and precision of certain techniques Kolpaczki et al. (2024).

### B.0.2 MECHANISTIC INTERPRETABILITY AND SPARSE AUTOENCODER

Mechanistic interpretability (MI), a key area of explainable AI (XAI), focuses on understanding the internal activation patterns of AI models by analyzing their fundamental components, such as features, neurons, layers, and connections. Unlike AtI, MI takes a bottom-up approach, aiming to uncover the causal relationships and precise computations that transform inputs into outputs. This method identifies specific neural circuits driving behavior and provides a reverse-engineering perspective. Insights from fields like physics, neuroscience, and systems biology further guide the development of transparent and value-aligned AI systems.

A core principle of MI is the concept of polysemanticity, where individual neurons encode multiple concepts, contrasted with monosemanticity, where neurons correspond to a single semantic concept. Polysemanticity reduces interpretability, as neurons represent overlapping features. Structures like sparse autoencoders (SAEs) address this by leveraging the superposition hypothesis, which posits that neural networks use high-dimensional spaces to represent more features than the number of neurons, encoding them in nearly orthogonal directions. SAEs decompose embeddings from deep layers, such as MLPs or transformer attention layers, into higher-dimensional monosemantic representations, aligning activation patterns with specific concepts of interest Cunningham et al. (2023); Elhage et al. (2022a).

Sparse Autoencoder architectures have significantly advanced our understanding of feature representations in language and vision models Gorton (2024). Neural network behavior is often interpreted through *computational circuits*—groups of neurons that compute meaningful functions, such as edge detectors Olah et al. (2020a) or word-copying units Olsson et al. (2022). Leveraging SAE-derived features instead of raw neurons has improved the interpretability of circuits associated with complex behaviors Marks et al. (2024). This shift enables clearer mappings between neuron activations and high-level functions, facilitating validation of model behavior Bereska & Gavves (2024). By aligning internal representations with privileged basis directions—distinct semantic vectors within network layers—researchers further enhance monosemanticity and advance the interpretability of deep models.

## B.1 ALZHEIMER DATASET AND PREPROCESSING

### B.1.1 PREPROCESSING

The ADNI data Mueller et al. (2005) was downloaded from the Image & Data Archive (IDA) Neu et al. (2023), run by the Laboratory of Neuro Imaging (LONI) at the USC Mark and Mary Stevens Neuroimaging and Informatics Institute. The download comprised folders including information about participants' enrollment, biospecimen, assessments, medical history, imaging and study information. In this work, only baseline ('bl') visit data was extracted, that is - the first visit the patient underwent when joining each study. The number of unique participant's RIDs (subject's roster ID) was then recorded, and the intersection of such identifiers across the baseline datasets was calculated through an overlap matrix assessing participant coverage by considering datasets symmetrically. The obtained result, underwent precise analysis and filtering. Non-informative and administrative columns (i.e.: SOURCE, update_stamp, SITEID, etc.) were removed across all datasets, to then perform a column-wise completeness check to retain only variables with at least 80% of values present and to balance data availability with feature retention. By prioritizing datasets with the highest number of unique RIDs at baseline, pairwise merging based on shared RIDs was performed (i.e.: inner joins), considering the following files: ADAS, NEUROBAT, FAQ, VITALS, DXSUM. Diagnosis data was sorted chronologically according to EXAMDATE and de-duplicated so as to obtain the first - baseline - diagnosis per subject. Moreover, to ensure robust classification, this was complemented by matching data from adni_diagnosisDXSUM files. For data augmentation purposes, demographics data was obtained from adni_demographic_PTDEMOG and merged according to matching RIDs. Biospecimen and medication data were filtered, cleaned and aggregated by participant - however, due to high sparsity and no adherence of column data to the completeness threshold, such information was not included in the final merge. Similarly, no genetic data was included, due to the lack of relevant biological variables with enough completeness, as remaining columns were primarily collection metadata. The final merged dataset - after excluding administrative columns - comprised 2791 unique participant RIDs with comprehensive neuropsychological, clinical, biospecimen, vital sign,

and demographic data at baseline, with the following diagnosis count: 1207 patients diagnosed with Early Mild Cognitive Impairment (EMCI), 441 with Late Mild Cognitive Impairment (LMCI), and 1143 control subjects. For the binary classification task, EMCI and LMCI subjects were unified into a unique MCI cohort - mimicking AD vs CN classification, while for the three-class task, all three subsets were retained, considering only 440 subjects per class, for balancing purposes. Variables from the obtained merged dataset, were mapped to their descriptions and categorical values, according to the DATADIC_adni123GO dictionary from ADNI Mueller et al. (2005). Text was then generated by iterating through each subject row, replacing column names with their description and appending the corresponding column value for the specific patient. Whereby categorical values were present, they were replaced with their corresponding textual value (i.e.: " 'sex': 0 " - was transformed into "The patient's sex is: male"). Two distinct datasets - one for training and one for testing - were generated from the obtained final datasets, and they were split into training, testing and validation sets.

Another dataset was utilized for further model refinement and finetuning. Specifically, the additional data was extrapolated from MRI files from the Latin American Brain Health Institute (BrainLat) dataset, a multi-site initiative that provides neuroimaging, cognitive, and clinical data across several countries in the Latin American region Prado et al. (2023). The data included cognition, demographic and records information of 780 subjects. A pre-processing pipeline similar to that employed for ADNI, was followed. Namely, after filtering throughout all MRI files, 760 unique and common MRI IDs - representing each subject - were identified. After dropping subjects with a higher proportion of data missing, and columns not fulfilling the completeness threshold, median imputation based on diagnosis group mean was applied for variables with less than 30% of data missing (such as 'Age' and 'years of education' for example) with the goal of obtaining a more complete dataset. After dropping administrative and non-informative columns, the final merged dataset comprised variables deriving from cognitive tests (MOCA - Montreal Cognitive Assessment test and the IFS - INECO Frontal Screening) and participants' demographics. The diagnosis distribution of the obtained dataset was the following: 101 control subjects (CN), 109 diagnosed with Fronto-Temporal Dementia (FTD), and 118 subjects with AD. For the binary classification task, here AD and Fronto-Temporal dementia were unified into a unique cognitively impaired cohort, similar as to what obtained for ADNI, while for the three-class task, the original labels were retained. The same process as for ADNI was followed to obtain textual descriptions of BrainLat patients' data, considering the related dictionary from Prado et al. (2023). Finally, training and testing files where obtained, whereby each class had 50 representative samples each, both for the binary and for the three-class classification. The handling of the final split into training, testing, and validation sets was handled as for ADNI. Throughout the manuscript, the label 'AD' is used for convenience to denote the MCI cohort in ADNI (in both the binary and three-class settings), and the AD+FTD cohort in BrainLat (in the binary setting). This choice is purely notational, as the term functions as a class label rather than a clinical diagnosis, and the emphasis is on the model's ability to discriminate between the defined classes.

### B.1.2 Demographic Comparison of Alzheimer's Cohorts and Matched Controls

To ensure demographic comparability and reduce confounding in downstream analyses, we examined age and sex distributions across each Alzheimer's disease (AD) cohort and control groups.

Considering the cohorts for the binary classification from ADNI Mueller et al. (2005), AD subjects ($n = 1207$) and the control group ($n = 1143$), it is worth noting that both groups consider subjects who were born between a range that goes from the 1930s to the 1960s with comparable distributions. The AD group exhibits sharper age peaks, (Figure 36(a)) , while the control group shows a more uniform spread. A similar pattern is evident from the three-class classification cohorts (Figure 36(c)), whereby patients diagnosed with LMCI and MCI tend to be demonstrate higher density at certain points, whereas healthy subjects' birth year distribution tends to be flatter.

The gender distribution is uniform across groups, both in binary and three-class classification (Figures 36(b) and 1(d)), with a slight predominance of female participants in AD groups, but overall disparity suggests minimal risk of demographic bias.

Regarding the BrainLat dataset Prado et al. (2023), similar patterns are evident. Control subjects are, on average, younger than subjects diagnosed with AD by 4 years, although the distribution

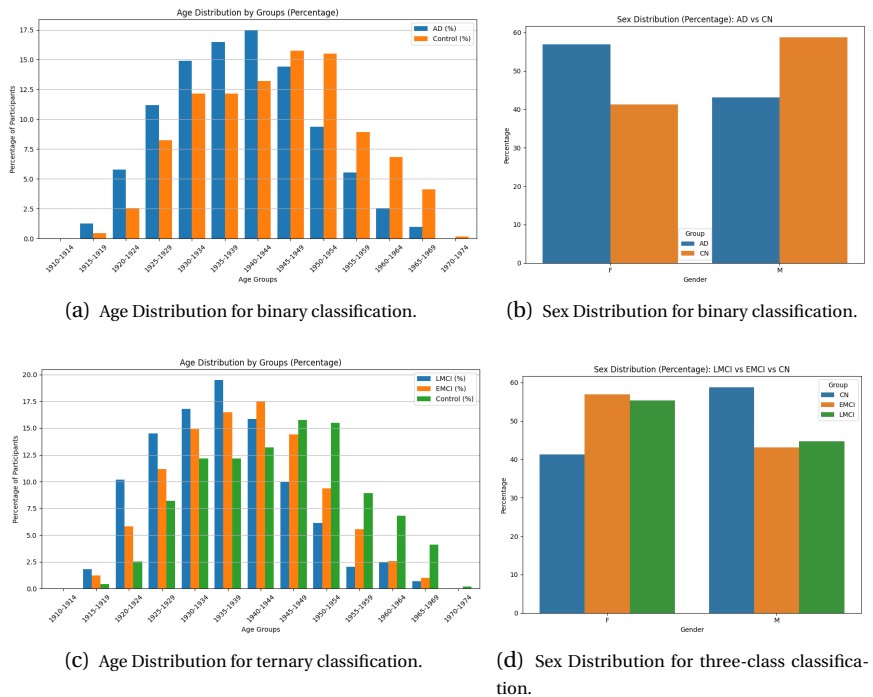

(a) Age Distribution for binary classification.

(b) Sex Distribution for binary classification.

(c) Age Distribution for ternary classification.

(d) Sex Distribution for three-class classification.

Figure 1: Demographic distributions (age and gender) for Alzheimer's cohorts and control groups for both binary and ternary classification tasks. The top row refers to the binary task, while the bottom row analyzes cohorts for the ternary classification task.

for AD tends to be more coherently spread than the one for CN (AD cohort mean age: 71, with a standard deviation of 8.7, CN cohort mean age: 67, with standard deviation of 8.5). In the cohorts obtained for the three-class classification task, the age difference remains the same - as AD subjects tend to be the oldest, followed by those belonging to the FTD cohort and CN cohort respectively. Age variability in this case, becomes more comparable between the different diagnoses. Similarly to what was found for ADNI, gender-wise, the data distribution tends to be more skewed toward female participants, both in the AD and in the CN cohorts. The same is found for the subsets obtained for the three-class classification task, whereby female patients diagnosed with AD and FTD represent a higher number than male ones.

### B.1.3 PHENOTYPIC AND LIFESTYLE PROFILING

To characterize the ADNI cohorts beyond age and sex, we analyzed phenotypic and lifestyle variables spanning physical health (e.g., systolic and diastolic blood pressure, respiratory and pulse rate, height, weight, body temperature, dominant hand) and behavioral and lifestyle factors (e.g., living situation, marital status, primary language). These features were compared across all four groups to identify significant inter-group differences Mueller et al. (2005).

In the comparison between AD and CN cohorts for the binary classification, a significant difference was found in subjects' pulse rate ($p < 0.05$) based on independent samples t-test - consistent with the nervous system dysfunction that Alzheimer's involves. Instead, no significance was found for systolic and diastolic blood pressure, respiratory rate, body temperature and weight. In terms of behavioral and lifestyle factors, a significant difference in marital status - based on Fisher's exact test - was observed between the two groups. Although most of subjects in the AD and CN groups were married, widowed individuals made up a larger proportion than divorced individuals in the AD group, while the opposite was true for CN subjects. Moreover, the CN group had a higher percentage of individuals who had never been married. Subjects also differed for living situation (Fisher's exact test). Most subjects diagnosed with AD, lived in a house and smaller proportions lived in - respectively - a condo, an apartment, and a mobile home, with the lowest percentages

residing in a retirement community and in an assisted living facility. Although CN subjects also predominantly lived in a house, they were more likely than AD subjects to live in an apartment or a condo, followed by a mobile home, an assisted living facility and lastly, a retirement community.

### B.1.4 ASSESSING COMPATIBILITY BETWEEN IID AND OOD COHORTS

The selection of ADNI (IID) and BrainLat (OOD) cohorts was motivated by their demographic comparability and complementary clinical profiles. As described in Section B.1.2, both datasets show overlapping age and sex distributions, with balanced ratios and only minor female predominance. These similarities minimize confounding, ensuring that performance differences reflect domain shifts rather than demographic bias.

Phenotypic and lifestyle profiling (Section B.1.3) revealed moderate inter-group differences, such as in pulse rate and marital status, consistent with disease-specific traits. ADNI primarily represents the Alzheimer's continuum (CN, MCI, LMCI), whereas BrainLat includes FTD, AD, and controls. Despite differing diagnostic labels, these groups share clinical overlap: FTD often exhibits MCI-like cognitive decline, and LMCI represents a prodromal AD stage Petersen et al. (1999); Jack et al. (2018); Gorno-Tempini et al. (2011).

This overlap establishes a natural testbed for generalization, challenging models trained on IID data to transfer to OOD settings with related but non-identical diagnoses. The IID/OOD pairing thus provides a rigorous, clinically meaningful framework to evaluate the adaptability and robustness of LLM-based diagnostic systems.

### B.1.5 MODALITIES SUBGROUP EXTRACTIONS

Table 1: Variables with character counts, generation order, and estimated tokens (tokens $\approx \lceil \text{chars}/4 \rceil$).

| Variable | Description | Chars | Order | Tokens (est.) |
|---|---|---|---|---|
| PTGENDER | The participant's sex is | 30 | 1 | 8 |
| PTDOB | Their Date of Birth is | 31 | 2 | 8 |
| PTDOBYY | Their Year of Birth is | 28 | 3 | 7 |
| PTHAND | Their Handedness is | 26 | 4 | 7 |
| PTMARRY | Their Marital status at baseline is | 44 | 5 | 11 |
| PTEDUCAT | Their education in years is | 31 | 6 | 8 |
| PTNOTRT | Participant Retired? | 25 | 7 | 7 |
| PTHOME | Type of Participant residence | 54 | 8 | 14 |
| PTTLANG | Language to be used for testing the Participant | 56 | 9 | 14 |
| PTPLANG | Participant's Primary Language | 39 | 10 | 10 |
| PTETHCAT | The participant's Ethnicity is | 54 | 11 | 14 |
| PTRACCAT | Trail Making Test: Race | 28 | 12 | 7 |
| PTSOURCE | Information Source | 37 | 13 | 10 |
| VSWEIGHT | The participant's weight is | 32 | 14 | 8 |
| VSWTUNIT | The weight was measured in | 34 | 15 | 9 |
| VSBPSYS | The participant's Systolic - mmHg | 40 | 16 | 10 |
| VSBPDIA | The participant's Diastolic - mmHg | 40 | 17 | 10 |
| VSPULSE | The participant's Seated Pulse Rate (per minute) is | 56 | 18 | 14 |
| VSRESP | The participant's Respirations (per minute) are | 51 | 19 | 13 |
| VSTEMP | The participant's Temperature is | 37 | 20 | 10 |
| VSTMPSRC | The Temperature Source was | 32 | 21 | 8 |
| VSTMPUNT | The Temperature Units were | 38 | 22 | 10 |
| DXDEP | Depressive symptoms present? | 32 | 23 | 8 |
| CLOCKCIRC | On the Clock Drawing Test the partecipant answered the following questions in this way: Approximately circular face | 126 | 24 | 32 |
| CLOCKSYM | Symmetry of number placement | 39 | 25 | 10 |
| CLOCKNUM | Correctness of numbers | 31 | 26 | 8 |

*Continued on next page*

| Variable | Description | Chars | Order | Tokens (est.) |
|---|---|---|---|---|
| CLOCKHAND | Presence of the two hands | 34 | 27 | 9 |
| CLOCKTIME | Presence of the two hands, set to ten after eleven | 59 | 28 | 15 |
| CLOCKSCOR | Clock Drawing Test: Total Score | 36 | 29 | 9 |
| COPYCIRC | On the Clock copying task the participant scored as follows: Approximately circular face | 95 | 30 | 24 |
| COPYSYM | Symmetry of number placement | 39 | 31 | 10 |
| COPYNUM | Correctness of numbers | 31 | 32 | 8 |
| COPYHAND | Presence of the two hands | 34 | 33 | 9 |
| COPYTIME | Presence of the two hands, set to ten after eleven | 59 | 34 | 15 |
| COPYSCOR | Clock copying task: Total Score | 36 | 35 | 9 |
| AVTOT1 | On the Auditory Verbal Learning Test the participant scored as follows in each trial: Trial 1 Total | 104 | 36 | 26 |
| AVERR1 | Total Intrusions | 19 | 37 | 5 |
| AVTOT2 | Trial 2 Total | 16 | 38 | 4 |
| AVERR2 | Total Intrusions | 19 | 39 | 5 |
| AVTOT3 | Trial 3 Total | 16 | 40 | 4 |
| AVERR3 | Total Intrusions | 19 | 41 | 5 |
| AVTOT4 | Trial 4 Total | 16 | 42 | 4 |
| AVERR4 | Total Intrusions | 19 | 43 | 5 |
| AVTOT5 | Trial 5 Total | 16 | 44 | 4 |
| AVERR5 | Total Intrusions | 19 | 45 | 5 |
| AVTOT6 | Trial 6 Total | 16 | 46 | 4 |
| AVERR6 | Total Intrusions | 19 | 47 | 5 |
| AVTOTB | List B Total | 15 | 48 | 4 |
| AVERRB | Total Intrusions | 19 | 49 | 5 |
| CATANIMSC | On the Category Fluency Test Animals the scores were: - Total Correct | 73 | 50 | 19 |
| CATANPERS | Perseverations | 17 | 51 | 5 |
| CATANINTR | Intrusions | 13 | 52 | 4 |
| TRAASCOR | Part A - Time to Complete | 29 | 53 | 8 |
| TRAAERRCOM | Errors of Commission | 23 | 54 | 6 |
| TRAAERROM | Errors of Omission | 21 | 55 | 6 |
| TRABSCOR | Part B - Time to complete | 30 | 56 | 8 |
| TRABERRCOM | Errors of Commission | 23 | 57 | 6 |
| TRABERROM | Errors of Omission | 21 | 58 | 6 |
| AVDEL30MIN | On the Auditory Verbal Learning Test the participant scored as follows: 30 Minute Delay Total | 96 | 59 | 24 |
| AVDELERR1 | Total Intrusions | 19 | 60 | 5 |
| AVDELTOT | Recognition Score | 20 | 61 | 5 |
| AVDELERR2 | Total Intrusions | 19 | 62 | 5 |
| ANARTERR | American National Adult Reading Test: ANART Total Score (Total # of errors) | 81 | 63 | 21 |
| FAQFINAN | For the Functional Activities Questionnaire the participant scored as follows for each question: Writing checks, paying bills, or balancing checkbook. | 151 | 64 | 38 |
| FAQFORM | Assembling tax records, business affairs, or other papers. | 58 | 65 | 15 |
| FAQSHOP | Shopping alone for clothes, household necessities, or groceries. | 64 | 66 | 16 |
| FAQGAME | Playing a game of skill such as bridge or chess, working on a hobby. | 68 | 67 | 17 |
| FAQBEVG | Heating water, making a cup of coffee, turing off the stove. | 60 | 68 | 15 |
| FAQMEAL | Preparing a balanced meal. | 26 | 69 | 7 |
| FAQEVENT | Keeping track of current events. | 32 | 70 | 8 |
| FAQTV | Paying attention to and understanding a TV program, book, or magazine. | 70 | 71 | 18 |
| FAQREM | Remembering appointments, family occasions, holidays, medications. | 66 | 72 | 17 |

| Variable | Description | Chars | Order | Tokens (est.) |
|---|---|---|---|---|
| FAQTRAVL | Trail Making Test for FAQ score: Traveling out of the neighborhood, driving, or arranging to take public transportation. | 121 | 73 | 31 |
| FAQTOTAL | Total Score for FAQ is | 26 | 74 | 7 |

Based on Table 1, we extracted nine subgroups as follows: Demographics, Vital Signs, Clock Drawing Test, Clock Copying Test, Auditory Verbal Learning Test (version 1), Category Fluency – Animal Test, Auditory Verbal Learning Test (version 2), American National Adult Reading Test, and Functional Activities Questionnaire.

### B.1.6 DATASETS CLAIMS

Data used in the preparation of this article was obtained from the Alzheimer's Disease Neuroimaging Initiative (ADNI) database (adni.loni.usc.edu) on August 8th 2025 (version: "08Aug2025") and it included all ADNI phases. ADNI was launched in 2003 as a public-private partnership, led by Principal Investigator Michael W. Weiner, MD. It aimed at testing whether cognitive, imaging, genetic, clinical, neuropsychological assessment and other biological markers, can be combined to measure the progression of mild cognitive impairment (MCI) and early Alzheimer's disease (AD). The goals also include the validation of biomarkers for clinical trials, and the provision of data concerning the diagnosis and progression of Alzheimer's disease to the scientific community. For up-to-date information, see adni.loni.usc.edu.

### B.2 SUMMARY OF TRAINING OUTCOMES FOR LLM ENCODERS ON IID AND OOD DATASETS

We systematically compared the performance of different *fine-tuned encoder models* (BERT, RoBERTa, DistilBERT, ALBERT, BioBERT, ModernBERT) on ADNI (in-domain, IID) and evaluated cross-dataset generalization to BRAINLAT (out-of-domain, OOD). On ADNI, ModernBERT is the strongest encoder across all metrics: *Binary*—Acc: 0.7237, F1: 0.7589, ROC-AUC: 0.8395, AUC-PR: 0.8641. *Three-class*— Acc: 0.6505, F1: 0.6880, ROC-AUC: 0.7867, AUC-PR: 0.7848. BioBERT and RoBERTa are the most competitive baselines but remain below ModernBERT. In the zero-shot transfer from ADNI to BRAINLAT and the binary classification task, ModernBERT achieved modest performance with an average accuracy of approximately 0.55. In a representative run, the model reached 0.53 accuracy, 0.52 precision, 0.70 recall, an F1 score of 0.60, and both *ROC-AUC* and *AUC-PR* near 0.58. These results highlight a conservative decision threshold and the difficulty of domain transfer without adaptation. Introducing few-shot supervision improved performance moderately. In the $K$-shot regime, accuracy increased by up to 0.10 compared to zero-shot, reaching approximately 0.62 at $K = 10$, with parallel gains in F1. The ROC-AUC and AUC-PR metrics remained high and stable, suggesting that limited supervision can partially mitigate domain shift but does not fully bridge the gap. LoRA-based parameter-efficient adaptation produced results comparable to few-shot training, offering efficiency in training without substantial additional gains in predictive performance. By contrast, full fine-tuning of all pretrained weights on BRAINLAT yielded the strongest improvements, with accuracy rising to 0.84 and consistent gains across F1, ROC-AUC, and AUC-PR. These results demonstrate that full supervised adaptation remains the most effective approach to address domain shift when sufficient labeled data are available. In the three-class BRAINLAT setting, zero-shot transfer from ADNI yielded limited generalization (*Accuracy*=0.40, *F1*=0.41, *ROC-AUC*=0.44), reflecting the challenge of domain and class shifts. Few-shot adaptation ($K = 10$) moderately improved performance (*Accuracy*=0.49, *F1*=0.48), while *LoRA*-based fine-tuning achieved comparable results (*Accuracy*=0.50, *F1*=0.48). Full fine-tuning produced the strongest gains, reaching *Accuracy*=0.69, *F1*=0.73, and *ROC-AUC*=0.81. These findings confirm that, although limited supervision aids adaptation, full parameter optimization is essential for robust multi-class generalization across cohorts. However, this setting is outside the scope of this work: we focus on explanation performance under OOD conditions without training on the OOD cohort (i.e., without full fine-tuning).

Therefore, for all downstream analyses we *stick with ModernBERT*: in the IID setting we use *ModernBERT* fine-tuned on ADNI (best overall on in-domain tasks), and in the OOD setting we use *ModernBERT* in a zero-shot configuration on BRAINLAT (best overall under out-of-domain

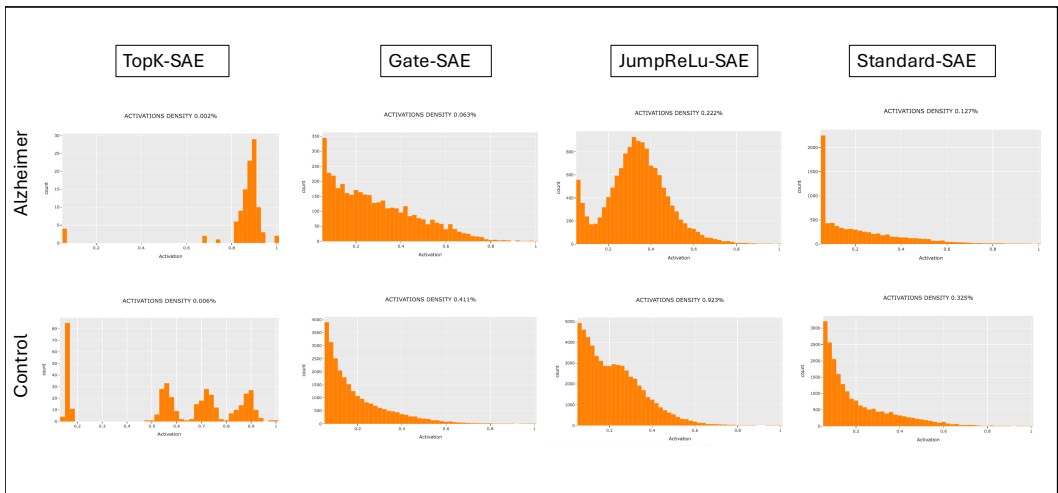

Figure 2: Latent space projections from four SAE variants (TopK-SAE, Gate-SAE, JumpReLU-SAE, Standard-SAE) applied to Alzheimer's and Control groups. TopK-SAE shows the clearest group separation, highlighting its superior ability to extract interpretable, clinically relevant features.

conditions). All subsequent explainability analyses were conducted using the final (22nd) layer of *ModernBERT*.

### B.3 HYPERPARAMETER TUNING FOR THE RECONSTRUCTION OPTIMIZER AND SAE MODELS.

A thorough hyperparameter tuning process was conducted for each simulation (Figures 3, 4, 5). The explanation optimizer was trained with learning rates of $2e^{-2}$, $2e^{-3}$, $2e^{-4}$, and $2e^{-5}$, with the best performance observed at $2e^{-4}$. Various combinations of the weighting parameters $\lambda_1, \lambda_2, \lambda_3, \lambda_4$ were tested—for example, (0.3, 0.2, 0.25, 0.25)—with the optimal configuration found to be (0.1, 0.3, 0.1, 0.5). For the UMAP constraints, subgroup levels were evaluated across several scales: no UMAP, every 4× batch size, 10× batch size, and full cohort level. The best performance was achieved at the 4× batch size level. Regarding the SAE (Sparse Autoencoder), different model variants were evaluated, including *Standard*, *TopK*, *JumpReLU*, and *GATE*, as described in the Methods section. Among these, the *TopK* variant achieved the best results. Feature space depths of 16×, 32×, and 64× were tested, with 32× providing the best trade-off between sparseness and reconstruction performance. The final simulation and training settings included the Adam optimizer (Kingma & Ba, 2014) with a learning rate of $2e^{-4}$, a batch size of 64, and 200 total training steps, using a 50/50 train-validation split. The learning rate schedule followed a fixed-step approach with a step size of 150 and a decay factor (gamma) of 0.95. For the SAE training, we used 6,000 training steps, 200,000 training tokens, a learning rate of $5e^{-5}$, and a model dimension of 768, consistent with the 22-layer *Modern-BERT* architecture. The context size was 512, with warm-up steps of 1,000, learning rate decay steps of 1,200, and L1 warm-up steps of 300. Finally, explanation metrics such as ROS, RIS, and sparseness were computed using default configurations from the `quantus` Python package (Hedström et al., 2023). Figure 3 presents a comparative visualization of activation patterns projections generated by different Sparse Autoencoder (SAE) variants—TopK-SAE, Gate-SAE, JumpReLU-SAE, and Standard-SAE—applied to two subject groups: Alzheimer's and Control. While the specific axes and metrics are not labeled, the separation between the two groups provides insight into the effectiveness of each SAE in producing disentangled, semantically meaningful representations. Among the models, the TopK-SAE exhibits the clearest separation between the Alzheimer's and Control cohorts, suggesting superior performance in capturing clinically relevant patterns. This visual evidence supports the paper's central claim that monosemantic representations enhance interpretability and robustness in clinical applications of LLMs.

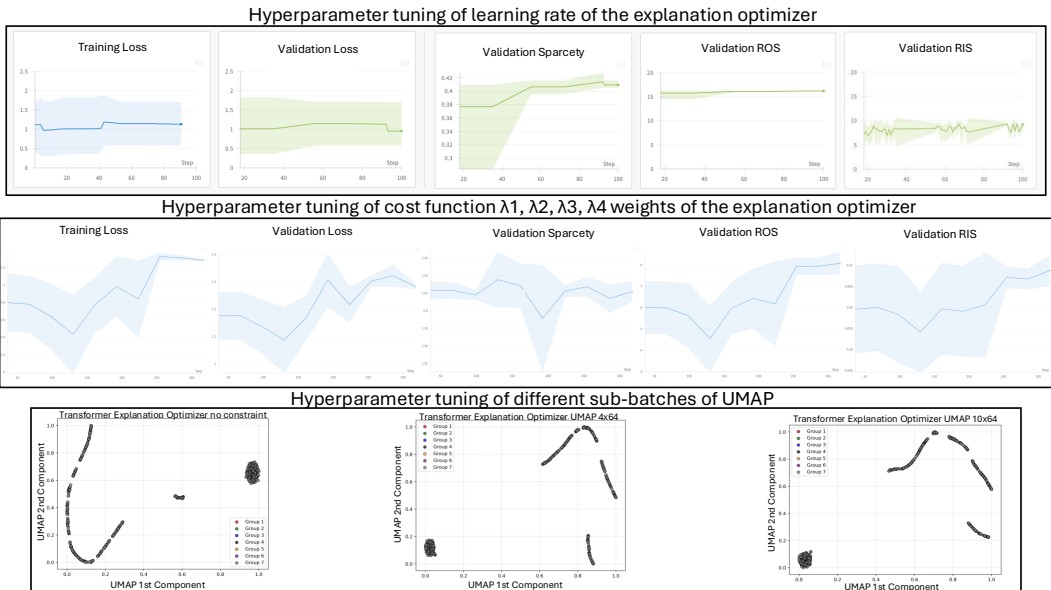

Figure 3: **Hyperparameter tuning of the explanation optimizer and UMAP settings.** Top row: impact of learning rate on training loss, validation loss, sparseness, ROS, and RIS metrics. Middle row: sensitivity analysis of the explanation cost weights $\lambda_1$, $\lambda_2$, $\lambda_3$, and $\lambda_4$, showing trade-offs between attribution sparseness and robustness. Bottom row: UMAP projections of token-level attribution spaces under different sub-batch configurations, revealing how UMAP resolution influences the geometric structure of explanations.

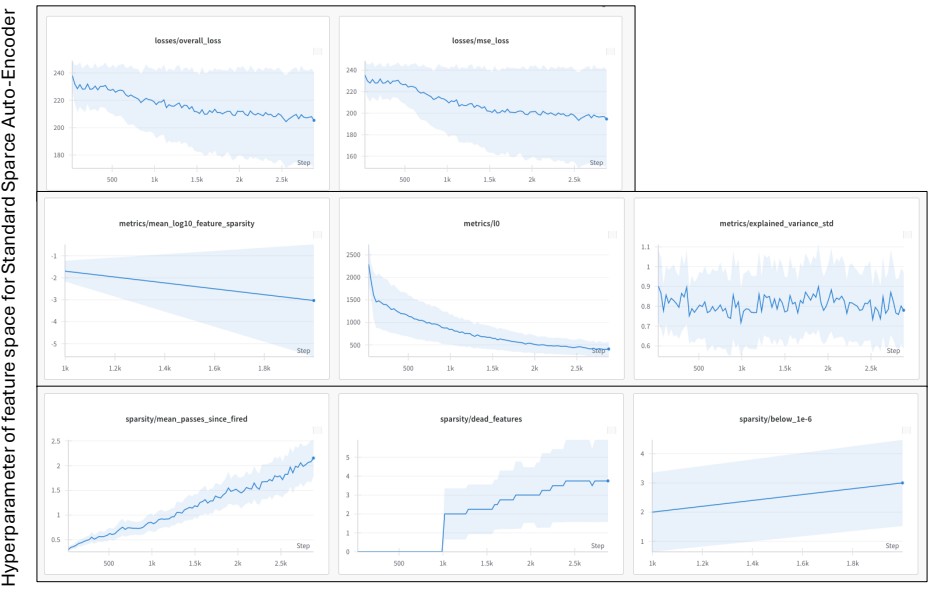

Figure 4: Hyperparameter tuning of the feature space for the Standard Sparse Autoencoder (SAE). The plots track training dynamics and sparseness characteristics across training steps. Top row: loss trends for overall and reconstruction loss. Middle row: log-sparsity metric, Kullback–Leibler divergence (KL), and explained variance standard deviation. Bottom row: progression of sparsity across mean-poisson stem-freed features, fixed features, and a threshold-based view (1e-6). These results guide optimal SAE configurations for producing monosemantic feature representations.

Performance of Variant of Sparce Auto-Encoder

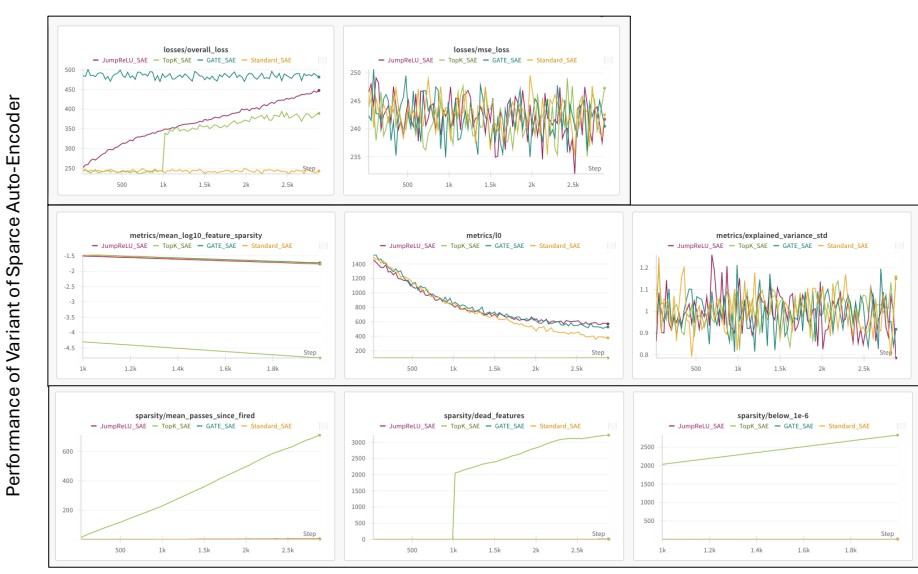

Figure 5: Performance comparison of different variants of the Sparse Autoencoder (SAE). Top row: overall and reconstruction loss across training steps for JumpInit-SAE, Top-$k$ SAE, Gated-SAE, and Standard-SAE. Middle row: log-sparsity metric, KL divergence, and explained variance standard deviation, showing divergence in regularization behavior. Bottom row: sparsity progression for mean-poisson stem-freed features, fixed feature count, and a thresholded view (1e-6). JumpInit-SAE shows early convergence in sparsity, while Gated-SAE maintains tighter control over variance. These results highlight trade-offs between sparsity enforcement mechanisms and attributional stability.

### B.4 EXTRA RESULTS

Figure 6 compares training dynamics and interpretability metrics on ADNI for binary (top row) and three-class (bottom row) classification. Each subfigure shows three variants of the Explanation Optimizer: *without* SAE (black), *with* SAE (brown), and *with* SAE + linear UMAP constraint (grey). Each row in the figure presents the model's behavior over training steps across six key metrics: train loss, validation loss, UMAP reconstruction error (MSE), Relative Output Stability (ROS), Relative Input Stability (RIS), and sparsity, . Across both tasks, all training ROS and RIS values for the SAE-based variants (brown/grey) are consistently lower than the no-SAE baseline (black), indicating improved attributional robustness. While sparseness does decrease when introducing SAE, the reduction is modest; adding the linear UMAP constraint (grey) achieves a better balance, maintaining relatively high sparsity while keeping low RIS/ROS. Finally, the training and validation curves track closely and remain smooth for the SAE variants, providing no evidence of overfitting: validation loss follows training loss without widening gaps in either the binary or three-class setting. Overall, the SAE-enhanced Explanation Optimizer demonstrates significantly improved performance across all interpretability metrics, supporting the hypothesis that enforcing monosemantic representations improves explanation clarity and reliability—especially in high-stakes clinical contexts like Alzheimer's disease classification.

Across IID (ADNI) and OOD (BrainLat) settings, and for both binary (Alzheimer vs. Control) and three-class (Control/LMCI/MCI) tasks, the tables reveal a consistent stability–sparsity frontier driven by the proposed explanation optimizers and the presence of a monosemantic bottleneck (SAE). In the binary IID case (Table 2), SAE substantially improves stability for explainers that learn features—most notably Layer Conductance and especially TEO—with large drops in RIS/ROS for both Alzheimer and Control, while Activation with SAE increases RIS/ROS and is therefore less robust. In the binary OOD case (Table 3), these patterns persist and even strengthen: TEO with SAE bottleneck attains the lowest RIS/ROS overall, demonstrating strong cross-dataset stability, whereas TEO–UMAP recovers higher sparseness (>0.40) at the cost of higher RIS/ROS than TEO

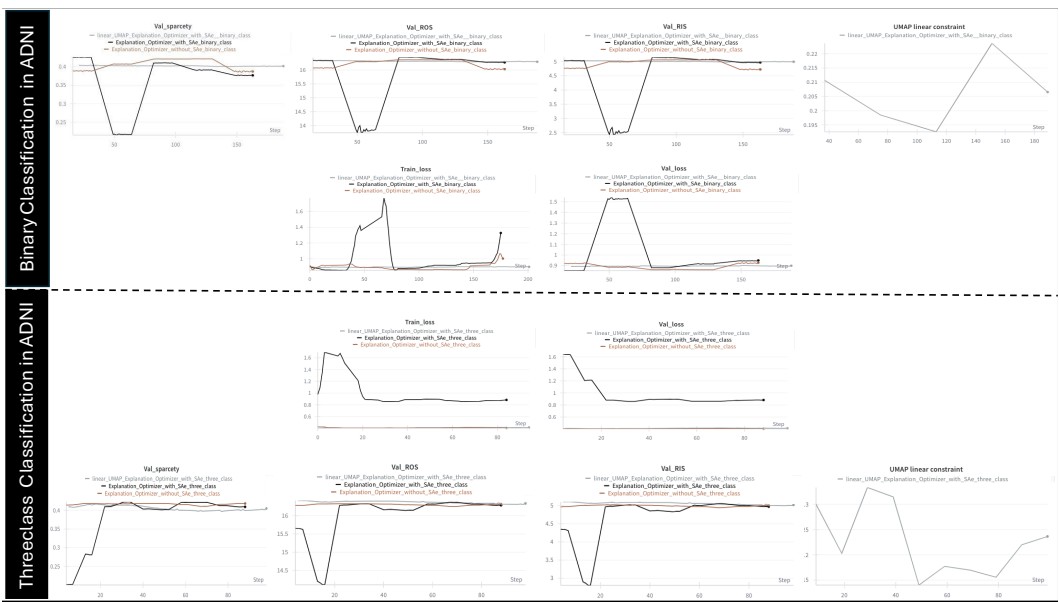

Figure 6: Training and interpretability dynamics on ADNI for binary (top) and three-class (bottom) classification. *Each subfigure includes three variants*: Explanation Optimizer *without* SAE (black), *with* SAE (brown), and *with* SAE + linear UMAP constraint (grey. For each variant we plot train loss, validation loss, UMAP reconstruction MSE (linear UMAP constrain), Relative Output Stability (ROS), Relative Input Stability (RIS), and sparcety, cohorts shown separately. SAE reduces volatility and lowers UMAP MSE and RIS/ROS versus the no-SAE baseline; adding a linear UMAP constraint on top of SAE further improves manifold structure and attribution stability, at a minor cost in sparsity.

with SAE, offering a tunable sparsity–stability trade-off. In the three-class IID setting (Table 4), Feature Ablation is the sparsity leader across Control/LMCI/MCI ( 0.52–0.53) with moderate, steady RIS/ROS; Layer Conductance with SAE markedly reduces RIS/ROS for LMCI/MCI; and TEO with SAE again delivers the most stable attributions across all classes (lowest RIS/ROS), albeit with reduced sparseness. The same rank ordering holds OOD (Table 5): TEO with SAE remains the stability winner for Control/LMCI/MCI, TEO–UMAP trades some stability for additional sparsity, and Feature Ablation remains the simplest high-sparsity baseline. Throughout all tables, gradient-formulaic methods (Grad-SHAP, Guided Backprop, Integrated Gradients) show near-invariant RIS/ROS ( 5.6/ 16.93) regardless of SAE, class, or domain, indicating that SAE chiefly benefits learned-attribution methods. Collectively, Tables 2, 3, 4, and 5 support three conclusions: (i) adding an SAE bottleneck reliably lowers RIS/ROS where explanations are learned (Layer Conductance, TEO), (ii) TEO with SAE is the default when stability is paramount, while TEO–UMAP is preferred when higher sparsity is required, and (iii) the class-wise and IID to OOD behaviors are consistent, underscoring the robustness of monosemantic representations for clinical explanation.

## B.5    STATISTICAL ANALYSIS

We conducted both parametric and non-parametric statistical tests on the binary and three-class classification performance of all classes and tasks in the ADNI cohort to assess the significance of differences introduced by the monosemantic bottleneck (SAE) in traditional attribution techniques, focusing on the metrics of Sparseness, RIS, and ROS.

For the binary classification task, in both the Control and Alzheimer's groups, paired testing demonstrated that SAE produced robust and statistically significant reductions in attribution-based measures and Complexity, while effects on RIS were smaller but still reliable, and changes in ROS were modest and often non-significant after correction. In the Control group, Complexity decreased from $0.3377 \pm 0.0017$ (no-SAE) to $0.3140 \pm 0.0010$ (SAE), yielding $t(29) = 64.0$, $p = 1.5 \times 10^{-47}$ (FDR $q < 10^{-46}$), and RIS declined from $9.313 \pm 0.143$ to $9.175 \pm 0.109$, $t(29) = 4.22$,

Table 2: Evaluation scores with and without SAe. Values are mean ± std. Classes: Alzheimer and Control. Metrics: Sparseness (higher is better), RIS (lower is better), ROS (lower is better). Column order: Alzheimer (No SAE), Alzheimer (SAE), Control (No SAE), Control (SAE). All evaluation metrics were calculated on 200 randomly selected patients from each class (binary-class classification task) in the **ADNI** testing cohort (IID). Abreviations, DEO: Diffusion Explanation Optimizer, TEO: Transformer Explanation Optimizer, TEO-UMAP: Transformer Explanation Optimizer with UMAP constraint.

| Method | Metric | Alzheimer | | Control | |
|---|---|---|---|---|---|
| | | No SAE | SAE | No SAE | SAE |
| Activation | Sparseness | 0.316364045 ± 0.007573187 | 0.296615553 ± 0.007063087 | 0.256150148 ± 0.01759176 | 0.251987915 ± 0.004701258 |
| | RIS | 14.30227 ± 0.368612837 | 21.3084024 ± 0.311021506 | 14.23653893 ± 0.338127875 | 19.32752421 ± 0.932339244 |
| | ROS | 25.54851914 ± 0.482826721 | 32.61738364 ± 0.307928076 | 25.54874248 ± 0.328628877 | 30.6394217 ± 0.933402318 |
| Layer Condact | Sparseness | 0.396588773 ± 0.026146226 | 0.391508745 ± 0.007549659 | 0.374476778 ± 0.007092037 | 0.247974319 ± 0.007908586 |
| | RIS | 12.39850671 ± 2.640648847 | 5.628509596 ± 0.023609264 | 5.650216593 ± 0.0391015 | 5.614140617 ± 0.018407327 |
| | ROS | 23.146556 ± 1.686524073 | 16.94708243 ± 0.010343602 | 16.9614573 ± 0.031757755 | 16.93014311 ± 0.005191254 |
| Feature Ablation | Sparseness | **0.523581491 ± 0.009806381** | **0.523492234 ± 0.010441342** | **0.525551296 ± 0.011012696** | **0.526520447 ± 0.008837301** |
| | RIS | 23.15233791 ± 0.819793812 | 23.56094785 ± 0.103321598 | 22.56110559 ± 0.288403561 | 23.62208862 ± 0.093292383 |
| | ROS | 33.90884482 ± 0.161311922 | 34.92976979 ± 0.074540131 | 33.90759033 ± 0.374703897 | 34.9726397 ± 0.10327352 |
| Gradinet-SHAP | Sparseness | 0.319169309 ± 0.004303288 | 0.082047681 ± 0.015464469 | 0.433346255 ± 0.003004736 | 0.133912362 ± 0.009943283 |
| | RIS | 5.623104979 ± 0.023650419 | 5.621792106 ± 0.022658996 | 5.632513362 ± 0.023548461 | 5.619618915 ± 0.019004514 |
| | ROS | 16.93566464 ± 0.001800535 | 16.93449562 ± 1.54604E-05 | 16.94606355 ± 0.002246403 | 16.93475655 ± 5.99931E-06 |
| Gradient Activation | Sparseness | 0.327713636 ± 0.03837053 | 0.203454702 ± 0.011686877 | 0.249969957 ± 0.023039465 | 0.16678783 ± 0.00722766 |
| | RIS | 5.614858128 ± 0.019338754 | 5.625220574 ± 0.021340754 | 5.616992957 ± 0.021780592 | 5.617270064 ± 0.022114697 |
| | ROS | 16.93028085 ± 0.003427604 | 16.93433453 ± 6.78428E-05 | 16.934673 ± 1.43645E-14 | 16.93473306 ± 4.02532E-05 |
| Integrated-Gradient | Sparseness | 0.298289818 ± 0.008006549 | 0.121161362 ± 0.005775061 | 0.430359021 ± 0.006572262 | 0.064427234 ± 0.005909053 |
| | RIS | 5.620585979 ± 0.018022403 | 5.622360787 ± 0.017750318 | 5.627793468 ± 0.018989203 | 5.621391149 ± 0.016872007 |
| | ROS | 16.93257772 ± 0.001464829 | 16.93453232 ± 1.20129E-05 | 16.94336632 ± 0.002408932 | 16.93456653 ± 8.33237E-06 |
| DEO | Sparseness | 0.338261111 ± 0.003260844 | 0.337375 ± 0.00290587 | 0.337742857 ± 0.001742551 | 0.314044444 ± 0.001036523 |
| | RIS | 9.283888889 ± 0.080010212 | 9.279 ± 0.064555158 | 9.313125 ± 0.142722049 | 9.175 ± 0.108803655 |
| | ROS | 20.63421053 ± 0.086558637 | 20.615 ± 0.088049029 | 20.61588235 ± 0.202578961 | 20.515 ± 0.129878486 |
| TEO | Sparseness | 0.421975723 ± 0.000305212 | 0.267210213 ± 0.001025675 | 0.419939638 ± 0.00048088 | 0.268167213 ± 0.000728522 |
| | RIS | **5.051961362 ± 0.019221728** | **1.622662574 ± 0.17080061** | **5.06881834 ± 0.01838977** | **0.996401319 ± 0.263922792** |
| | ROS | **16.35285123 ± 0.00563874** | **12.92504253 ± 0.170261034** | **16.37765691 ± 0.001096906** | **12.29830928 ± 0.261259725** |
| TEO-UMAP | Sparseness | *N/A* | 0.39891406 ± 0.000414208 | *N/A* | 0.40566988 ± 0.00031341 |
| | RIS | *N/A* | 5.439373370 ± 0.033211570 | *N/A* | 5.47087230 ± 0.17460810 |
| | ROS | *N/A* | 16.3036705 ± 0.00333634 | *N/A* | 16.21021807 ± 0.0078926 |

$p = 9.5 \times 10^{-5}$ (FDR $q = 1.9 \times 10^{-4}$), both clearly rejecting the null hypothesis, whereas ROS decreased slightly from $20.616 \pm 0.203$ to $20.515 \pm 0.131$, $t(29) = 2.30$, $p = 0.026$ (FDR $q = 0.026$), a marginal result that did not withstand correction. Attribution metrics showed the largest SAE effects: Grad-SHAP dropped from $0.4333 \pm 0.0030$ to $0.1339 \pm 0.0099$ ($p < 10^{-50}$), Guided Backprop from $0.2500 \pm 0.0230$ to $0.1668 \pm 0.0072$ ($p < 10^{-19}$), Integrated Gradients from $0.4304 \pm 0.0066$ to $0.0644 \pm 0.0059$ ($p < 10^{-80}$), and Optimizer from $0.4199 \pm 0.0005$ to $0.2682 \pm 0.0007$ ($p < 10^{-100}$), all leading to decisive rejection of the null. For the Alzheimer's group, the same direction of effects was observed: Complexity decreased by $-0.024$ ($p < 10^{-10}$), RIS by $-0.12$ ($p = 4.6 \times 10^{-4}$), both rejecting the null, while ROS declined by $-0.09$ but did not reach significance ($p = 0.073$, FDR $q = 0.11$). Attribution metrics again showed dramatic reductions under SAE, with Grad-SHAP ($p < 10^{-45}$), Guided Backprop ($p = 3.2 \times 10^{-7}$), Integrated Gradients ($p < 10^{-55}$), and Optimizer ($p < 10^{-95}$) all supporting strong rejection of the null. Together these results demonstrate that SAE reliably improves attribution stability and reduces Complexity and RIS in both groups, with ROS showing only weak or inconsistent improvement.

For the three-class classification task, we evaluated whether SAE changed the three target metrics (Complexity, RIS, ROS) relative to no-SAE using paired $t$-tests and Wilcoxon signed-rank tests for each clinical group (Control, MCI, LMCI), applying Benjamini–Hochberg FDR to control multiplicity. For the MCI group, ROS showed the clearest and most consistent improvement with SAE: the paired $t$-test yielded $t(17) = -10.12$, $p = 1.30 \times 10^{-8}$ (FDR $q = 3.90 \times 10^{-8}$), and the Wilcoxon test yielded $W = 0$, $p = 8.0 \times 10^{-6}$ (FDR $q = 2.3 \times 10^{-5}$), with a very large paired Cohen's $d = -2.39$ and rank-biserial correlation $r_{rb} = -1.00$, indicating markedly lower ROS under SAE (mean difference $-0.904$; SAE 20.672 vs. no-SAE 21.576). RIS in MCI also decreased with SAE by non-parametric testing: the paired $t$-test did not reach significance ($t(18) = -0.785$, $p = 0.443$, FDR $q = 0.443$), whereas the Wilcoxon test detected a reduction ($W = 19$, $p = 0.00117$, FDR $q = 0.00117$), with small effect sizes ($d = -0.18$, $r_{rb} = -0.80$; mean difference $-0.481$; SAE 10.528 vs. no-SAE 11.010). In contrast, Complexity in MCI increased with SAE according to

Table 3: Evaluation scores with and without SAE. Values are mean ± std. Classes: Alzheimer and Control. Metrics: Sparseness (higher is better), RIS (lower is better), ROS (lower is better). Column order: Alzheimer (No SAE), Alzheimer (SAE), Control (No SAE), Control (SAE). All evaluation metrics were calculated on 50 randomly selected patients from each class (binary-class classification task) in the **BrainLat** testing cohort (OOD). Abreviations, TEO: Transformer Explanation Optimizer, TEO-UMAP: Transformer Explanation Optimizer with UMAP constraint.

| Method | Metric | Alzheimer (No SAE) | Alzheimer (SAE) | Control (No SAE) | Control (SAE) |
|---|---|---|---|---|---|
| Activation | Sparseness | N/A | 0.1533105 ± 0.010287697 | N/A | 0.3965415 ± 0.030322127 |
| | RIS | N/A | 19.162526 ± 0.364196762 | N/A | 18.2411505 ± 0.539197156 |
| | ROS | N/A | 31.28270375 ± 1.541354976 | N/A | 29.04058325 ± 0.473071038 |
| Layer Condact | Sparseness | N/A | 0.239227 ± 0.029777681 | N/A | 0.25431875 ± 0.020993978 |
| | RIS | N/A | 6.1620695 ± 0.149481666 | N/A | 6.21494775 ± 0.20764754 |
| | ROS | N/A | 16.94383425 ± 0.007089038 | N/A | 16.9402505 ± 0.004966063 |
| Feature Ablation | Sparseness | N/A | **0.5287845 ± 0.00700955** | N/A | **0.52849725 ± 0.004358269** |
| | RIS | N/A | 23.5834065 ± 0.064530846 | N/A | 24.14743175 ± 0.115957516 |
| | ROS | N/A | 34.65309725 ± 0.252584222 | N/A | 34.961296 ± 0.220544503 |
| Gradinet-SHAP | Sparseness | N/A | 0.120076 ± 0.014392456 | N/A | 0.057057 ± 0.027064338 |
| | RIS | N/A | 6.04400225 ± 0.039573077 | N/A | 6.030265 ± 0.047136969 |
| | ROS | N/A | 16.93474475 ± 5.76852E-05 | N/A | 16.93475925 ± 5.7373E-06 |
| Gradient Activation | Sparseness | N/A | 0.1139535 ± 0.01766843 | N/A | 0.062973 ± 0.006903384 |
| | RIS | N/A | 6.032837 ± 0.027736802 | N/A | 6.0338695 ± 0.039792395 |
| | ROS | N/A | 16.93468825 ± 3.59398E-06 | N/A | 16.934848 ± 3.74789E-05 |
| Integrated-Gradient | Sparseness | N/A | 0.0642685 ± 0.005166108 | N/A | 0.0143455 ± 0.000312693 |
| | RIS | N/A | 6.05793275 ± 0.045559192 | N/A | 6.0275535 ± 0.033936686 |
| | ROS | N/A | 16.9347635 ± 7.76745E-06 | N/A | 16.934873 ± 1.06145E-05 |
| TEO | Sparseness | N/A | 0.26914625 ± 0.001645095 | N/A | 0.272516 ± 0.000382866 |
| | RIS | N/A | **0.683544 ± 0.667616072** | N/A | **0.47335295 ± 0.280125046** |
| | ROS | N/A | **11.52356 ± 0.659063208** | N/A | **11.213036 ± 0.51496551** |
| TEO-UMAP | Sparseness | N/A | 0.398914 ± 0.00047836 | N/A | 0.40425175 ± 0.002851775 |
| | RIS | N/A | 5.43937375 ± 0.038349421 | N/A | 5.42815425 ± 0.194389712 |
| | ROS | N/A | 16.303675 ± 0.003852471 | N/A | 16.157664 ± 0.105405246 |

Table 4: Evaluation scores with and without SAE. Values are mean ± std. Classes: Control, LMCI, and MCI. Metrics: Sparseness (higher is better), RIS (lower is better), ROS (lower is better). All the evaluation metrics were computed on 100 randomly selected patients from each class (three-class classification task) in the testing cohort in **ADNI** dataset (IID). Column order: Control (No SAE), Control (SAE), LMCI (No SAE), LMCI (SAE), MCI (No SAE), MCI (SAE). Abreviations, TEO: Transformer Explanation Optimizer, TEO-UMAP: Transformer Explanation Optimizer with UMAP constraint.

| Method | Metric | Control | | LMCI | | MCI | |
|---|---|---|---|---|---|---|---|
| | | No SAE | SAE | No SAE | SAE | No SAE | SAE |
| Activation | Sparseness | 0.302953824 ± 0.037699004 | 0.345031647 ± 0.009533629 | 0.271540783 ± 0.038404292 | 0.264362053 ± 0.062996342 | 0.262558 ± 0.03794219 | 0.309052111 ± 0.060296425 |
| | RIS | 14.40424929 ± 0.165969844 | 18.99683335 ± 4.483970104 | 15.07860783 ± 1.975358824 | 18.42309221 ± 2.351829228 | 16.65684576 ± 2.820076491 | 19.42272406 ± 3.474528306 |
| | ROS | 25.721685 ± 0.170545275 | 30.30262794 ± 0.263575977 | 26.39512961 ± 1.972727766 | 29.73331658 ± 4.847156437 | 27.96063871 ± 2.823294834 | 30.74189461 ± 6.104484774 |
| Layer Condact | Sparseness | 0.231524647 ± 0.009587223 | 0.331457882 ± 0.006159197 | 0.362282261 ± 0.00636308 | 0.246395316 ± 0.062800978 | 0.305312706 ± 0.007627518 | 0.292950278 ± 0.057776928 |
| | RIS | 5.626004529 ± 0.020886163 | 5.622249412 ± 0.014860465 | 13.14289435 ± 0.325483065 | 5.623582526 ± 0.909878534 | 5.629072111 ± 1.130689807 | 5.629072111 ± 1.130689807 |
| | ROS | 16.94396553 ± 0.004532066 | 16.93904894 ± 0.010478577 | 24.50603352 ± 0.411948845 | 16.93375379 ± 2.745745658 | 17.91907076 ± 2.258068808 | 16.93837556 ± 3.406698804 |
| Feature Ablation | Sparseness | **0.523915176 ± 0.006693817** | 0.526105 ± 0.01204565 | **0.522595565 ± 0.009666847** | 0.526753263 ± 0.083976662 | **0.522188941 ± 0.009398367** | **0.525710278 ± 0.104819481** |
| | RIS | 23.32498194 ± 0.410939712 | 23.07664553 ± 0.140278365 | 22.24471861 ± 0.162941689 | 21.97939553 ± 3.68022107 | 23.49843053 ± 0.458719189 | 23.00058106 ± 4.540229242 |
| | ROS | 34.66532071 ± 0.457995824 | 34.41792582 ± 0.146270917 | 33.60637357 ± 0.161507737 | 33.30711989 ± 5.499686014 | 34.87369265 ± 0.440653725 | 34.31362572 ± 6.805012455 |
| Gradient-SHAP | Sparseness | 0.231029588 ± 0.020644371 | 0.184435333 ± 0.014766351 | 0.129241348 ± 0.032633243 | 0.301055444 ± 0.072119068 | 0.089142118 ± 0.013138265 | 0.288114875 ± 0.061769427 |
| | RIS | 5.618949294 ± 0.013910382 | 5.621940533 ± 0.025299231 | 5.615247522 ± 0.014396146 | 5.621698722 ± 0.946218825 | 5.629231 ± 0.018682697 | 5.618608438 ± 1.166983293 |
| | ROS | 16.93582388 ± 0.002076107 | 16.934845 ± 0.0000140509 | 16.92551835 ± 0.001442338 | 16.93477544 ± 2.862502663 | 16.93917776 ± 0.002118976 | 16.93476019 ± 3.524101451 |
| Guided Backprop | Sparseness | 0.269674235 ± 0.006145842 | 0.229587733 ± 0.00357389 | 0.383890696 ± 0.017652114 | 0.431001667 ± 0.115600632 | 0.291671824 ± 0.020033511 | 0.257909625 ± 0.109537561 |
| | RIS | 5.629017941 ± 0.022520137 | 5.621027267 ± 0.019431586 | 5.627154783 ± 0.021213565 | 5.629664833 ± 0.947801076 | 5.626876882 ± 0.019349509 | 5.617237063 ± 1.168361854 |
| | ROS | 16.934673 ± 0 | 16.9348278 ± 0.0000122544 | 16.93392839 ± 0.000750952 | 16.93466433 ± 2.862491655 | 16.93401118 ± 0.00064598 | 16.93471594 ± 3.524085042 |
| Integrated Gradient | Sparseness | 0.045146294 ± 0.007116898 | 0.263864 ± 0.004189771 | 0.10839887 ± 0.026164396 | 0.3889465 ± 0.084086627 | 0.110163824 ± 0.015744148 | 0.266043 ± 0.090525251 |
| | RIS | 5.620727706 ± 0.021492346 | 5.6209158 ± 0.020949259 | 5.609370435 ± 0.017753051 | 5.628201333 ± 0.947585532 | 5.628253647 ± 0.020904475 | 5.620282063 ± 1.168468694 |
| | ROS | 16.93310612 ± 0.001097405 | 16.93475353 ± 0.0000121647 | 16.92756339 ± 0.000606244 | 16.93434683 ± 2.862457357 | 16.93584506 ± 0.001969998 | 16.93460281 ± 3.524039 |
| TEO | Sparseness | 0.391835667 ± 0.000814648 | 0.268163938 ± 0.064942519 | 0.413087063 ± 0.000325772 | 0.285971625 ± 0.037421241 | 0.390886118 ± 0.004742559 | 0.283782105 ± 0.052291259 |
| | RIS | **4.807986067 ± 0.018432249** | **1.546787813 ± 0.11712595** | **5.093836 ± 0.018806243** | **2.264221 ± 0.487706388** | **4.828254294 ± 0.037727688** | **2.161698368 ± 0.454717751** |
| | ROS | **16.1172194 ± 0.008995191** | **12.856954 ± 0.117943866** | **16.40431106 ± 0.002382358** | **13.56455925 ± 2.274547046** | **16.13535541 ± 0.032411959** | **13.46760021 ± 2.764087874** |
| TEO-UMAP | Sparseness | N/A | 0.39734881 ± 0.07492051 | N/A | 0.41611163 ± 0.08698112 | N/A | 0.41715421 ± 0.237175073 |
| | RIS | N/A | 5.10862522 ± 0.20827341 | N/A | 5.10165242 ± 0.16974677 | N/A | 5.11160575 ± 0.1072146 |
| | ROS | N/A | 16.412262 ± 6.84387466 | N/A | 16.4031485 ± 3.86158978 | N/A | 16.4088329 ± 0.492439623 |

the Wilcoxon test ($W = 17$, $p = 7.90 \times 10^{-4}$, FDR $q = 0.00117$), while the paired $t$-test was non-significant ($t(18) = 1.112$, $p = 0.281$, FDR $q = 0.421$); effect sizes were small-to-moderate ($d = 0.26$, $r_{\rm rb} = 0.821$; mean difference +0.510; SAE 1.175 vs. no-SAE 0.665). For the **Control** and **LMCI** groups, the pasted records contained incomplete pairs that prevented reliable paired testing

Table 5: Evaluation scores with and without SAE. Values are mean ± std. Classes: Control, LMCI, and MCI. Metrics: Sparseness (higher is better), RIS (lower is better), ROS (lower is better). Column order: Control (No SAE), Control (SAE), LMCI (No SAE), LMCI (SAE), MCI (No SAE), MCI (SAE). All evaluation metrics were calculated on 50 randomly selected patients from each class (three-class classification task) in the **BrainLat** testing cohort (OOD). Abreviations, TEO: Transformer Explanation Optimizer, TEO-UMAP: Transformer Explanation Optimizer with UMAP constraint.

| Method | Metric | Control | | LMCI | | MCI | |
|---|---|---|---|---|---|---|---|
| | | No SAE | SAE | No SAE | SAE | No SAE | SAE |
| Activation | Sparseness | N/A | 0.4504596 ± 0.037517308 | N/A | 0.1907182 ± 0.001584043 | N/A | 0.140032667 ± 0.012393238 |
| | RIS | N/A | 19.0939584 ± 0.179649958 | N/A | 18.6406378 ± 0.911739674 | N/A | 18.07866133 ± 0.031343749 |
| | ROS | N/A | 29.9240256 ± 0.253797244 | N/A | 29.5628004 ± 0.89780058 | N/A | 28.858315 ± 0.045164984 |
| Layer Condact | Sparseness | N/A | 0.3252352 ± 0.014541625 | N/A | 0.185706 ± 0.007343355 | N/A | 0.200561 ± 0.011873024 |
| | RIS | N/A | 6.2120432 ± 0.245036193 | N/A | 6.054585 ± 0.047710839 | N/A | 6.268400333 ± 0.040128533 |
| | ROS | N/A | 16.9581856 ± 0.017257696 | N/A | 16.9636788 ± 0.007069011 | N/A | 17.01458633 ± 0.020606806 |
| Feature Ablation | Sparseness | N/A | **0.5280516 ± 0.00583783** | N/A | **0.526218 ± 0.005664214** | N/A | **0.529347333 ± 0.011164491** |
| | RIS | N/A | 22.640559 ± 0.033054016 | N/A | 23.5692968 ± 0.057715633 | N/A | 23.59159233 ± 0.111776179 |
| | ROS | N/A | 33.485311 ± 0.233803422 | N/A | 34.5168758 ± 0.073725495 | N/A | 34.37202467 ± 0.080383672 |
| Gradinet-SHAP | Sparseness | N/A | 0.195138 ± 0.026410904 | N/A | 0.0637004 ± 0.026472282 | N/A | 0.113682167 ± 0.042008189 |
| | RIS | N/A | 6.122703167 ± 0.161879331 | N/A | 6.0314946 ± 0.029788567 | N/A | 6.127377667 ± 0.120700168 |
| | ROS | N/A | 16.93462983 ± 4.27103E-05 | N/A | 16.9348698 ± 7.57377E-05 | N/A | 16.93466933 ± 8.86649E-05 |
| Gradient Activation | Sparseness | N/A | 0.177242 ± 0.011243388 | N/A | 0.1835882 ± 0.001632398 | N/A | 0.430289167 ± 0.00215046 |
| | RIS | N/A | 6.123393833 ± 0.191171312 | N/A | 6.0269246 ± 0.030177701 | N/A | 6.144976167 ± 0.120931658 |
| | ROS | N/A | 16.93457917 ± 1.47434E-05 | N/A | 16.9347678 ± 2.58844E-06 | N/A | 16.934534 ± 2.79285E-05 |
| Integrated-Gradient | Sparseness | N/A | 0.067058 ± 0.012083245 | N/A | 0.0071952 ± 0.000900684 | N/A | 0.036059 ± 0.004866926 |
| | RIS | N/A | 6.1224575 ± 0.150190804 | N/A | 6.035594 ± 0.01894045 | N/A | 6.147797667 ± 0.09243145 |
| | ROS | N/A | 16.93462633 ± 1.30486E-05 | N/A | 16.9347694 ± 1.14018E-06 | N/A | 16.934618 ± 8.89944E-06 |
| TEO | Sparseness | N/A | 0.416191667 ± 0.002863111 | N/A | 0.3715978 ± 0.000948703 | N/A | 0.42242125 ± 0.000173513 |
| | RIS | N/A | 5.752004667 ± 0.364536772 | N/A | 4.9396128 ± 0.014829469 | N/A | **5.5420845 ± 0.061116734** |
| | ROS | N/A | **16.379228 ± 0.003422144** | N/A | **15.8121004 ± 0.00994492** | N/A | **16.277279 ± 0.001043319** |
| TEO-UMAP | Sparseness | N/A | 0.423819167 ± 0.00056124 | N/A | 0.423865 ± 5.01946E-05 | N/A | 0.424567429 ± 0.000241638 |
| | RIS | N/A | **5.552539167 ± 0.186236877** | N/A | 5.458319 ± 0.029725596 | N/A | 5.557568429 ± 0.095350875 |
| | ROS | N/A | **16.3661035 ± 0.00731861** | N/A | 16.372555 ± 0.001689458 | N/A | 16.357192 ± 0.003475647 |

and FDR-adjusted inference in the same aggregate framework; consequently, we do not report hypothesis tests for these groups here to avoid bias from unmatched rows. Overall, across the three groups, the most robust and reproducible effect we could quantify was the *reduction in ROS under SAE* (clearly demonstrated in MCI with converged paired comparisons), while RIS showed a smaller SAE-related decrease by non-parametric testing and Complexity tended to increase under SAE for MCI.

## B.6 INDIVIDUAL-LEVEL EXPLANATIONS AND PATTERNS

Figures 7–16 present qualitative local attribution examples for the binary (Control and Alzheimer) and three-class classification task (Control, LMCI, MCI) of ADNI cohorts across six explanation methods, each evaluated without (Figures 7, 9, 11, 13, 15) and with (Figures 8, 10, 12, 14, 16) the Sparse Autoencoder (SAE) layer. Each cell shows token-level attributions using colour-coded highlights (green = positive relevance; red = negative relevance). In general, higher Sparseness is associated with a more balanced distribution of positive and negative highlights (i.e., less diffuse maps), particularly for Layer Conduction, Feature Ablation, Gradient SHAP, and Integrated Gradient. For the Control class (see Figures 7 and 8), the qualitative highlighting patterns are broadly consistent across the six attribution techniques, Activation, Layer Conduction, Feature Ablation, Gradient SHAP, Gradient Activation, and Integrated Gradient—with no marked visual discrepancies. Notably, Feature Ablation, despite exhibiting the strongest Sparseness in the box plots, shows poorer stability (higher variability in inputs/outputs; elevated RIS/ROS), and the addition of the SAE layer tends to worsen this by exposing a larger set of features due to the decoder "decompression" effect; a similar trend is observed for Activation. For the Alzheimer's class (Figures 9 and 10), Layer Conduction demonstrates a reduction in Sparseness with the SAE but a gain in stability (decreased RIS/ROS). Comparable improvements in stability with SAE are also observed for Gradient Activation, Integrated Gradient, and Gradient SHAP. In contrast, Activation and Feature Ablation perform worst under SAE, again exposing many more features and yielding less stable explanations. Across the remaining examples (Figures 11–17), similar patterns hold: instances with low Sparseness and high RIS/ROS tend to produce saturated red/green

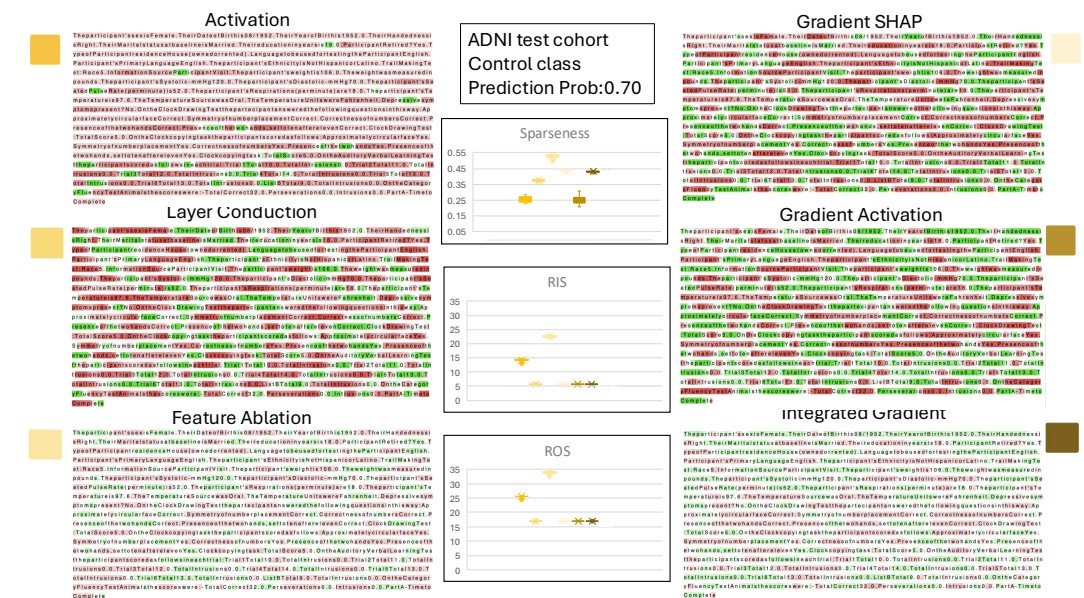

Figure 7: Local attribution examples across different explanation methods without the SAE layer. The colour scale ranges from −1 (dark red; negative attribution), through 0 (white; neutral), to +1 (dark green; positive attribution). For each of the six panels, the small colour swatches at the top-left and top-right indicate the colour keys used for the three summary box-plot metrics—Sparseness, RIS, and ROS—for the corresponding attribution technique. The task is a binary classification (Alzheimer's disease vs Control) on the ADNI cohort; the examples shown here are from the Control class.

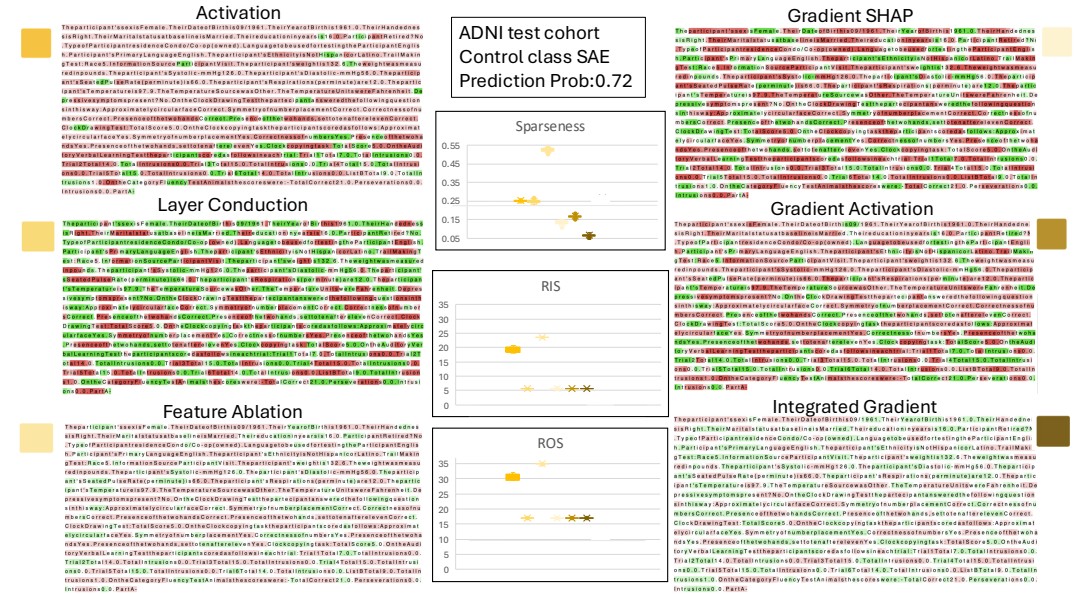

Figure 8: Local attribution examples across different explanation methods with the SAE layer. The colour scale ranges from −1 (dark red; negative attribution), through 0 (white; neutral), to +1 (dark green; positive attribution). For each of the six panels, the small colour swatches at the top-left and top-right indicate the colour keys used for the three summary box-plot metrics—Sparseness, RIS, and ROS—for the corresponding attribution technique. The task is a binary classification (Alzheimer's disease vs Control) on the ADNI cohort; the examples shown here are from the Control class.

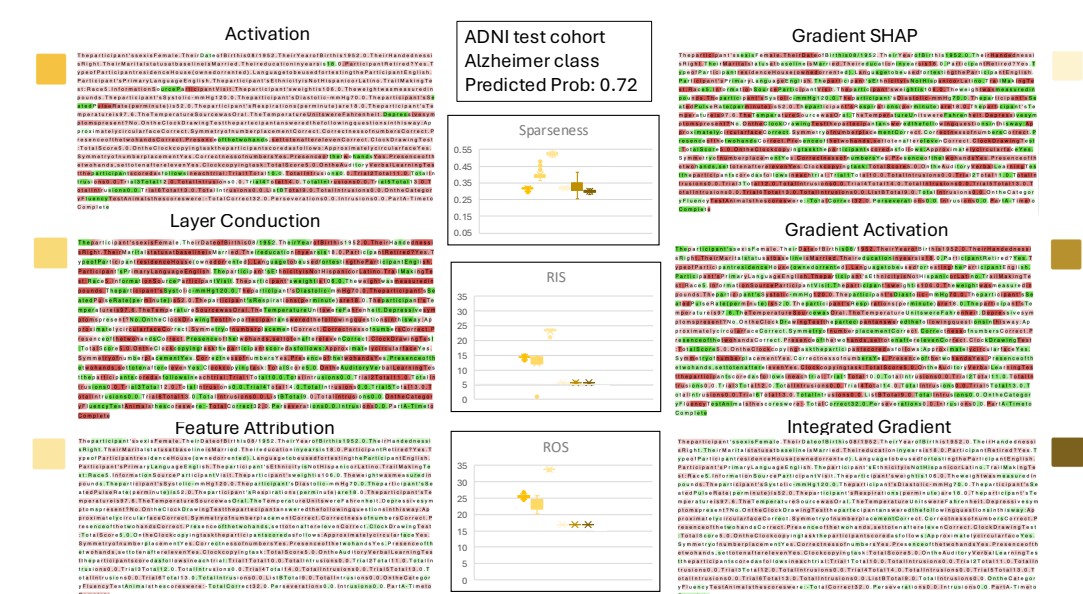

Figure 9: Local attribution examples across different explanation methods without the SAE layer. The colour scale ranges from −1 (dark red; negative attribution), through 0 (white; neutral), to +1 (dark green; positive attribution). For each of the six panels, the small colour swatches at the top-left and top-right indicate the colour keys used for the three summary box-plot metrics—Sparseness, RIS, and ROS—for the corresponding attribution technique. The task is a binary classification (Alzheimer's disease vs Control) on the ADNI cohort; the examples shown here are from the Alzheimer class.

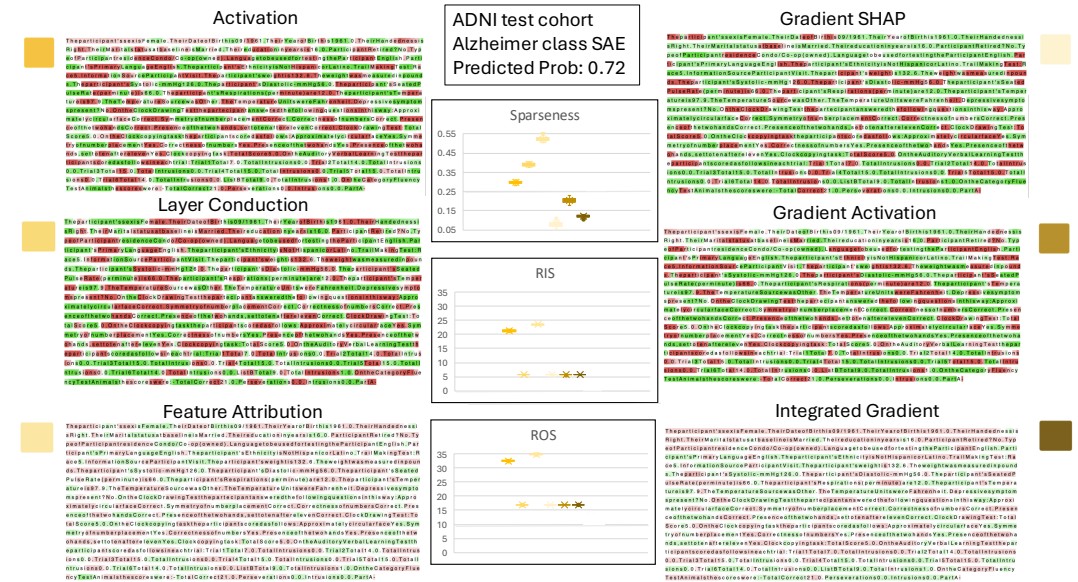

Figure 10: Local attribution examples across different explanation methods with the SAE layer. The colour scale ranges from −1 (dark red; negative attribution), through 0 (white; neutral), to +1 (dark green; positive attribution). For each of the six panels, the small colour swatches at the top-left and top-right indicate the colour keys used for the three summary box-plot metrics—Sparseness, RIS, and ROS—for the corresponding attribution technique. The task is a binary classification (Alzheimer's disease vs Control) on the ADNI cohort; the examples shown here are from the Alzheimer class.

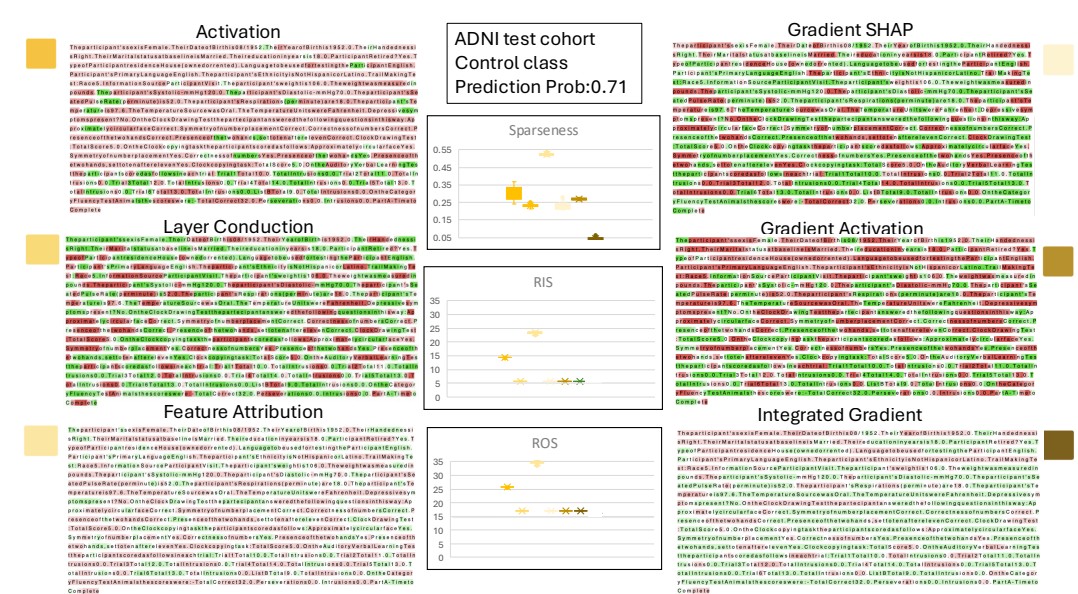

Figure 11: Local attribution examples across different explanation methods without the SAE layer. The colour scale ranges from −1 (dark red; negative attribution), through 0 (white; neutral), to +1 (dark green; positive attribution). For each of the six panels, the small colour swatches at the top-left and top-right indicate the colour keys used for the three summary box-plot metrics—Sparseness, RIS, and ROS—for the corresponding attribution technique. The task is a three-class classification (LMCI, MCI disease vs Control) on the ADNI cohort; the examples shown here are from the Control class.

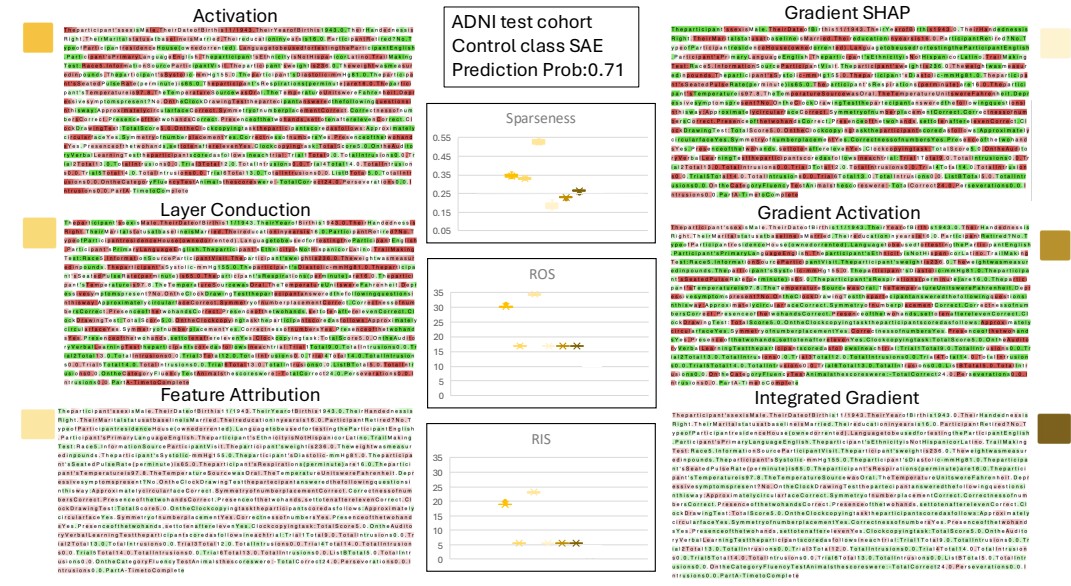

Figure 12: Local attribution examples across different explanation methods with the SAE layer. The colour scale ranges from −1 (dark red; negative attribution), through 0 (white; neutral), to +1 (dark green; positive attribution). For each of the six panels, the small colour swatches at the top-left and top-right indicate the colour keys used for the three summary box-plot metrics—Sparseness, RIS, and ROS—for the corresponding attribution technique. The task is a three-class classification (LMCI, MCI disease vs Control) on the ADNI cohort; the examples shown here are from the Control class.

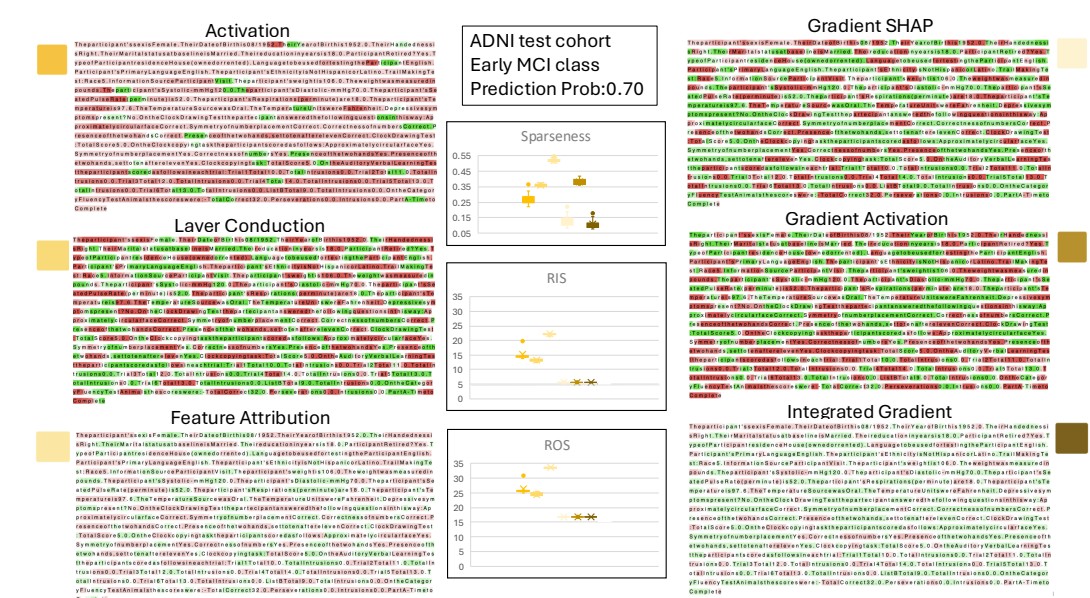

Figure 13: Local attribution examples across different explanation methods without the SAE layer. The colour scale ranges from −1 (dark red; negative attribution), through 0 (white; neutral), to +1 (dark green; positive attribution). For each of the six panels, the small colour swatches at the top-left and top-right indicate the colour keys used for the three summary box-plot metrics—Sparseness, RIS, and ROS—for the corresponding attribution technique. The task is a three-class classification (LMCI, MCI disease vs Control) on the ADNI cohort; the examples shown here are from the LMCI class.

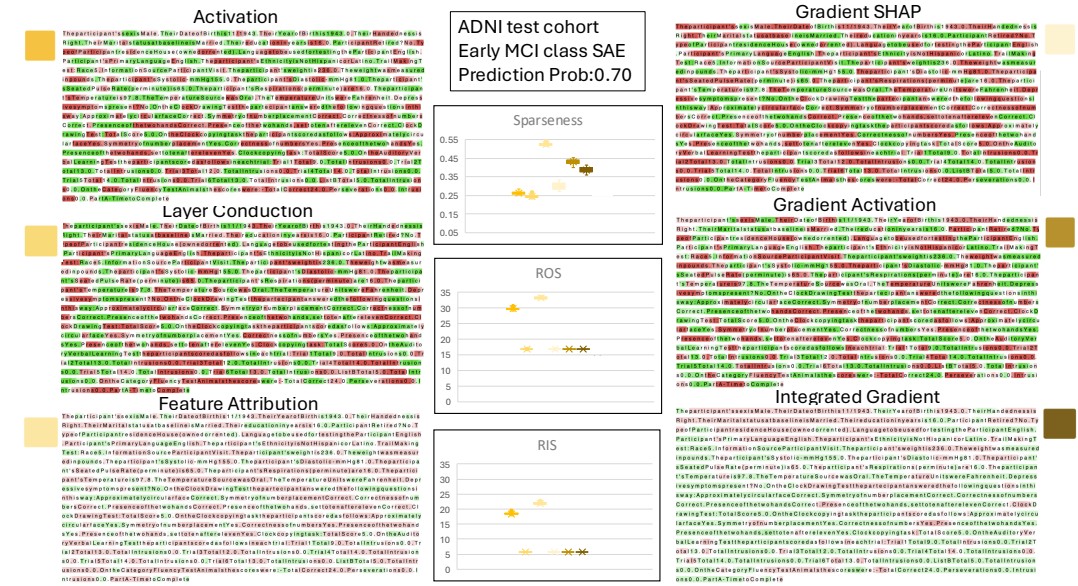

Figure 14: Local attribution examples across different explanation methods with the SAE layer. The colour scale ranges from −1 (dark red; negative attribution), through 0 (white; neutral), to +1 (dark green; positive attribution). For each of the six panels, the small colour swatches at the top-left and top-right indicate the colour keys used for the three summary box-plot metrics—Sparseness, RIS, and ROS—for the corresponding attribution technique. The task is a three-class classification (LMCI, MCI disease vs Control) on the ADNI cohort; the examples shown here are from the LMCI class.

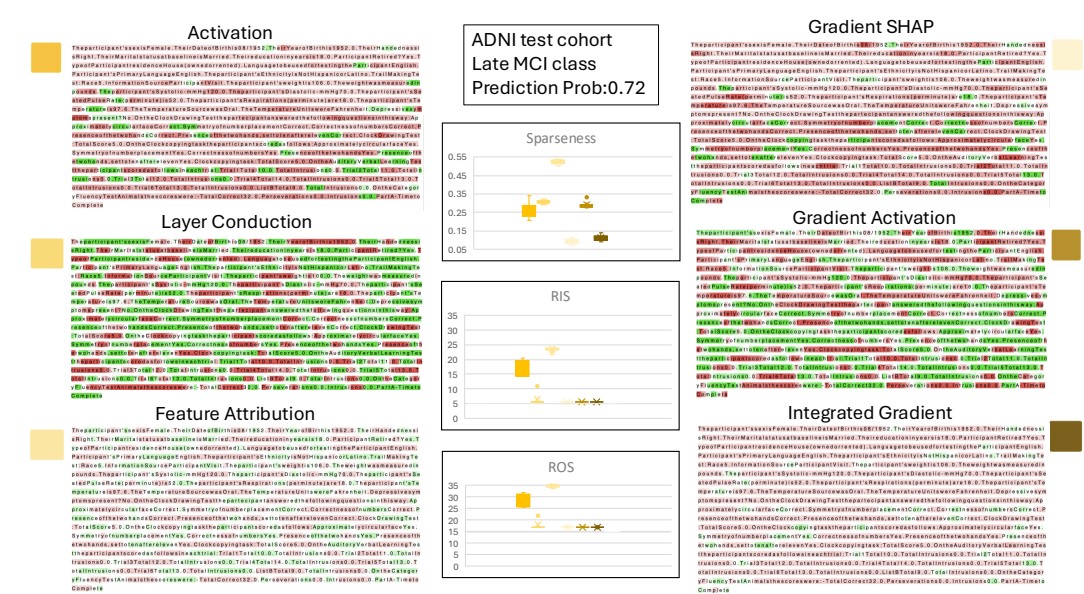

Figure 15: Local attribution examples across different explanation methods without the SAE layer. The colour scale ranges from −1 (dark red; negative attribution), through 0 (white; neutral), to +1 (dark green; positive attribution). For each of the six panels, the small colour swatches at the top-left and top-right indicate the colour keys used for the three summary box-plot metrics—Sparseness, RIS, and ROS—for the corresponding attribution technique. The task is a three-class classification (LMCI, MCI disease vs Control) on the ADNI cohort; the examples shown here are from the MCI class.

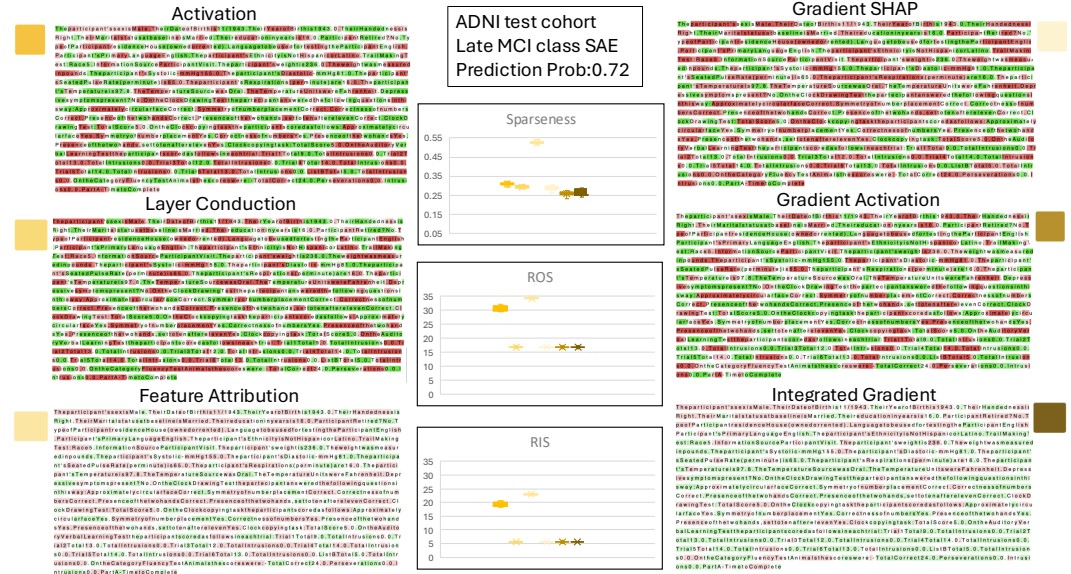

Figure 16: Local attribution examples across different explanation methods with the SAE layer. The colour scale ranges from −1 (dark red; negative attribution), through 0 (white; neutral), to +1 (dark green; positive attribution). For each of the six panels, the small colour swatches at the top-left and top-right indicate the colour keys used for the three summary box-plot metrics—Sparseness, RIS, and ROS—for the corresponding attribution technique. The task is a three-class classification (LMCI, MCI disease vs Control) on the ADNI cohort; the examples shown here are from the MCI class.

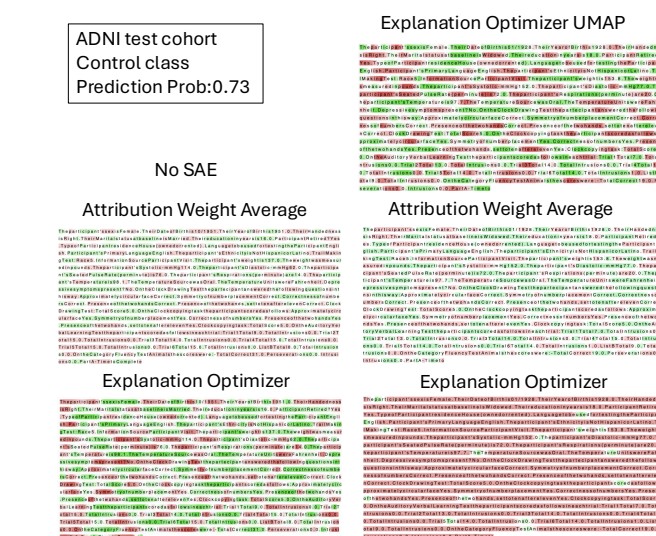

Figure 17: Local attribution examples across different explanation methods with the SAE layer. The colour scale ranges from −1 (dark red; negative attribution), through 0 (white; neutral), to +1 (dark green; positive attribution). For each of the six panels, the small colour swatches at the top-left and top-right indicate the colour keys used for the three summary box-plot metrics—Sparseness, RIS, and ROS—for the corresponding attribution technique. The task is a binary classification (Alzheimer's disease vs Control) on the ADNI cohort; the examples shown here are from the Control class.

explanations (strongly negative or positive attributions), whereas higher Sparseness with lower RIS/ROS yields more compact and stable saliency patterns.

Figures 17–21 present qualitative local attribution examples, analogous to Figures 7–17, for the no-SAE analyses of (i) the attributional weighted average (computed from the six base methods), (ii) the Transformer Explanation Optimizer (TEO), and (iii) TEO with a linear UMAP constraint (TEO-UMAP). As shown in the previous subsection, with the SAE layer TEO achieves the best stability—i.e., the lowest RIS and ROS—but at the cost of a marked reduction in Sparseness; this reduction is clearly visible in the binary task (Figures 17–18). Introducing the UMAP constraint yields a more balanced trade-off, producing explanations that are more compact and clinically interpretable; the same behaviour is observed across all classes in the three-class setting (Figures 19–21). By contrast, the weighted-average approach—a linear combination of the six attribution techniques—does not yield superior explanations, consistent with Mamalakis et al. (2025).

## B.7 UMAP AND COHORT-LEVEL EXPLANATION AND PATTERNS.

Figures 22–31 present cohort-level attribution examples for both the binary (Control vs Alzheimer's disease) and three-class (Control, LMCI, MCI) classification tasks on the ADNI test cohort across six explanation methods. Each method is shown without (Figures 22, 24, 26, 28, 30) and with (Figures 23, 25, 27, 29, 31) the Sparse Autoencoder (SAE) layer. The 2D panel shows a UMAP embedding (UMAP-1 vs UMAP-2) computed on the ADNI test set; the 1D panel shows attribution scores along PCA-first component. All plotted values are normalised to [0,1] and represent positive contributions only. Colours (red→purple) denote the nine ADNI subgroups (see §B3). In general, moving from the no-SAE to the SAE condition broadens the distribution of features in 2D and increases the density of high-significance points (upper-right boxed region), consistent with a decoder-induced decompression effect and a corresponding reduction in sparsity in the attribution maps.

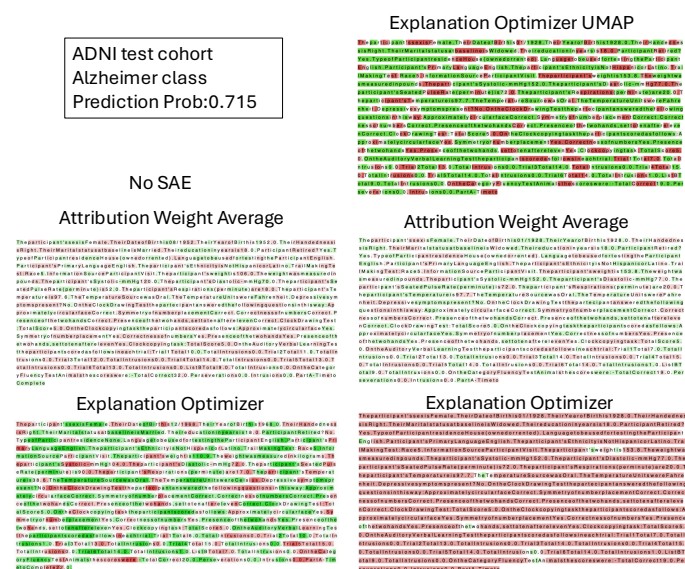

Figure 18: Local attribution examples across different explanation methods without the SAE layer. The colour scale ranges from −1 (dark red; negative attribution), through 0 (white; neutral), to +1 (dark green; positive attribution). For each of the six panels, the small colour swatches at the top-left and top-right indicate the colour keys used for the three summary box-plot metrics—Sparseness, RIS, and ROS—for the corresponding attribution technique. The task is a binary classification (Alzheimer's disease vs Control) on the ADNI cohort; the examples shown here are from the Alzheimer class.

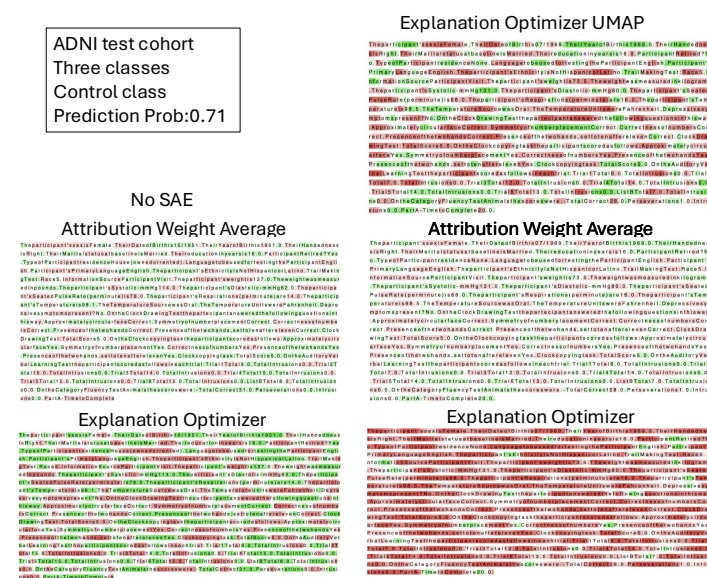

Figure 19: Local attribution examples across different explanation methods with the SAE layer. The colour scale ranges from −1 (dark red; negative attribution), through 0 (white; neutral), to +1 (dark green; positive attribution). For each of the six panels, the small colour swatches at the top-left and top-right indicate the colour keys used for the three summary box-plot metrics—Sparseness, RIS, and ROS—for the corresponding attribution technique. The task is a three-class classification (LMCI, MCI disease vs Control) on the ADNI cohort; the examples shown here are from the Control class.

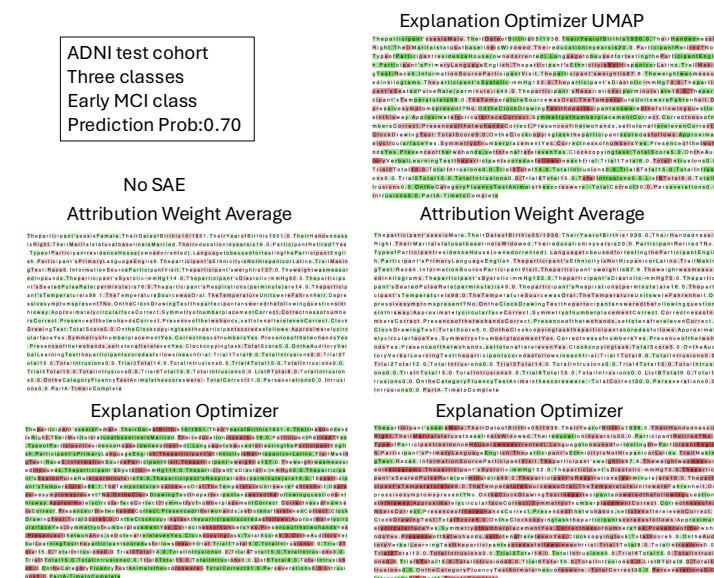

Figure 20: Local attribution examples across different explanation methods without the SAE layer. The colour scale ranges from −1 (dark red; negative attribution), through 0 (white; neutral), to +1 (dark green; positive attribution). For each of the six panels, the small colour swatches at the top-left and top-right indicate the colour keys used for the three summary box-plot metrics—Sparseness, RIS, and ROS—for the corresponding attribution technique. The task is a three-class classification (LMCI, MCI disease vs Control) on the ADNI cohort; the examples shown here are from the LMCI class.

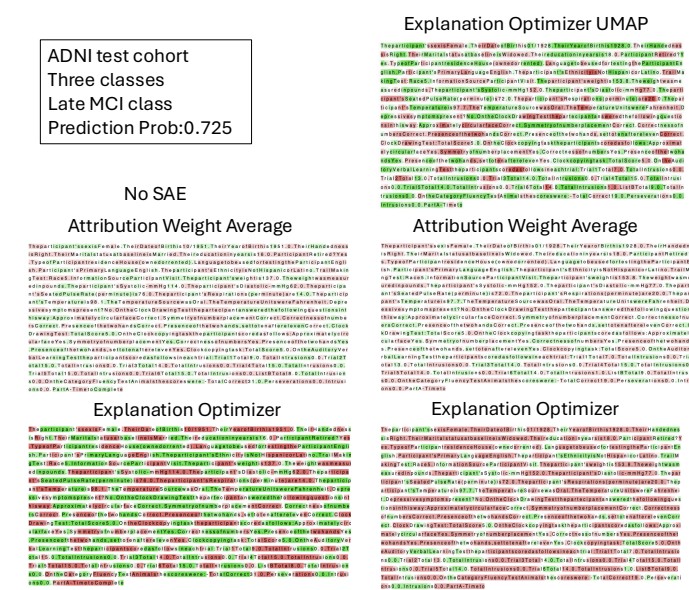

Figure 21: Local attribution examples across different explanation methods without the SAE layer. The colour scale ranges from −1 (dark red; negative attribution), through 0 (white; neutral), to +1 (dark green; positive attribution). For each of the six panels, the small colour swatches at the top-left and top-right indicate the colour keys used for the three summary box-plot metrics—Sparseness, RIS, and ROS—for the corresponding attribution technique. The task is a three-class classification (LMCI, MCI disease vs Control) on the ADNI cohort; the examples shown here are from the MCI class.

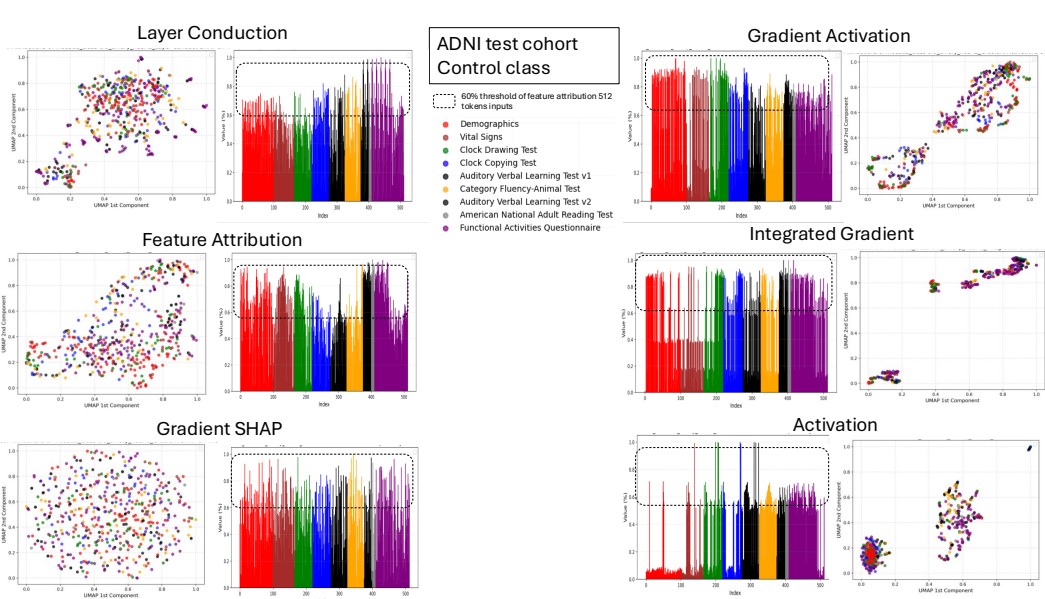

Figure 22: Global (cohort-level) feature attribution across explanation methods without the SAE layer. The 2D panel shows a UMAP embedding (UMAP-1 vs UMAP-2) computed on the ADNI test set; the 1D panel shows attribution scores along PCA first component. All plotted values are normalised to [0, 1] and represent positive contributions only. Colours (red→purple) denote the nine ADNI subgroups (see §B3). Square boxes mark the 0.6–1.0 interval, highlighting the most significant tokens in the 1D views. The task is binary classification (Alzheimer's vs Control) on the ADNI cohort; the examples shown here are from the Control class.The task is a three-class classification (LMCI, MCI disease vs Control) on the ADNI cohort; the examples shown here are from the MCI class.

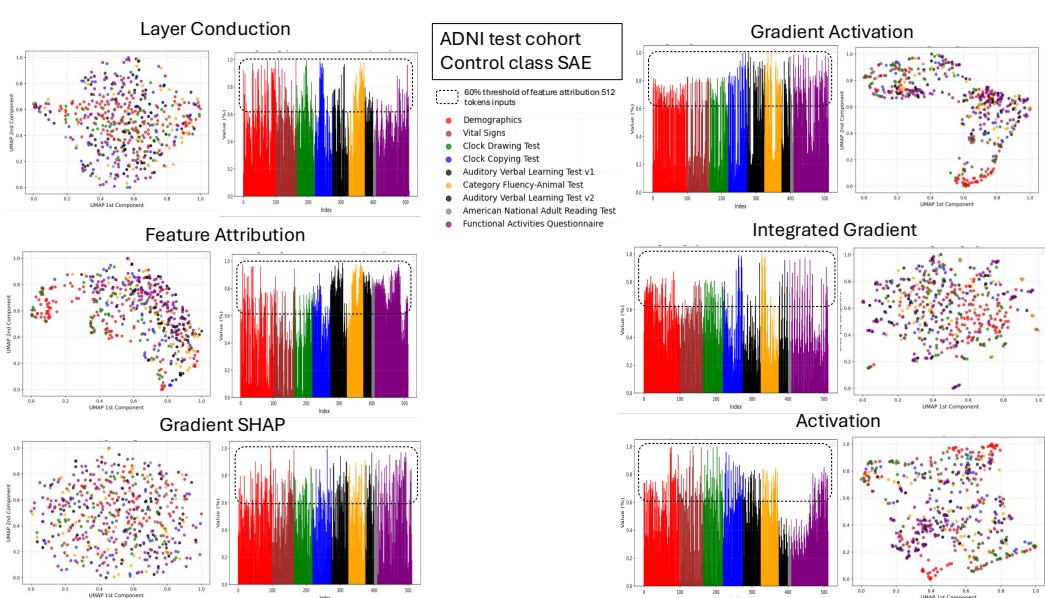

Figure 23: Global (cohort-level) feature attribution across explanation methods with the SAE layer. The 2D panel shows a UMAP embedding (UMAP-1 vs UMAP-2) computed on the ADNI test set; the 1D panel shows attribution scores along PCA first component. All plotted values are normalised to [0, 1] and represent positive contributions only. Colours (red→purple) denote the nine ADNI subgroups (see §B3). Square boxes mark the 0.6–1.0 interval, highlighting the most significant tokens in the 1D views. The task is binary classification (Alzheimer's vs Control) on the ADNI cohort; the examples shown here are from the Control class.

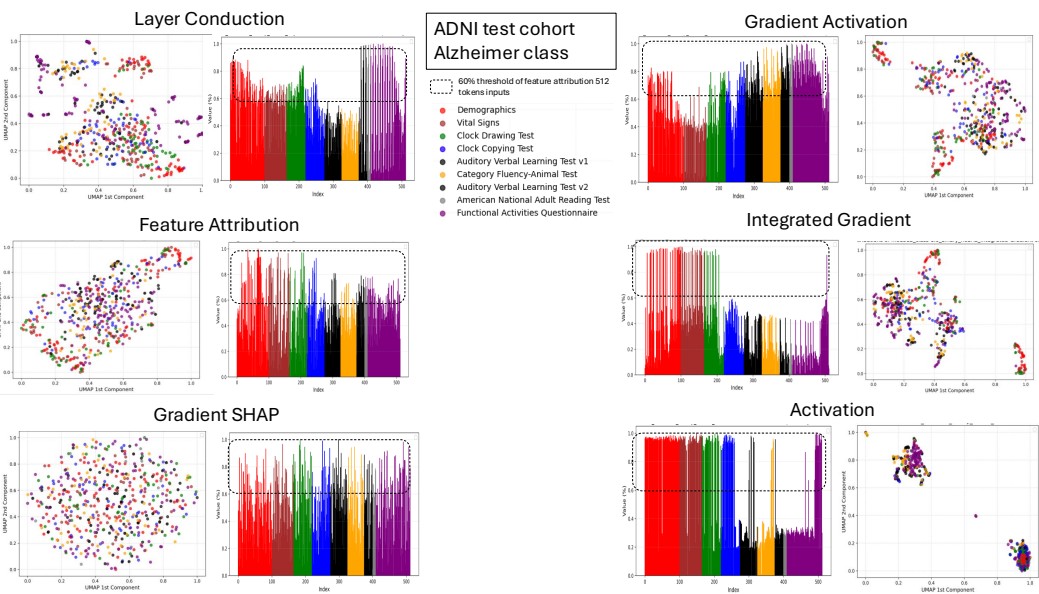

Figure 24: Global (cohort-level) feature attribution across explanation methods without the SAE layer. The 2D panel shows a UMAP embedding (UMAP-1 vs UMAP-2) computed on the ADNI test set;the 1D panel shows attribution scores along PCA first component. All plotted values are normalised to [0, 1] and represent positive contributions only. Colours (red→purple) denote the nine ADNI subgroups (see §B3). Square boxes mark the 0.6–1.0 interval, highlighting the most significant tokens in the 1D views. The task is binary classification (Alzheimer's vs Control) on the ADNI cohort; the examples shown here are from the Alzheimer class.

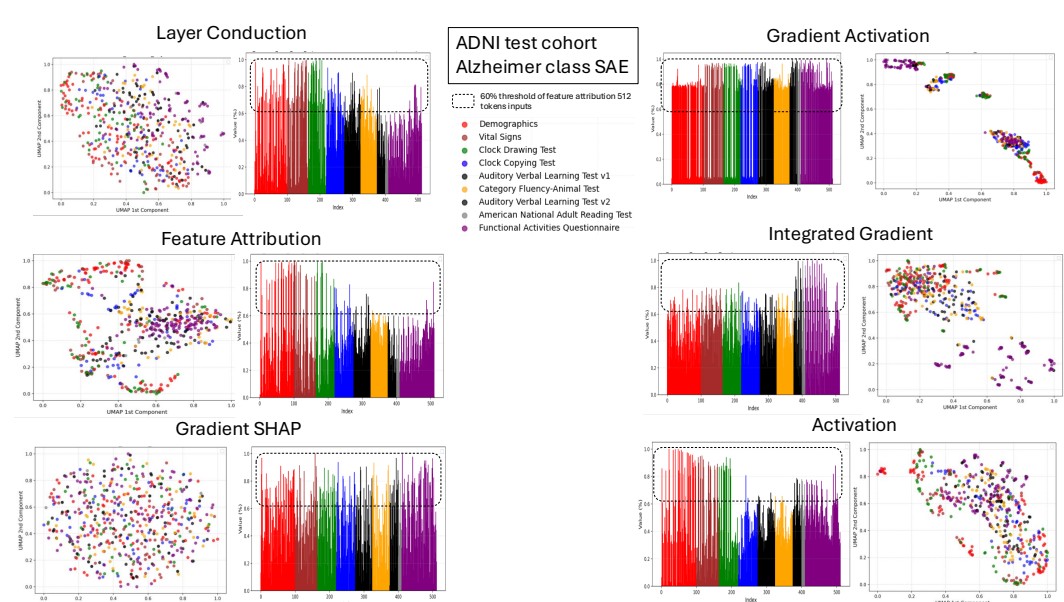

Figure 25: Global (cohort-level) feature attribution across explanation methods with the SAE layer. The 2D panel shows a UMAP embedding (UMAP-1 vs UMAP-2) computed on the ADNI test setthe 1D panel shows attribution scores along PCA first component. All plotted values are normalised to [0, 1] and represent positive contributions only. Colours (red→purple) denote the nine ADNI subgroups (see §B3). Square boxes mark the 0.6–1.0 interval, highlighting the most significant tokens in the 1D views. The task is binary classification (Alzheimer's vs Control) on the ADNI cohort; the examples shown here are from the Alzheimer class.

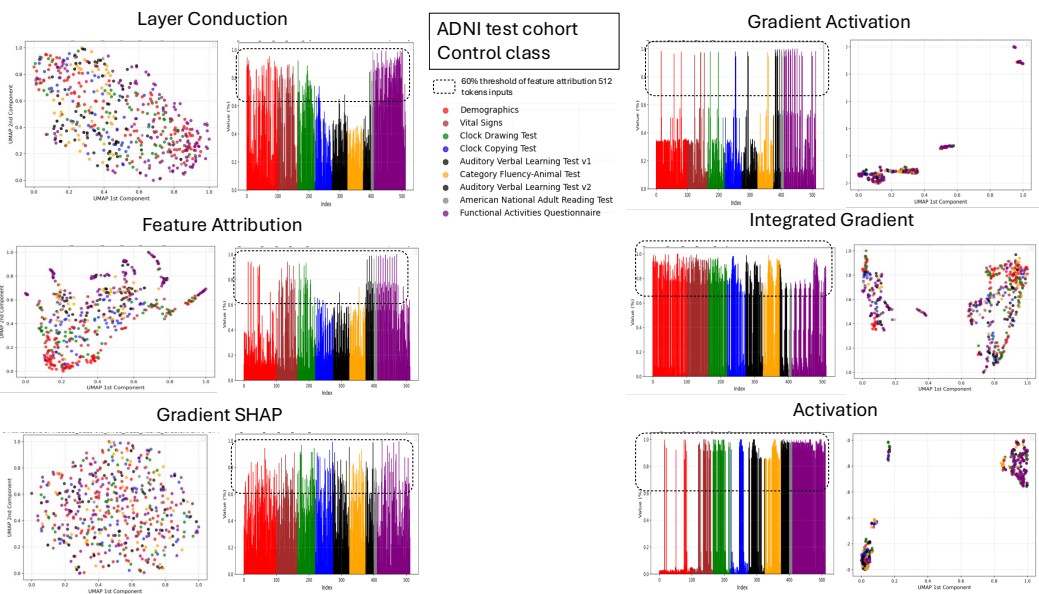

Figure 26: Global (cohort-level) feature attribution across explanation methods without the SAE layer. The 2D panel shows a UMAP embedding (UMAP-1 vs UMAP-2) computed on the ADNI test setthe 1D panel shows attribution scores along PCA first component. All plotted values are normalised to [0, 1] and represent positive contributions only. Colours (red→purple) denote the nine ADNI subgroups (see §B3). Square boxes mark the 0.6–1.0 interval, highlighting the most significant tokens in the 1D views. The task is a three-class classification (LMCI, MCI disease vs Control) on the ADNI cohort; the examples shown here are from the Control class.

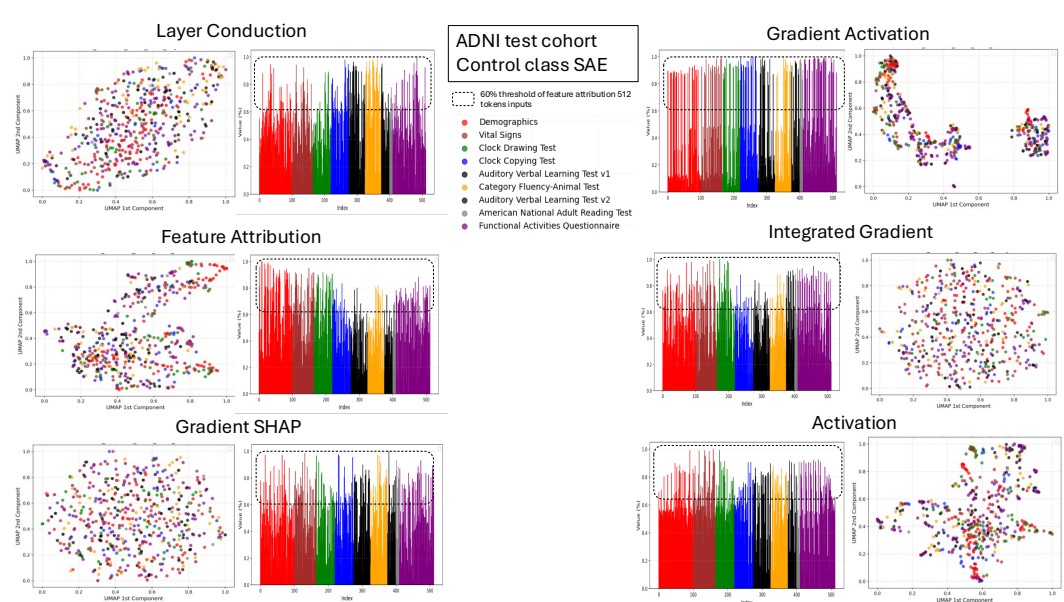

Figure 27: Global (cohort-level) feature attribution across explanation methods with the SAE layer. The 2D panel shows a UMAP embedding (UMAP-1 vs UMAP-2) computed on the ADNI test setthe 1D panel shows attribution scores along PCA first component. All plotted values are normalised to [0, 1] and represent positive contributions only. Colours (red→purple) denote the nine ADNI subgroups (see §B3). Square boxes mark the 0.6–1.0 interval, highlighting the most significant tokens in the 1D views. The task is a three-class classification (LMCI, MCI disease vs Control) on the ADNI cohort; the examples shown here are from the Control class.

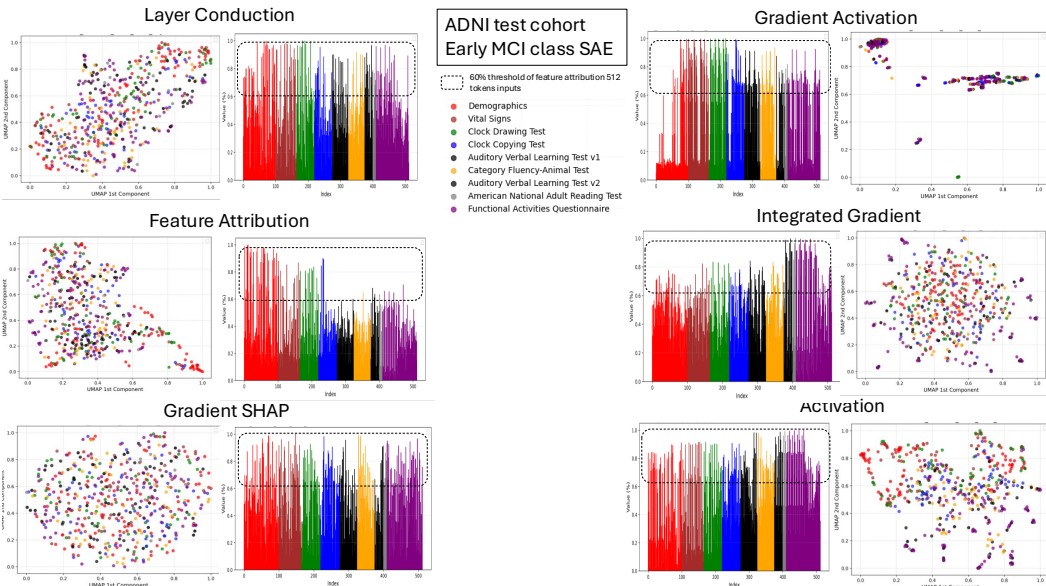

Figure 28: Global (cohort-level) feature attribution across explanation methods with the SAE layer. The 2D panel shows a UMAP embedding (UMAP-1 vs UMAP-2) computed on the ADNI test set; the 1D panel shows attribution scores along PCA first component. All plotted values are normalised to [0, 1] and represent positive contributions only. Colours (red→purple) denote the nine ADNI subgroups (see §B3). Square boxes mark the 0.6–1.0 interval, highlighting the most significant tokens in the 1D views. The task is a three-class classification (LMCI, MCI disease vs Control) on the ADNI cohort; the examples shown here are from the LMCI class.

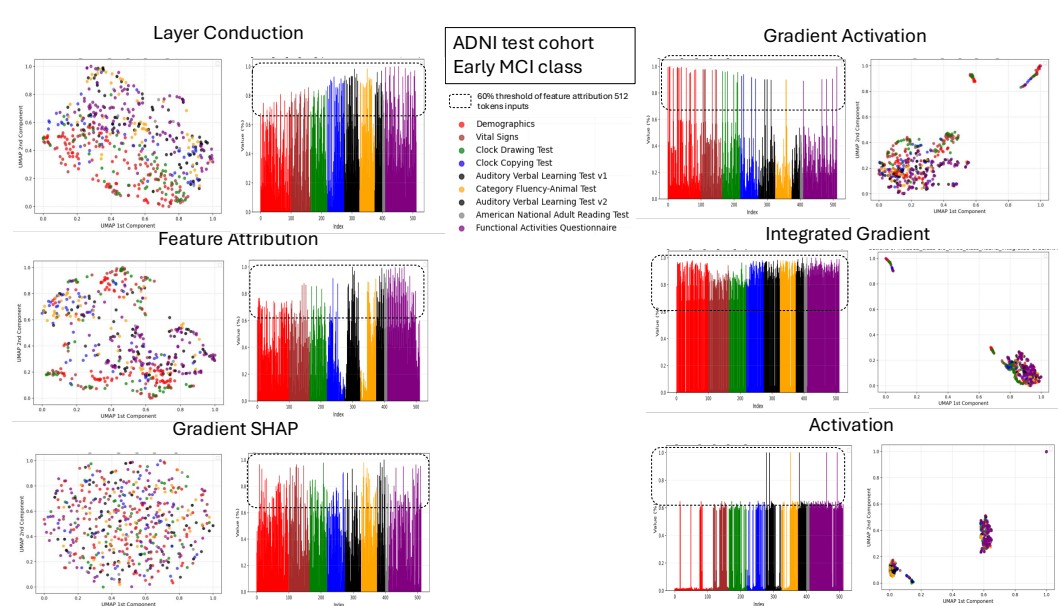

Figure 29: Global (cohort-level) feature attribution across explanation methods without the SAE layer. The 2D panel shows a UMAP embedding (UMAP-1 vs UMAP-2) computed on the ADNI test set; the 1D panel shows attribution scores along PCA first component. All plotted values are normalised to [0, 1] and represent positive contributions only. Colours (red→purple) denote the nine ADNI subgroups (see §B3). Square boxes mark the 0.6–1.0 interval, highlighting the most significant tokens in the 1D views. The task is a three-class classification (LMCI, MCI disease vs Control) on the ADNI cohort; the examples shown here are from the LMCI class.

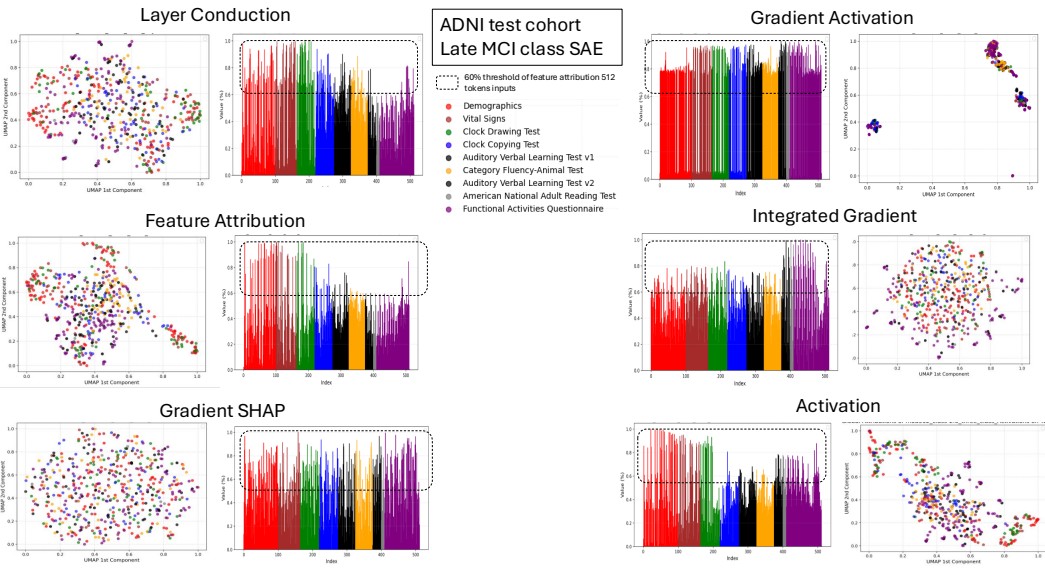

Figure 30: Global (cohort-level) feature attribution across explanation methods with the SAE layer. The 2D panel shows a UMAP embedding (UMAP-1 vs UMAP-2) computed on the ADNI test set; the 1D panel shows attribution scores along PCA first component. All plotted values are normalised to [0, 1] and represent positive contributions only. Colours (red→purple) denote the nine ADNI subgroups (see §B3). Square boxes mark the 0.6–1.0 interval, highlighting the most significant tokens in the 1D views. The task is a three-class classification (LMCI, MCI disease vs Control) on the ADNI cohort; the examples shown here are from the MCI class.

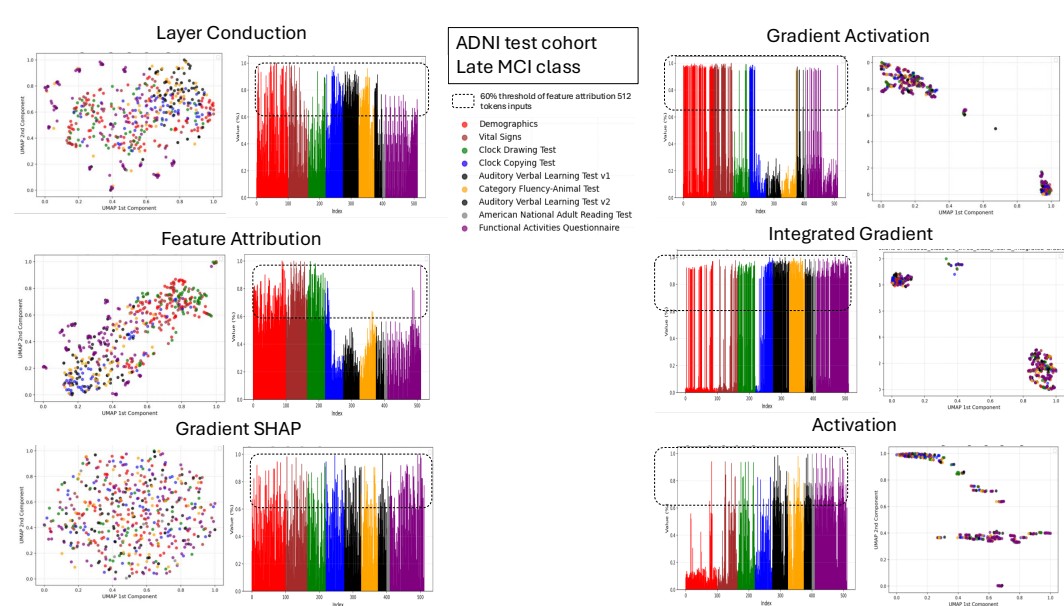

Figure 31: Global (cohort-level) feature attribution across explanation methods without the SAE layer. The 2D panel shows a UMAP embedding (UMAP-1 vs UMAP-2) computed on the ADNI test set; the 1D panel shows attribution scores along PCA first component. All plotted values are normalised to [0, 1] and represent positive contributions only. Colours (red→purple) denote the nine ADNI subgroups (see §B3). Square boxes mark the 0.6–1.0 interval, highlighting the most significant tokens in the 1D views. The task is a three-class classification (LMCI, MCI disease vs Control) on the ADNI cohort; the examples shown here are from the MCI class.

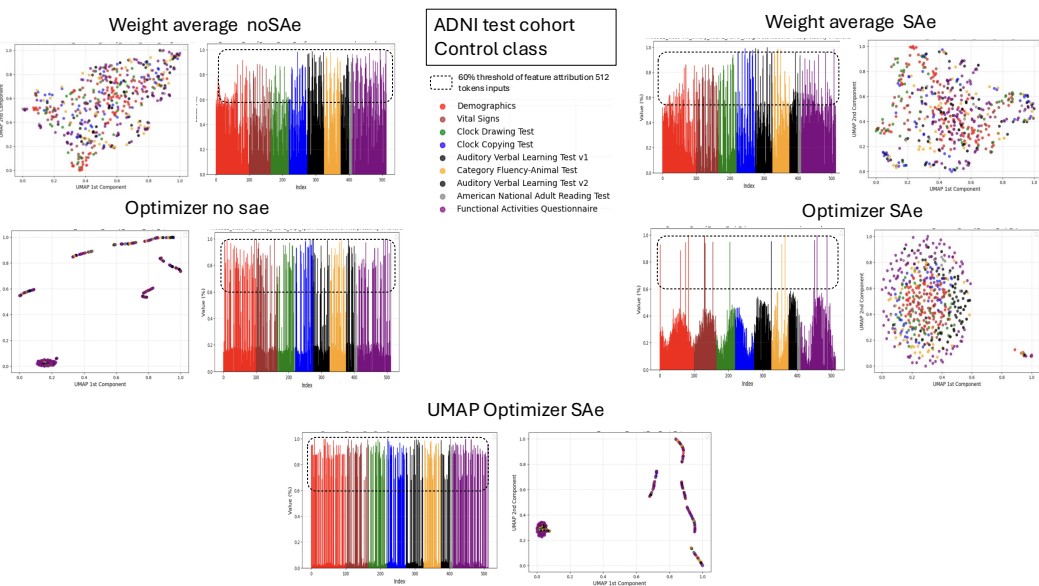

Figure 32: Global (cohort-level) feature attribution across explanation methods with the SAE layer. The 2D panel shows a UMAP embedding (UMAP-1 vs UMAP-2) computed on the ADNI test set; the 1D panel shows attribution scores along PCA first component. All plotted values are normalised to [0, 1] and represent positive contributions only. Colours (red→purple) denote the nine ADNI subgroups (see §B3). Square boxes mark the 0.6–1.0 interval, highlighting the most significant tokens in the 1D views. The task is a binary classification (Alzheimer vs Control) on the ADNI cohort; the examples shown here are from the Control class.

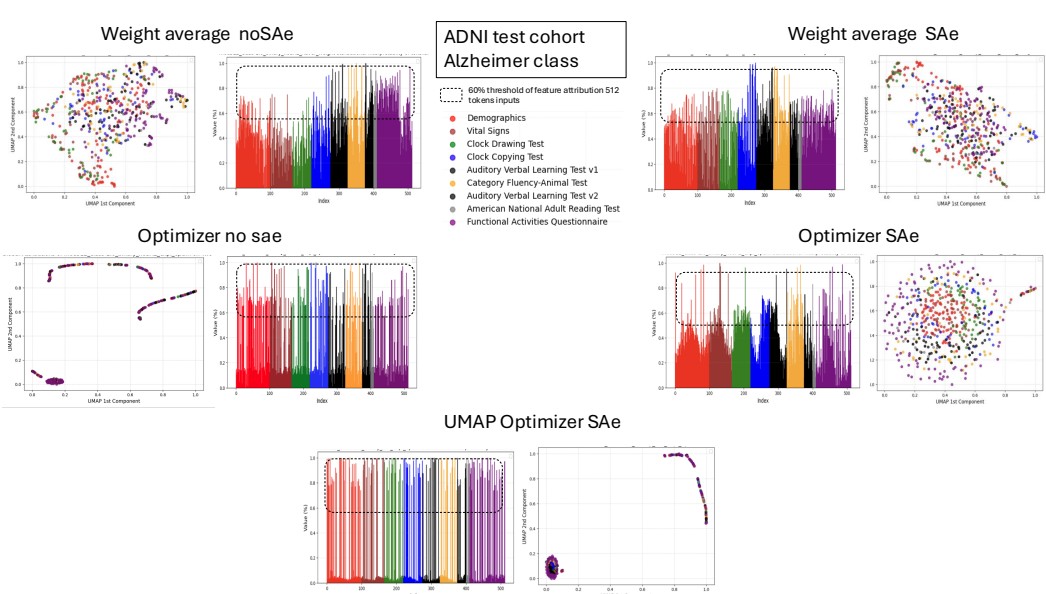

Figure 33: Global (cohort-level) feature attribution across explanation methods with the SAE layer. The 2D panel shows a UMAP embedding (UMAP-1 vs UMAP-2) computed on the ADNI test set; the 1D panel shows attribution scores along PCA first component. All plotted values are normalised to [0, 1] and represent positive contributions only. Colours (red→purple) denote the nine ADNI subgroups (see §B3). Square boxes mark the 0.6–1.0 interval, highlighting the most significant tokens in the 1D views. The task is a binary classification (Alzheimer vs Control) on the ADNI cohort; the examples shown here are from the Alzheimer class.

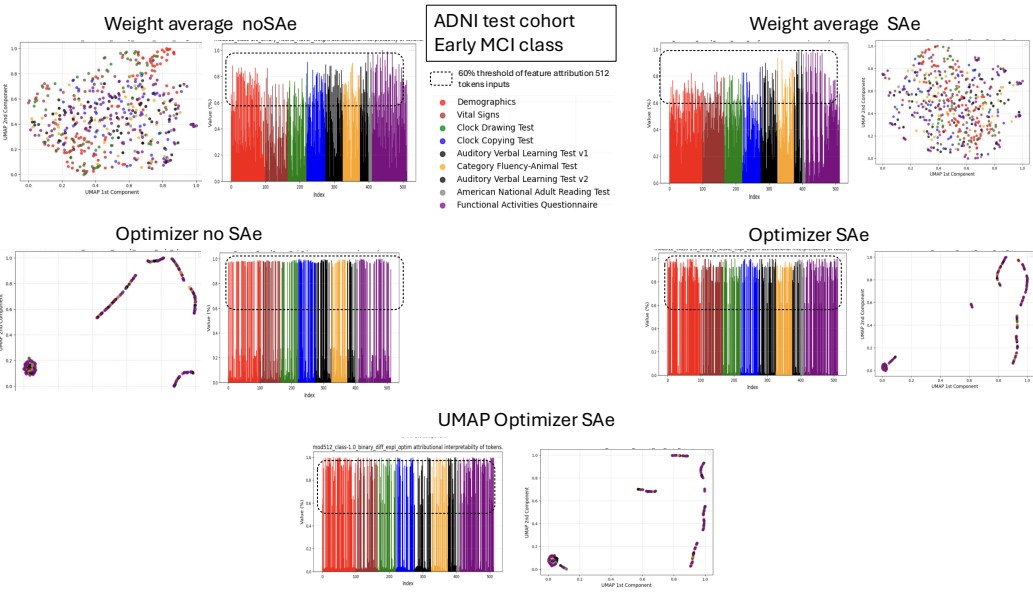

Figure 34: Global (cohort-level) feature attribution across explanation methods without the SAE layer. The 2D panel shows a UMAP embedding (UMAP-1 vs UMAP-2) computed on the ADNI test set; the 1D panel shows attribution scores along PCA first component. All plotted values are normalised to [0, 1] and represent positive contributions only. Colours (red→purple) denote the nine ADNI subgroups (see §B3). Square boxes mark the 0.6–1.0 interval, highlighting the most significant tokens in the 1D views. The task is a three-class classification (LMCI, MCI disease vs Control) on the ADNI cohort; the examples shown here are from the LMCI class.

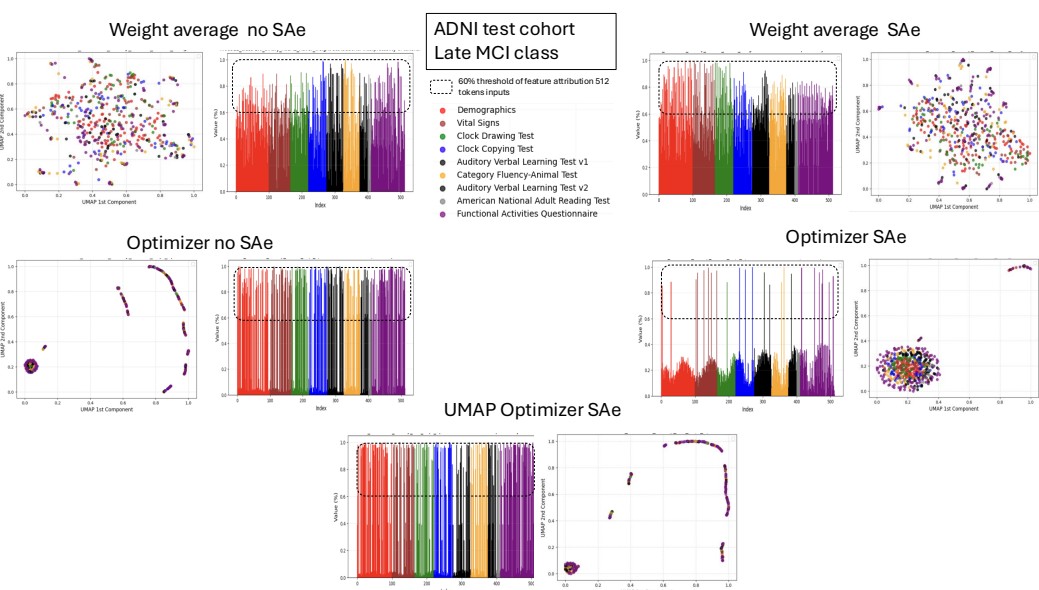

Figure 35: Global (cohort-level) feature attribution across explanation methods with the SAE layer. The 2D panel shows a UMAP embedding (UMAP-1 vs UMAP-2) computed on the ADNI test set; the 1D panel shows attribution scores along PCA first component. All plotted values are normalised to [0, 1] and represent positive contributions only. Colours (red→purple) denote the nine ADNI subgroups (see §B3). Square boxes mark the 0.6–1.0 interval, highlighting the most significant tokens in the 1D views. The task is a three-class classification (LMCI, MCI disease vs Control) on the ADNI cohort; the examples shown here are from the MCI class.

Figures 32–35 present cohort-level attribution examples for both the binary (Control vs Alzheimer's disease) and three-class (Control, LMCI, MCI) classification tasks on the ADNI test cohort, analogous to Figures 22–31, for the no-SAE analyses of (i) the attributional weighted average (computed from the six base methods), (ii) the Transformer Explanation Optimizer (TEO), and (iii) TEO with a linear UMAP constraint (UMAP Optimizer). As shown in the previous subsection, with the SAE layer TEO achieves the best stability—i.e., the lowest RIS and ROS—but at the cost of a marked reduction in Sparseness; this reduction is clearly visible in the binary task (Figures 32–35), where a spreading of tokens in 2D is observed when moving from no-SAE to SAE, as with the other methods. TEO with SAE reorganises the space, yielding a more homogeneous low-to-high attribution gradient. The drawback is that, without appropriate guidance, there may be too few features in the squares denoting significant contribution, and not all subgroups in the global observations are represented (e.g., Figure 32). However, this can be mitigated by constraining the 2D manifold in the attribution space. To that end, we proposed a linear constraint to further smooth the regrouping of tokens in the attribution manifold. Introducing the UMAP linear constraint yields an even more balanced trade-off compared with unconstrained TEO with SAE, producing explanations that share similar significant traits across the different subgroups (colours) and are more homogeneous (very clear in Figures 32, 33, and 35, less so in 34). Consequently, the maps are more compact and clinically interpretable; the same behaviour is observed across all classes in the three-class setting (Figures 33–35). By contrast, at both cohort and local levels, the weighted-average approach—a linear combination of the six attribution techniques—does not yield superior explanations, consistent with Mamalakis et al. (2025).

### B.8 CLINICAL RELEVANCE IN ALZHEIMER'S DIAGNOSIS PROGRESSION: EVIDENCE THAT SAE-GUIDED ATTRIBUTION YIELDS MORE RELIABLE EXPLANATIONS THAN TRADITIONAL ATTRIBUTION.

To further validate the role of the SAE layer in shaping the attribution space into a more monosemantic and clinically coherent feature representation, we conducted an auxiliary evaluation. Specifically, for each test input, we extracted the top 50% most influential token attributions

produced by our attribution framework—TEO without SAE, TEO with SAE, and TEO-UMAP. We then generated a CSV file for each classification class, in which the highlighted characters for the three attribution methods were organized column-wise. The complete character sequence for each sample, beginning with the CLS token, was included in the first column to ensure clear sample-level distinction. These CSV files were subsequently provided as input to a large language model (ChatGPT-5.1 OpenAI (2024)) using a fixed prompt to obtain an external, model-agnostic assessment of the interpretability structure encoded by each explanation space.

Three experiments were performed:

1. **Binary ADNI** (Control vs. Alzheimer's disease), with each class provided as a separate CSV file.

2. **Binary BrainLAT** (Control vs. Alzheimer's disease), also split into two class-specific CSV files.

3. **Three-class ADNI** (Control, MCI, LMCI), with each diagnostic category represented in its own CSV file.

We evaluated two primary criteria: (i) whether the language model could distinguish, based solely on the highlighted features, which CSV corresponded to the pathological versus the healthy control class; and (ii) whether the model could identify meaningful pathology-related biomarkers.

For the first two experiments, the model was prompted with:

> *Given the two CSV files, and recognizing that medical biases exist in the* char *column with each sample beginning with the character sequence [CLS], determine how each of the three attribution methods (attr1, attr2, attr3) highlight features associated with healthy or unhealthy interpretations, and analyze the reasons for these differences. Predict the pathology and specify which of the two CSV files corresponds to the pathological case for each attribution technique.*

For the three-class ADNI experiment, we used:

> *Given the three CSV files, and recognizing that medical biases exist in the* char *column with each sample beginning with the character sequence [CLS], determine how each attribution method (attr1, attr2, attr3) highlights features associated with healthy or pathological interpretations. Predict the pathology or pathologies, identify which CSVs correspond to the pathological and healthy groups for each attribution method, and specify the associated conditions.*

The resulting GPT-generated interpretations, shown in Figures 36, provide an external linguistic lens on each explanation space.

In the binary ADNI experiment, all the three attribution frameworks, TEO without SAE, TEO with SAE, and TEO-UMAP, correctly identified the pathological file (CSV1) and associated it with Alzheimer's disease, noting that the signal was more consistent with late-stage rather than early cognitive impairment. A similar pattern emerged for the binary BrainLAT experiment: the pathological CSV was attributed to Alzheimer's disease rather than MCI, with clear differentiation from the healthy control (Figures 36a,b).

Across both binary tasks, TEO without SAE exhibited erratic and clinically uninformative behavior, frequently attending to task labels, instruction counts, or other artefactual patterns rather than neurocognitive biomarkers (Figures 36a,b). In contrast, TEO-SAE and TEO-UMAP consistently highlighted clinically meaningful domains, including demographic risk factors, processing-speed impairments, and neurophysiological indicators.

In the more complex three-class ADNI experiment, the advantages of the SAE-induced monosemantic structure became even clearer (Figure 36c). TEO without SAE failed to correctly identify pathological classes and did not surface meaningful biomarkers. Conversely, TEO-SAE achieved clearer diagnostic separation and more coherent feature concentrations, while TEO-UMAP further emphasized structured biomarkers, particularly within demographics and vitals, and also provided correct class-level predictions.

| Group | Dem | VS | CDT | CCT | AVLT1 | CFA | AVLT2 | ANART | FAQ |
|-------|-----|-----|-----|-----|-------|-----|-------|-------|-----|
| Control TEO | 0.00 | 0.00 | 0.00 | 0.00 | 0.02 | 0.00 | 0.00 | 0.00 | 0.11 |
| Control TEO-UMAP | 0.33 | 0.21 | 0.29 | 0.29 | 0.12 | 0.30 | 0.32 | 0.30 | 0.36 |
| Alzheimer TEO | 0.05 | 0.00 | 0.09 | 0.01 | 0.12 | 0.12 | 0.00 | 0.00 | 0.13 |
| Alzheimer TEO-UMAP | 0.33 | 0.29 | 0.30 | 0.32 | 0.22 | 0.30 | 0.32 | 0.50 | 0.35 |
| Control TEO | 0.74 | 0.55 | 0.87 | 0.63 | 0.88 | 0.68 | 0.20 | 0.80 | 0.38 |
| Control TEO-UMAP | 0.51 | 0.61 | 0.67 | 0.60 | 0.72 | 0.66 | 0.56 | 0.60 | 0.45 |
| MCI TEO | 0.23 | 0.30 | 0.29 | 0.38 | 0.22 | 0.30 | 0.36 | 0.10 | 0.31 |
| MCI TEO-UMAP | 0.31 | 0.26 | 0.21 | 0.22 | 0.22 | 0.36 | 0.40 | 0.40 | 0.23 |
| LMCI TEO | 0.19 | 0.19 | 0.20 | 0.10 | 0.22 | 0.16 | 0.24 | 0.10 | 0.22 |
| LMCI TEO-UMAP | 0.32 | 0.27 | 0.23 | 0.25 | 0.24 | 0.28 | 0.40 | 0.40 | 0.43 |

Table 6: Abbreviations: Dem = Demographics; VS = Vital Signs; CDT = Clock Drawing Test; CCT = Clock Copying Test; AVLT1/2 = Auditory Verbal Learning Test (v1/v2); CFA = Category Fluency (Animals); ANART = American National Adult Reading Test; FAQ = Functional Activities Questionnaire.

Collectively, these findings demonstrate that incorporating the SAE layer—thereby enforcing a more monosemantic, disentangled attribution representation—substantially enhances the clinical meaningfulness, stability, and diagnostic alignment of the resulting explanations.

### B.9 THE CLINICAL IMPACT AND OUTCOME IN THE DIAGNOSIS OF ALZHEIMER, EARLY MCI AND LATE MCI.

This study shows that the Transformer Explanation Optimizer (TEO) with a Sparse Autoencoder (SAE) and TEO-UMAP provide the most reliable identification of informative sources across nine multimodal subgroups: Demographics (DEM), Vital Signs (VS), Clock Drawing Test (CDT), Clock Copying Test (CCT), Auditory Verbal Learning Test v1 (AVLT1), Category Fluency—Animals (CFA), Auditory Verbal Learning Test v2 (AVLT2), American National Adult Reading Test (ANART), and Functional Activities Questionnaire (FAQ). Using a significance threshold of 0.6 on PCA principal components PC1, we observe in the binary task that, for Control, TEO-SAE is dominated by FAQ, whereas TEO-UMAP emphasises DEM, AVLT2, and FAQ; for Alzheimer's, TEO prioritises FAQ, AVLT1, and CFA, while TEO-UMAP highlights ANART, FAQ, and DEM. In the three-class task, for Control the main contributors are AVLT1, CDT, and ANART under TEO, and AVLT1, CDT, and CFA under TEO-UMAP; for MCI, TEO favours CCT, AVLT2, and FAQ, whereas TEO-UMAP favours AVLT2, ANART, and CFA; and for LMCI, TEO elevates AVLT1, FAQ, and CDT, while TEO-UMAP elevates FAQ, ANART, and AVLT2. These patterns, summarised in Table 6, support the clinical interpretability of the proposed optimisers.

Across ADNI cohorts, the most stable signals for clinical stratification are functional status (FAQ) and memory measures (AVLT1/AVLT2), with visuospatial performance (CDT) recurrent in Control/LMCI. TEO+SAE preferentially elevates neuropsychological performance features (AVLT1/2, CDT, CCT), while TEO-UMAP surfaces complementary contextual/language markers (DEM, ANART, CFA), yielding class-specific, interpretable profiles: Control—FAQ/AVLT1/CDT; Alzheimer's—FAQ with AVLT1/CFA (TEO) or ANART/DEM (TEO-UMAP); MCI—AVLT2 with CCT/FAQ (TEO) or ANART/CFA (TEO-UMAP); LMCI—FAQ with AVLT1/CDT (TEO) or ANART/AVLT2 (TEO-UMAP). Using a simple PC1 0.6 significance rule, these optimisers provide actionable attribution maps that can prioritise assessments, reduce testing burden, support trial enrichment, and guide personalised monitoring. Together, they offer a practically deployable, transparent framework for clinically meaningful multimodal reasoning in neurodegenerative disease.

**Binary ADNI Control (0)**                    **Binary ADNI Alzheimer (1)**

Given the two CSV files, and recognizing that medical biases exist in the char column with each of samples beginning with the character sequence [C L S], determine how each of the three attribution methods (attr1, attr2, attr3) highlights features that may be associated with healthy or unhealthy interpretations, and analyze the reasons for these differences. Try to predict the pathology. Mention which of the two csv files highlight the pathological case for each of the three attr techniques and try to specify the pathology.

| | Method | What It Highlights Most | Which CSV Shows the Pathological Pattern | Why This CSV Looks Pathological | Likely Pathology Suggested |
|---|---|---|---|---|---|
| TEO-SAE | attr1 | Score terms (Total, Intrusions, Correct), demographic fields (Birth year, education, Race), test-related error wording | CSV 1 | Stronger emphasis on demographic risk indicators and on error-related cognitive test features | Amnestic cognitive disorder (likely early Alzheimer's / amnestic MCI) |
| TEO-No-SAE | attr2 | Task names, specific scores, intrusion counts, slow Trails B areas | CSV 1 | More attention to the worst-performing cognitive segments and to demographic correlates | Amnestic cognitive disorder (likely early Alzheimer's / amnestic MCI) |
| TEO-UMAP | attr3 | Whole neuropsych blocks (Clock, AVLT, Trails), demographic segments, sections with intrusions or slow processing | CSV 1 | Highlights both demographic risk factors and impaired memory/executive performance more strongly | Amnestic cognitive disorder (likely early Alzheimer's / amnestic MCI) |

(a) Two class ADNI classification experiment

**Binary BrainLAT Control (BL_0)**                    **Binary BrainLAT Alzheimer (BL_1)**

Given the two CSV files, and recognizing that medical biases exist in the char column with each of samples beginning with the character sequence [C L S], determine how each of the three attribution methods (attr1, attr2, attr3) highlights features that may be associated with healthy or unhealthy interpretations, and analyze the reasons for these differences. Try to predict the pathology. Mention which of the two csv files highlight the pathological case for each of the three attr techniques and try to specify the pathology.

| | Method | What It Highlights Most | Pathological CSV | Why This CSV Appears Pathological | Suggested Pathology |
|---|---|---|---|---|---|
| TEO-SAE | attr1 | Notable emphasis on key cognitive-test terms, scoring markers, and error-related text segments | BL_1 | Shows greater sensitivity to segments containing performance weakness indicators and demographic risk markers | Memory-dominant cognitive deficit (amnestic MCI / early Alzheimer's) |
| TEO-No-SAE | attr2 | Focus on specific task labels and parts of the text associated with reduced performance or irregular responses | BL_1 | Highlights sections where performance is most impaired, such as memory intrusions or slowed sequences | Amnestic pattern consistent with early neurodegenerative disease |
| TEO-UMAP | attr3 | Broad blocks of neuropsych content, especially where errors, slow timing, or intrusions appear | BL_1 | Shows widespread emphasis across impaired-performance regions and demographic variables | Early Alzheimer's-type cognitive impairment |

(b) Two class BrainLAT classification experiment

**Three class ADNI Control (0)**          **Three class ADNI MCI (1)**          **Three class ADNI LMCI (2)**

Given the three CSV files, and recognizing that medical biases exist in the char column with each of samples beginning with the character sequence [C L S], determine how each of the three attribution methods (attr1, attr2, attr3) highlights features that may be associated with healthy or unhealthy interpretations, and analyze the reasons for these differences. Try to predict the pathology or pathologies and the healthy if exist. Mention which of the three csv files highlight the pathological case/s for each of the three attr techniques and try to specify the pathology/ies and the same for the healthy (control if exist).

| | Attribution Pattern Type | Behavior Observed in CSV File 0 | Behavior Observed in CSV File 1 | Behavior Observed in CSV File 2 | Interpretation (Pathology / Healthy) |
|---|---|---|---|---|---|
| TEO-SAE | Pattern Type A (very sparse, near-zero emphasis overall) | Minimal highlighting of any medically relevant fields; almost all characters left unmarked. | Slight but still very limited emphasis on demographic fields and test labels. | Noticeably more highlighting on demographics, test headers, and structural elements of the note compared to CSV 0 and 1. | CSV 0 corresponds best to **healthy / control class**. CSV 1 corresponds to **milder pathology**, CSV 2 to **more severe pathology**. |
| TEO-No-SAE | Pattern Type B (extremely flat, low-information pattern) | Very low number of highlighted characters; almost uniformly neutral. | Similarly low highlighting; no meaningful differentiation. | Same low-density pattern; virtually no medically decisive areas marked. | This pattern does **not meaningfully distinguish** healthy vs pathological classes across any CSV files. |
| TEO-UMAP | Pattern Type C (dense, diffuse, broad coverage) | Broad but slightly lower coverage of demographic fields, vitals, and test labels. | Broad coverage similar to CSV 2 but with slightly less emphasis. | Highest density: demographics, vitals, and test labels consistently highlighted. | CSV 0 again aligns with **healthy / control**. CSV 1 and CSV 2 align with **pathology classes**, with CSV 2 indicating **more severe pathology**. |

(c) Three class ADNI classification experiment.

Figure 36: GPT-5.1 generated global explanations characterizing attribution-score performance in biomarker identification for TEO, both with and without the SAE layer, and for the TEO-UMAP model. (a) Binary ADNI: Control vs. Alzheimer's disease, with each class provided as a separate CSV file for external evaluation. (b) Binary BrainLAT: Control vs. Alzheimer's disease, similarly split into two class-specific CSV files. (c) Three-class ADNI: Control, MCI, and LMCI categories, each represented in its own CSV file.

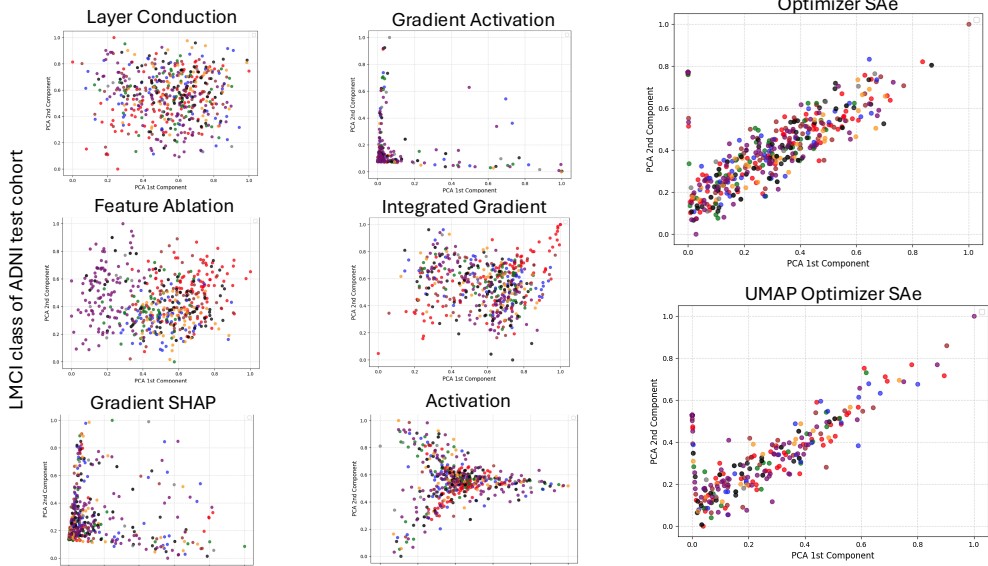

Figure 37: Principal Component Analysis (PCA) of token-level attribution representations across methods. Each scatter plot shows the first two principal components extracted from the top eight PCA components of the attribution matrix for the LMCI class of the ADNI test cohort. PCA is used here to assess whether the proposed UMAP linear constraint encourages the attribution space to adopt an approximately linear structure. Traditional attribution methods (Layer Conduction, Feature Ablation, Gradient SHAP, Gradient Activation, Integrated Gradients, and Activation) exhibit more dispersed or irregular PCA trajectories, indicating nonlinear or highly variable attribution topologies. In contrast, the Optimizer SAE and UMAP Optimizer SAE methods show a clear linear trend along the first two principal components, demonstrating that the constrained UMAP formulation produces a substantially more linear and stable representation of attribution scores across the shared tokenizer feature space. These results support the hypothesis that enforcing a linear constraint in the UMAP embedding enhances structural consistency and robustness of feature-level attributions.

### B.10 VALIDATION OF THE UMAP LINEAR CONSTRAINT VIA PCA STRUCTURE ANALYSIS

To verify the claim that the proposed UMAP linear constraint effectively linearizes the majority of the attribution space and yields robust attribution scores within the same tokenized feature, we performed an additional PCA analysis. Specifically, we extracted the top eight principal components from the attribution matrix and visualized the first two components, which capture the highest proportion of variance in the data. This allows us to assess whether the tokenized features exhibit an approximately linear structure in their dominant statistical directions. The results (Figure 37) demonstrate that the proposed method indeed produces an embedding that approaches a linear configuration, thereby supporting our hypothesis that the UMAP linear constraint leads to more stable and structurally consistent attribution representations.

### B.11 BROAD IMPACT STATEMENT

The clinical deployment of large language models (LLMs) in high-stakes neurodegenerative disease diagnosis, such as Alzheimer's Disease (AD), is hindered by the inherent polysemanticity of their representations, which renders traditional attribution methods (e.g., gradients, SHAP) unreliable due to ambiguous or inconsistent explanations. By aligning LLM explanations with clinical reasoning and enforcing statistical fidelity, this work establishes a foundation for trustworthy, deployable AI systems in medicine, transforming complex models into transparent partners for life-critical decision-making and paving the way for safer, ethically sound integration of advanced AI into cognitive health applications. Critically, this framework's adaptability and rigorous validation position it for immediate real-world deployment in healthcare settings, enabling clinicians to

harness LLMs' diagnostic power without compromising transparency, thereby accelerating the translation of AI research into measurable improvements in patient care.