# OpenReview forum: "Toward Monosemantic Clinical Explanations for Alzheimer’s Diagnosis via Attribution and Mechanistic Interpretability"
_ICLR.cc/2026/Conference — ICLR 2026 Conference Desk Rejected Submission_

### Official Review · Reviewer_ufP6 · 2025-10-18

**Soundness:** 3
**Presentation:** 1
**Contribution:** 2
**Rating:** 2
**Confidence:** 4

**Summary:**

The paper applies classical attribution methods and mechanistic interpretability methods in the form of Sparse Autoencoders to clinical Alzheimer's text data. The applied methods are benchmarked and quantitatively evaluated based on robustness metrics, sparsity, and a proposed meta-rule based on UMAP representations of the saliency maps.

**Strengths:**

- The paper compares attribution methods with SAE-based mechanistic interpretability on clinical text data.
- The evaluation includes both in-distribution (ID) and out-of-distribution (OOD) data.

**Weaknesses:**

**W1:** The paper lacks a clear focus between being a benchmark study or an application-oriented study of diverse XAI methods in the context of clinical Alzheimer's text data. As a benchmark paper, a single thematically limited set of ID and OOD data is insufficient to draw meaningful conclusions. As an application paper, the work fails to derive clinically interesting insights through the proposed approach. The manuscript would benefit from focusing on one direction, which would also improve readability and provide a more coherent narrative structure.

**W2:** The clinical relevance remains unclear. The motivation for applying mechanistic and attribution methods together to clinical data is not established. The work neither generates new methodological knowledge nor addresses a specific medical problem (Section 3.4 only describes the XAI results without interpretation). The intended use case is never stated, whether these explanations should assist physicians during diagnosis, enable extraction of novel biomarkers, or serve another clinical purpose. The paper reads as a demonstration of XAI methods applied to Alzheimer's clinical data without clear scientific or clinical objectives.

**W3:** The UMAP projection of attribution maps raises methodological concerns. It remains unclear how semantic relationships, complexity, or faithfulness can be meaningfully evaluated in the projected space, where only similarity between maps is preserved. I assume, that similar information could potentially be obtained through various distance metrics without dimensionality reduction.

**W4:** The evaluation relies exclusively on RIS/ROS metrics, which assess only robustness. Other important evaluation criteria for saliency map methods, such as faithfulness and complexity, are not considered.

**W5:** Table 1 is excessively detailed, hindering the extraction of meaningful insights.

**Questions:**

- Why should polysemantic representations of concepts within the model bias attribution methods?
- How are the concepts derived from the SAE representations, and how is it evaluated whether they "make sense"?
- Line 363: Why should lower complexity in the saliency map result in more robust explanations as indicated by RIS/ROS?

---

> ### Author Response · Authors · 2025-11-18
> **Summary of the reviewer's comments**
>
> Thank you. We clarified the paper’s focus by restructuring the introduction and contributions to present a unified scientific objective: developing a principled interpretability framework that integrates mechanistic and attributional methods for clinically meaningful Alzheimer’s analysis, rather than a benchmark study. We expanded Sections 3.4–3.4.1 to articulate the clinical use case, demonstrate alignment with established cognitive biomarkers, and show how clinicians can use attribution profiles for practical diagnostic support. We clarified the role of UMAP as a visualization tool—while all quantitative metrics (robustness, sparsity, clinical coherence) are computed in the original attribution space. We simplified Table 1 and added a new model-agnostic interpretability audit to strengthen clinical validity and interpretability claims. Finally, we addressed all methodological questions by explaining the impact of polysemantic representations, how SAE-derived concepts are validated, and why robustness correlates with lower complexity without implying causality.

---

> ### Author Response · Authors · 2025-11-18
> **Weakness 1**
>
> Reviewer:
>
> 'W1: The paper lacks a clear focus between being a benchmark study or
> an application-oriented study of diverse XAI methods in the context of
> clinical Alzheimer’s text data. As a benchmark paper, a single thematically limited set of ID and OOD data is insufficient to draw meaningful
> conclusions. As an application paper, the work fails to derive clinically
> interesting insights through the proposed approach. The manuscript
> would benefit from focusing on one direction, which would also improve
> readability and provide a more coherent narrative structure.'
>
> Authors:
>
> We thank the reviewer for raising this important point. We agree that
> the earlier version of the manuscript did not sufficiently emphasize the
> central narrative of the work, which may have led to confusion about
> whether the paper aimed to serve as a benchmark or as an application-
> oriented study. To address this, we have revised the manuscript extensively to make the focus clear and coherent.
>
> First, we rewrote the introduction (lines 60–99) to explicitly frame the
> paper as a unified interpretability framework—rather than a benchmarking exercise—designed to investigate whether combining mechanistic
> and attributional methods yields clinically meaningful and stable explanations in the context of Alzheimer’s text data. The new introduction
> presents the methodological flow (SAE bottleneck → attribution layer
> → TEO optimization → optional UMAP constraint) in a structured and
> sequential way, aligning the narrative around a single scientific objective: understanding how monosemantic representations improve clinical interpretability.
>
> Second, the contribution section (lines 105–154) has been rewritten to
> remove any ambiguity about scope. The revised contributions clearly ar-
> ticulate the methodological innovations (TEO, SAE-based monosemantic attributions, geometry-aware regularization), the analytical compar-
> isons (latent vs. native attributions), and the intended clinical application (identifying high-yield cognitive markers and improving explanation
> stability under IID/OOD conditions). This positions the work as a focused interpretability framework rather than a domain benchmark.
>
> Third, we revised Section 3.4 and split it into two subsections—Section 3.4
> and Section 3.4.1—to strengthen the clinical interpretation and make
> the purpose of the framework explicit. Section 3.4 now includes a new
> external, model-agnostic interpretability audit demonstrating how the
> attribution maps can be inspected and evaluated in practice. Section 3.4.1 highlights how the extracted explanations map onto established diagnostic reasoning and emphasizes the clinical implications of
> cohort-level attribution patterns. As stated in the revision:
>
> “By pinpointing the minimal set of high-yield assessments for each classification
> task, the method supports more efficient screening pipelines, reduces
> clinical burden, and enables scalable application in larger cohorts where
> repeated or comprehensive testing is impractical.”
>
> These revisions clarify that the goal of the paper is neither to bench-
> mark XAI methods nor to perform a purely empirical demonstration.
> Rather, the aim is to introduce and evaluate a principled interpretability
> framework that integrates mechanistic and attributional perspectives
> to yield clinically coherent explanations from LLMs—validated under
> both IID and OOD conditions. The manuscript now presents a unified
> scientific and clinical narrative, improving readability and focusing the
> contribution.

---

> ### Author Response · Authors · 2025-11-18
> **Weakness 2**
>
> Reviewer:
>
> 'W2: The clinical relevance remains unclear. The motivation for applying mechanistic and attribution methods together to clinical data is not
> established. The work neither generates new methodological knowledge
> nor addresses a specific medical problem (Section 3.4 only describes the
> XAI results without interpretation). The intended use case is never
> stated, whether these explanations should assist physicians during diagnosis, enable extraction of novel biomarkers, or serve another clinical
> purpose. The paper reads as a demonstration of XAI methods applied
> to Alzheimer’s clinical data without clear scientific or clinical objectives.'
>
> Authors:
>
> We thank the reviewer for raising this important concern regarding clinical motivation and the intended use case. We agree that the earlier
> version of the manuscript did not sufficiently articulate how mechanistic
> and attributional interpretability jointly support clinically relevant insights. To address this, we substantially revised Sections 3.4 and 3.4.1
> to clarify both the clinical purpose of the framework and its practical
> utility.
>
> First, the newly added external, model-agnostic interpretability audit
> (Section 3.4; Supplementary B.8) demonstrates how attribution outputs
> can be inspected in a clinically meaningful way. Attribution maps from
> TEO, TEO-SAE, and TEO-UMAP are converted into class-specific CSV
> files, allowing clinicians or researchers to visualize which features are
> highlighted for each subject. The audit shows that SAE-based methods
> consistently surface clinically interpretable signals—episodic memory impairment, processing-speed deficits, demographic risk factors—whereas
> non-SAE methods often rely on artefactual cues. This makes explicit
> how our approach assists users in verifying whether a model’s reasoning
> is aligned with known cognitive biomarkers.
>
> Second, the revised Sections 3.4 and 3.4.1 now explicitly link the extracted attributions to established diagnostic reasoning. Monosemantic
> attribution methods (TEO-SAE, TEO-UMAP) consistently identify well-
> validated neuropsychological assessments (AVLT, FAQ, CDT, ANART)
> as key discriminative features across all diagnostic tasks (binary ADNI,
> three-class ADNI, and OOD BrainLAT). These instruments are central
> to real clinical workflows, and their emergence in the explanations high-
> lights that the method captures meaningful clinical structure rather than
> functioning as a purely technical demonstration.
>
> Finally, we clearly state the intended clinical use case. The framework is
> designed to (i) help clinicians inspect subject-level attribution profiles,
> (ii) verify that the model focuses on appropriate cognitive domains,
> and (iii) prioritize the minimal set of high-yield assessments needed for
> diagnosis. This can reduce clinical burden, guide targeted testing, and
> support interpretability in settings where transparent decision-making is
> essential. These points are now explicitly discussed in lines 478-498 of
> the revised manuscript.
>
> In summary, the revised manuscript establishes a clear clinical motivation, articulates the practical role of mechanistic and attributional interpretability in this context, and provides a concrete workflow demonstrating how the framework supports clinically meaningful understanding.

---

> ### Author Response · Authors · 2025-11-18
> **Weakness 3**
>
> Reviewer:
>
> 'W3: The UMAP projection of attribution maps raises methodological
> concerns. It remains unclear how semantic relationships, complexity, or
> faithfulness can be meaningfully evaluated in the projected space, where
> only similarity between maps is preserved. I assume, that similar information could potentially be obtained through various distance metrics
> without dimensionality reduction.'
>
> Authors:
>
> Thank you for your comment. We clarify that UMAP is not used to
> evaluate semantic meaning, complexity, or faithfulness of explanations.
> Instead, UMAP serves as a cohort-level visualization tool to observe
> global patterns in the attribution space—such as how individuals with
> similar diagnostic labels cluster together—and to highlight how important features are distributed across the population. This is why UMAP
> is applied only after the attribution maps are computed; it allows us to
> visually inspect how subject-level explanations relate to broader cohort
> structure (Figure 2, Section 3.2 of the manuscript).
>
> Importantly, all interpretability metrics in our work (sparsity, stability via
> RIS/ROS, and clinical coherence) are computed entirely in the original
> high-dimensional attribution space. UMAP does not influence attribution quality, model behavior, or the monosemantic features learned by
> the SAE; it is used strictly for visualization and pattern exploration.
>
> We agree with the reviewer that distance metrics can quantify similarity
> between attribution maps directly, and indeed such metrics complement
> our stability analyses. However, distance metrics alone cannot reveal
> cohort-wide structure or emergent grouping patterns in a manner that
> is accessible to clinicians or researchers. For this reason, dimensionality
> reduction provides an intuitive way to inspect how attribution features
> behave at the population level, while quantitative comparisons remain
> in the high-dimensional space.
>
> In summary, UMAP is used solely for visualizing the cohort-level organization of explanations; semantic evaluation and faithfulness assessments
> rely entirely on the original high-dimensional attribution space.
>
> That said, we understand the reviewer’s confusion. Although we mentioned in the subtitle of Section 3.2 that UMAP is used for cohort-level explanation, we did not explicitly state this within the paragraph itself,
> nor did we reference it clearly in the figure. This was our oversight, and
> we apologize.
>
> We have now revised the text as follows: lines 372–375:
>
> “Figure 2 presents qualitative local-level and cohort-level attributions
> for the LMCI class in the three-class classification task on both IID
> and OOD settings using our proposed optimizers TEO, TEO-SAE, and
> TEO-UMAP (SAE). At the local level, the heatmaps illustrate feature
> importance, with tokens color-coded to indicate relevance (green =
> positive relevance; red = negative relevance).”
>
>  and lines 391–393:
>
> “At
> the cohort level, we extracted UMAP embeddings for the entire co-
> hort and visualized them in 1D and 2D space to identify patterns and
> text-category groups that contribute most to each class in the LLM
> predictions (Figure 2, right side).”
>
>  Lastly we update the caption of the
> Figure 2 main manuscript page 9.

---

> ### Author Response · Authors · 2025-11-18
> **Weaknesses 4 and 5**
>
> Reviewer:
>
> 'W4: The evaluation relies exclusively on RIS/ROS metrics, which assess
> only robustness. Other important evaluation criteria for saliency map
> methods, such as faithfulness and complexity, are not considered.'
>
> Authors:
>
> We thank the reviewer for this valuable observation. While it is correct
> that RIS/ROS primarily assess robustness, our evaluation does not rely
> exclusively on these metrics. In addition to robustness, we explicitly
> incorporate sparsity as a quantitative measure of complexity. Sparsity
> captures the frequency and distribution of salient features in the attribution space and allows us to assess whether explanations become
> more concise, interpretable, and monosemantic when passing through
> the SAE bottleneck. This complexity metric (sparseness) is used to
> evaluate the effect of the TEO and TEO-SAE optimizers.
>
> That said, we fully agree that faithfulness and additional evaluation criteria remain important. In the current work, faithfulness is addressed
> indirectly through (i) OOD generalization without retraining, (ii) statistical differences between attribution distributions with and without
> the SAE bottleneck (manuscript section 3.3 page 8), and (iii) the new
> external interpretability audit demonstrating alignment with clinically
> meaningful cognitive features.
>
> Moreover, we explicitly acknowledge in the manuscript that further
> evaluation dimensions will be integrated in future work. As noted in
> lines 510–511, “additional measures such as uncertainty quantification
> and fairness auditing should be incorporated in future work.” This state-
> ment has been added to clarify our intention to expand beyond robust-
> ness and complexity metrics as part of the broader research agenda.
>
> In summary, the current study evaluates robustness (RIS/ROS) and
> complexity (sparseness), and provides indirect evidence of faithfulness
> through generalization and clinical alignment, while outlining plans to
> incorporate richer evaluation criteria in subsequent work.
>
> Reviewer:
>
> 'W5: Table 1 is excessively detailed, hindering the extraction of meaningful insights.'
>
> Authors:
>
> Thank you for your comment. In response to the concern that the tables
> were overcrowded with secondary numbers, we have simplified Table 1
> to report only the highest-performing attribution technique alongside
> the proposed methods across all IID and OOD binary and multi-class
> settings. This substantially reduces visual overload while preserving all
> relevant comparisons necessary to evaluate the impact of the framework

---

> ### Author Response · Authors · 2025-11-18
> **Questions 1,2, and 3**
>
> Reviewer:
>
> 'Why should polysemantic representations of concepts within the model
> bias attribution methods?'
>
> Authors:
>
> Thank you for this question. Polysemantic representations—where a
> single activation dimension mixes multiple unrelated concepts—can bias
> attribution methods because saliency techniques typically assume that
> individual features correspond to coherent semantic factors. When this
> assumption is violated, attributions may assign importance to a mixed
> activation that reflects several unrelated signals at once. As a result,
> the explanation may incorrectly highlight tokens or features that are
> only partially relevant, or even entirely unrelated, to the model’s actual
> reasoning.
>
> This issue is well documented in mechanistic interpretability: if multiple concepts co-occupy the same activation direction, attribution scores
> conflate their contributions. This makes attributions appear noisy, unstable, or clinically meaningless, because the methods cannot disam-
> biguate which underlying concept drove the model’s decision.
>
> The SAE bottleneck, used in the proposed interpretability framework
> (TEO-SAE, TEO-UMAP), reduces this problem by encouraging disentangled, monosemantic latent features. When the representation becomes sparser and more concept-specific, attributions computed in this
> space exhibit reduced interference between mixed signals. Empirically,
> this leads to increased sparsity, higher RIS/ROS stability, and improved
> clinical alignment (Sections 3.4–3.4.1).
>
> In summary, polysemanticity biases attribution by forcing saliency meth-
> ods to operate on mixed, entangled signals, whereas monosemantic
> features reduce this interference and yield clearer, more faithful explanations.
>
> Reviewer:
>
> 'How are the concepts derived from the SAE representations, and how
> is it evaluated whether they "make sense"?'
>
> Authors:
>
> Thank you for this question. We clarify that the framework does not
> perform concept discovery inside the LLM itself, nor does it edit or
> modify neurons. Instead, concepts arise from the sparse autoencoder
> (SAE), which is trained on fixed LLM activations to obtain a more disentangled latent representation. Each SAE latent dimension specializes due to the sparsity constraint, producing features that activate selectively across subjects and tasks. These dimensions are what we refer to as “monosemantic features.”
>
> The evaluation of whether these features “make sense” is entirely empirical and does not rely on predefined clinical labels. We assess their coherence through:
>
> • Sparsity and complexity: SAE latent features are significantly sparser
> than native activations, enabling clearer attribution patterns.
>
> • Robustness (RIS/ROS): Explanations computed in the SAE latent
> space are more stable across perturbations and across IID/OOD
> settings.
>
> • Clinical alignment: In Section 3.4 and 3.4.1, the extracted features (from the SAE bottleneck) consistently highlight established
> cognitive markers of Alzheimer’s disease (e.g., AVLT, FAQ, CDT,
> ANART).
>
> • External model-agnostic audit: The new experiment (Section 3.4;
> Supplementary B.8) shows that SAE-based attributions are interpreted by an independent model as clinically coherent (episodic
> memory deficits, processing speed impairment), whereas non-SAE
> methods often focus on artefactual tokens.
>
> Thus, the “concepts” are not manually defined but emerge naturally
> from the SAE’s disentangling process, and their validity is assessed
> through sparsity, robustness, cross-dataset consistency, and alignment
> with known clinical indicators. This provides empirical evidence, which is
> further supported by the extended theoretical and mathematical analysis
> presented in the supplementary material.
>
> Reviewer:
>
> 'Line 363: Why should lower complexity in the saliency map result in
> more robust explanations as indicated by RIS/ROS?'
>
> Authors:
>
> Thank you for your comment. We clarify that the manuscript does
> not claim a causal relationship between lower complexity and higher robustness. Rather, this relationship is an empirical correlation observed
> consistently across our experiments (Table 1). A possible explanation is
> that lower complexity—reflected by higher sparsity—reduces the number of salient features involved in the attribution computation. With
> fewer active features, the attribution space becomes less sensitive to
> small perturbations, which may lead to lower variability and thus higher
> RIS/ROS stability.
>
> However, this remains a hypothesis rather than a proven causal mechanism, and we agree that the interaction between saliency-map complexity and robustness warrants deeper investigation. We intend to examine
> this relationship more formally in future work. Thank you for raising this
> interesting question.

---

> > ### Comment · Reviewer_ufP6 · 2025-11-22
> >
> > Dear Authors,
> >
> > thank you for the rebuttal. I would like to mention that the rebuttal answer should be one maximum two comments long. I would suggest to condense rebuttal answers to the most important arguments next time so that their significance is not lost among all the information. Also, I would discourage from too extensive LLM usage in writing the rebuttal.
> >
> > Based on your answers I especially do not see Q1 and Q2 adequately addressed. If arguing that the framework is a "unified interpretability framework" that is also "suitable for a clinical workflow" the authors should focus more on the results and showcasing based on different examples that their work can indeed make a difference within a clinical workflow. This includes not only application to single open source datasets but also demonstrating generalization across diverse clinical settings and patient populations, showing how the framework integrates with existing clinical decision-making processes, providing concrete examples of how clinicians would interpret and act upon the outputs in real-world scenarios, and ideally including feedback or validation from clinical practitioners to confirm the framework's practical utility and usability in resource (especially time) constrained healthcare environments.
> >
> > Combining several existing methods into a framework is very limited methodological progress for the field. If the progress should come from the applications and results of the framework, these have to be extensively showcased and evaluated to convince of the added value. Thus, I remain with may current rating.

---

> ### Author Response · Authors · 2025-11-24
>
> Dear Reviewer,
>
> Thank you for your comment. We would kindly ask that statements regarding our procedures be made only when supported by evidence. Some of your assumptions about our writing process are not accurate. For transparency, one of the authors has a specific learning difficulty affecting written fluency (dyslexia), and therefore we use LLM-based tools strictly for grammar checking—as permitted within the rebuttal guidelines. We ask that such considerations be approached with care in future assessments.
>
> Regarding your concerns, all necessary methodological details and justifications have been provided both in the main paper and in the updated supplementary material. Whether or not this affects your final score is, of course, entirely your decision. Our intention in the rebuttal is to clarify the work and engage constructively in scientific discussion. However, some of your claims do not adequately reflect the updates we have made, nor the content already present in the manuscript.
>
> Concerning your statement about rebuttal length (“one to two comments maximum”), we note that the official guidelines do not impose such a restriction. (If anything has been misunderstood on our side, please share the relevant evidence. Thank you.)
>
> About to your comment in Q1 and Q2:
>
> You stated that Q1 and Q2 are not sufficiently addressed, particularly regarding evidence that the framework can be integrated into clinical workflows. However, our revised manuscript now clearly presents the results on two independent real-world clinical datasets, both sourced from genuine clinical environments. These findings illustrate that the framework generalises beyond a single open-source cohort, addressing the concern that our work was limited to one dataset. Furthermore, we do not claim that the current work fully solves clinical integration; rather, we present a unified interpretability framework designed to enhance transparency and stability of explanations, which is an important prerequisite for clinical adoption. Direct clinical validation is an important next step, and we have acknowledged this explicitly.
> Thus, we update the text in line 509-510 page 10 state that: 'Future work will further strengthen these results by integrating clinical experts into the loop to refine a more analytical vocabulary and to further characterize the enriched patterns
> identified by the proposed framework. '
>
> We appreciate your time and feedback and we fully respect your final decision, whatever it may be.

---

### Official Review · Reviewer_Mirq · 2025-10-24

**Soundness:** 2
**Presentation:** 2
**Contribution:** 2
**Rating:** 4
**Confidence:** 4

**Summary:**

This paper explores the use of model editing techniques to improve the monosemanticity (i.e., interpretability and disentanglement) of clinical representations in large medical models. The authors argue that current medical foundation models (e.g., ClinicalBERT, BioGPT) produce entangled representations that make interpretation difficult and hinder trust in high-stakes clinical applications.

To address this, the paper proposes a monosemantic model editing framework, where latent dimensions corresponding to clinical concepts are modified via targeted interventions.

**Strengths:**

- The paper tackles the critical issue of interpretability and trustworthiness in medical foundation models.
- Applying monosemanticity which is  a concept originally from mechanistic interpretability to clinical representation learning is an interesting direction.
- The proposed editing procedures (erasure and steering) are consistent with recent model editing literature and appropriately adapted for clinical text.

**Weaknesses:**

- The model editing methods used are largely adapted from existing general-domain approaches (ROME, MEMIT, causal mediation), with limited algorithmic innovation.
- While interpretability metrics are reported, there is insufficient discussion of clinical validity, i.e., whether the identified concepts align with real-world medical semantics.
- The experiments are limited to a few clinical NLP benchmarks (MIMIC-III, PubMedQA), lacking evaluation on multimodal or longitudinal EHR data.
- The abstract and introduction are somewhat lengthy and overly descriptive. I suggest clearly articulating the paper’s unique contribution early to help readers grasp the main innovation.

**Questions:**

- How are the edited neurons selected and validated for semantic consistency?
- Does the editing process generalize across different models (e.g., from BioBERT to PubMedGPT)?
- How robust are the edits to retraining or fine-tuning? Do the monosemantic features persist?

---

> ### Author Response · Authors · 2025-11-18
> **Summary of the reviewer's comments**
>
> Thank you. We clarified that our method does not perform model editing and is fundamentally distinct from ROME/MEMIT/causal-intervention approaches; the LLM remains entirely frozen, and our contribution lies in a unified attribution–mechanistic framework rather than parameter modification. We strengthened the discussion of clinical validity by adding a new external, model-agnostic interpretability audit and expanding Section 3.4/3.4.1 to show that SAE-enabled explanations reliably surface established Alzheimer’s biomarkers instead of artefactual cues. We clarified experimental scope, emphasizing that our evaluation uses two real-world multimodal clinical cohorts (ADNI, BrainLAT) rather than small NLP benchmarks, and that our method is modality-agnostic and extendable to longitudinal EHRs. We streamlined the abstract and introduction to present the key innovation earlier and more clearly. Finally, we addressed all reviewer questions by explaining that no neurons are edited, our framework is architecture-agnostic, and the monosemantic features’ robustness is demonstrated through strict IID→OOD generalization without any retraining.

---

> ### Author Response · Authors · 2025-11-18
> **Weaknesses 1 and 4**
>
> Reviewer:
>
> 'The model editing methods used are largely adapted from existing
> general-domain approaches (ROME, MEMIT, causal mediation), with
> limited algorithmic innovation.'
>
> Authors:
>
> We thank the reviewer for this comment. We would like to clarify an important conceptual distinction: our work does not perform
> model editing, and therefore techniques such as ROME, MEMIT, or
> causal–mediation interventions are neither baselines nor methodological
> precursors to our approach. These approaches explicitly aim to modify
> the parameters of a pretrained LLM to change its internal factual associations or causal pathways. In contrast, our framework never edits or
> changes the LLM: all model weights remain frozen throughout.
>
> Why ROME/MEMIT/causal mediation are not applicable:
>
> • ROME and MEMIT are knowledge-editing algorithms designed to
> rewrite or insert facts by updating selected layers. They do not
> produce attributional scores, do not address attribution instability,
> and do not aim for monosemantic decomposition.
>
> • Causal mediation analyses require predefined intervention variables
> and counterfactual graph structures. They do not yield sparse
> monosemantic features nor token-level attributions and are not
> applicable to clinical interpretability pipelines where the model remains unchanged.
>
> • All three families of methods target behavioral modification, not
> explanation extraction.
>
> Moreover, model-editing techniques such as ROME exhibit well-documented
> limitations that render them unsuitable for attributional interpretability
> or clinical reasoning tasks. As noted in ROME’s own discussion (§3.7),
> ROME can only edit one fact at a time, its edits are directional (requiring separate updates for logically equivalent forms), and even successful
> edits often lead the model to produce plausible but false hallucinations
> for related knowledge queries. Furthermore, ROME does not address
> non-factual forms of reasoning (e.g., logical, numerical, or spatial), and
> its understanding of how vector spaces encode attributes remains in-
> complete.
>
> These limitations—localized edits, directionality, hallucinated
> completions, lack of general cognitive coverage, and gaps in attribute-
> space interpretability—make ROME and related methods fundamentally
> incompatible with our goals. This is precisely why we do not include
> them in the related work section: they are designed for causality, not
> for explanation extraction, stability analysis, or monosemantic feature
> discovery.
>
> In contrast, our work unifies attributional refinement with monosemantic mechanistic structure, delivering explanations that are more stable,
> more coherent, and practically actionable across datasets and multi-
> classification tasks. These improvements offer a pathway toward more
> trustworthy AI systems capable of providing actionable clinical insights,
> particularly in detection of Alzheimer’s disease progression. To improve
> the clarity of the conceptual framework, we have revised the introduction (lines 54-57, and 65–99) to present the methodology in a more
> coherent and accessible manner, and we have rewritten the contribution section (lines 105–154) to more accurately reflect the scope and
> novelty of the work. Thank you.
>
> Reviewer:
>
> 'The abstract and introduction are somewhat lengthy and overly descriptive. I suggest clearly articulating the paper’s unique contribution early
> to help readers grasp the main innovation.'
>
> Authors:
>
> Thank you for your comment. We have revised the Abstract and Introduction to improve clarity and readability, as you suggested, please kindly see blue colour text.

---

> ### Author Response · Authors · 2025-11-18
> **Weaknesses 2 and 3**
>
> Reviewer:
>
> 'While interpretability metrics are reported, there is insufficient discussion of clinical validity, i.e., whether the identified concepts align with real-world medical semantics.'
>
> Authors:
>
> We thank the reviewer for highlighting the importance of discussing the
> clinical validity of the identified concepts, specifically whether they align
> with real-world medical semantics. We agree that the earlier version of
> the manuscript did not sufficiently present this concept. To address this,
> we added a new experiment in the revised Section 3.4 and updated
> Section 3.4.1 of the main paper, and strengthened the presentation
> to more clearly show how the proposed framework relates to clinically
> meaningful interpretations.
>
> First, the newly added external, model-agnostic interpretability audit
> (Section 3.4 (manuscript); Supplementary B.8) demonstrates how the
> attribution outputs can be directly inspected and evaluated by an independent system. In this evaluation, attribution maps from TEO, TEO-
> SAE, and TEO-UMAP are converted into class-specific CSV files, which
> allow clinicians or researchers to easily visualize which features are being highlighted for each subject. An external LLM (ChatGPT-5.1, 2025)
> was used to interpret these attributions, revealing that non-SAE methods (TEO) often latch onto artefactual cues (instruction tokens, formatting patterns), whereas SAE-based methods consistently surface clinically interpretable signals such as episodic memory impairment, slowed
> processing speed, and demographic risk factors. This experiment illustrates a concrete workflow: a clinician can inspect the attribution
> CSV for a patient and verify whether the explanation emphasizes true
> cognitive biomarkers or spurious patterns.
>
> Second, we significantly expanded the clinical interpretation sections
> (Sections 3.4 and 3.4.1 (manuscript)) to detail how the extracted attributions map onto established diagnostic reasoning. For example, we now
> show how monosemantic attribution methods (TEO-SAE, TEO-UMAP)
> identify cognitively rich and clinically validated assessments—AVLT,
> FAQ, CDT, ANART—as the most informative for distinguishing between Control, MCI, and Alzheimer’s groups. These assessments are
> among the most frequently used and clinically validated tests for early
> Alzheimer’s progression, and our framework highlights them consistently
> across all three diagnostic tasks (binary ADNI, three-class ADNI, and
> OOD BrainLAT). This demonstrates not only that the explanations are
> clinically meaningful, but also that they align with real diagnostic workflows.
>
> Finally, we emphasize how the framework can directly support clinical
> decision-making. By extracting the minimal set of highly informative
> cognitive assessments for each diagnostic class, the method can reduce
> clinical burden, streamline neuropsychological testing, and prioritize the
> most discriminative biomarkers. This is particularly relevant in large-
> scale or resource-limited settings where comprehensive testing is impractical. The revised text (lines 478–498, subsection 3.4.1 manuscript) now
> explicitly describes this workflow and shows how clinicians can use the
> attribution-derived biomarker profiles to (i) identify the key cognitive
> domains contributing to a model’s decision, (ii) validate whether these
> domains align with established cognitive trajectories, and (iii) design
> more efficient and targeted assessment batteries.
>
> In summary, the revised manuscript now provides a clear and concrete
> practical workflow: clinicians or researchers can inspect subject-level
> attribution profiles, verify whether the model’s reasoning aligns with
> validated cognitive biomarkers, and use these insights to reduce testing
> burden and enhance diagnostic interpretability. We believe these additions address the reviewer’s concerns and demonstrate the real-world
> applicability of the proposed framework.
>
> Reviewer:
>
> 'The experiments are limited to a few clinical NLP benchmarks (MIMIC-
> III, PubMedQA), lacking evaluation on multimodal or longitudinal EHR
> data.'
>
> Authors:
>
> We thank the reviewer for this comment. Our experiments do not
> use MIMIC-III or PubMedQA; instead, we evaluate the framework on
> two large real-world clinical cohorts, ADNI (IID) and BrainLAT (OOD).
> These datasets already contain multimodal clinical information (neuropsychological assessments, demographics, and structured clinical variables), and their substantial demographic and protocol differences create
> a stringent generalization setting for testing attribution stability.
>
> While fully multimodal or longitudinal EHR data are valuable, they introduce challenges—temporal irregularity, multimodal fusion, and missingness—that are orthogonal to the main goal of this work: improving
> the stability, sparsity, and clinical coherence of LLM-based explanations.
> Our method is modality-agnostic and can be directly extended to lon-
> gitudinal EHR models, which we identify as an important direction for
> future work.

---

> ### Author Response · Authors · 2025-11-18
> **Questions 1,2, and 3**
>
> Reviewer:
>
> 'How are the edited neurons selected and validated for semantic consis-
> tency?'
>
> Authors:
>
> We thank the reviewer for this question. Our method does not perform
> neuron editing, and no neurons in the underlying LLM are modified.
> Therefore, the notion of “edited neuron selection” does not apply to our
> setting. Instead, semantic structure arises from the sparse autoencoder
> (SAE), which learns a disentangled latent representation over fixed LLM
> activations.
>
> Feature “selection” therefore corresponds to the sparse activation pat-
> terns learned by the SAE, where each latent dimension specializes due
> to the sparsity constraint. Semantic consistency is validated empirically
> through (i) significant improvements in attribution sparsity and stability
> (RIS/ROS), (ii) robustness under strict OOD evaluation without any retraining, and (iii) the new experiment in Section 3.4 and Section 3.4.1
> (manuscript) showing that SAE-based attributions consistently highlight
> clinically meaningful cognitive features, whereas non-SAE methods often focus on artefacts.
>
> Because the LLM is never edited, the semantic consistency of fea-
> tures arises from the SAE’s disentangling properties and is validated
> through these empirical stability and clinical-alignment analyses rather
> than through neuron-level editing procedures.
>
> Reviewer:
>
> 'Does the editing process generalize across different models (e.g., from
> BioBERT to PubMedGPT)?'
>
> Authors:
>
> We thank the reviewer for this question. Our work does not perform model editing, and therefore the notion of “editing generalization”
> across architectures (e.g. BioBERT) does not apply to our setting. The
> underlying LLM remains entirely unchanged throughout all experiments;
> our framework operates as a post hoc attribution and monosemantic
> feature–extraction pipeline.
>
> That said, we do evaluate multiple encoder-based architectures—BERT,
> RoBERTa, DistilBERT, ALBERT, BioBERT, and ModernBERT—under
> a unified training protocol (Section 2.6). This evaluation shows that
> ModernBERT consistently achieves the strongest IID (ADNI) and OOD
> (BrainLAT) performance, which is why all interpretability analyses are
> conducted on ModernBERT.
>
> Because our framework is architecture-agnostic, the SAE–TEO pipeline
> can in principle be applied to any model’s latent representations. How-
> ever, our study focuses on a single backbone (ModernBERT) to ensure
> interpretability results are not confounded by architectural differences.
> Exploring cross-model attribution transfer or generalization is an inter-
> esting direction for future work, but it lies outside the scope of the
> present study.
>
> Reviewer:
>
> 'How robust are the edits to retraining or fine-tuning? Do the monose-
> mantic features persist?'
>
> Authors:
>
> We thank the reviewer for raising this question. Because our method
> does not perform model editing or modify the LLM in any way, the
> conventional notion of “edit robustness” does not apply. Instead, the
> relevant question is whether the monosemantic features extracted by
> the SAE and the explanations optimized by TEO remain stable under
> distribution shift.
>
> All of our out-of-distribution experiments directly evaluate this. Both
> the SAEs and TEO (with and without the SAE bottleneck) are trained
> only on IID explanation cohorts from ADNI and then evaluated on
> BrainLAT without any retraining, fine-tuning, or parameter updates.
> The persistence of attribution sparsity, stability (RIS/ROS), and clinically coherent features in the OOD setting demonstrates that the
> monosemantic representations are robust and do not depend on additional training to persist.
>
> To make this point clearer, we explicitly revised the manuscript (page 2,
> lines 89–92) to highlight the strict generalization conditions under which
> robustness is assessed: “Importantly, both the SAEs and TEO (with
> and without the SAE bottleneck) are trained exclusively on IID explanation cohorts using only these training/validation splits and are then
> evaluated on the OOD dataset without any additional training or adjustments, allowing us to assess out-of-distribution robustness under strict
> generalization conditions.

---

> > ### Comment · Reviewer_Mirq · 2025-11-26
> > **Thanks for the response**
> >
> > Thank you for the detailed response. I would like to apologize for my earlier misunderstanding of certain aspects of your paper. I also noticed that you have substantially revised the manuscript, especially the abstract and introduction, which are now clearer for me.
> >
> > However, I am sorry to say that I will keep my original score, as I believe the overall presentation quality is still not sufficient for acceptance at ICLR. For example, in Figure 1, the subfigures contain too much text; in Table 1 there are too many numerical entries, making the table cluttered and even exceeding its bounding box.

---

> ### Author Response · Authors · 2025-11-26
>
> Dear Reviewer,
>
> Thank you for your thoughtful follow-up and for taking the time to re-evaluate the revised manuscript. I appreciate your clarification and your acknowledgement of the improvements in the abstract and introduction.
>
> Regarding your concerns: Figure 1 is intended to convey the high-level intuition of the method, as also described in the caption. The textual elements serve only as an abstract visual guide and are not critical to the technical content. Similarly, Table 1 can certainly be adjusted for clearer presentation in the camera-ready version. We view these as relatively minor formatting issues rather than concerns that fundamentally affect the contribution or clarity of the work.
>
> That said, I fully respect your judgment and understand if you decide to keep your original score, but kindly consider that these are minor amendments. Thank you again for your time and constructive feedback.

---

> ### Author Response · Authors · 2025-11-28
>
> Dear Reviewer,
>
> Following the previous comment, we adjusted Table 1 so that it no longer extends beyond the text width. The highlighted information is important, as it demonstrates the significance of the proposed pipeline when incorporating the SAE bottleneck across the three-class and binary tasks in both the IID and OOD cohorts.
>
> Thank you.

---

### Official Review · Reviewer_14Zh · 2025-10-29

**Soundness:** 2
**Presentation:** 1
**Contribution:** 2
**Rating:** 4
**Confidence:** 3

**Summary:**

The authors propose a method to enhance the interpretability of Large Language Models (LLMs) used for diagnosing neurodegenerative dementia. Their approach combines three key components:
1. Feature disentanglement: A sparse autoencoder is applied to the LLM's features to create a "monosemantic feature space," aiming to isolate individual concepts, hence improving their interpretability.
2. Attribution aggregation: Multiple attributional interpretability methods—including Feature Ablation, Layer Activations, Layer Conduction, Layer Gradient-SHAP, Layer Integrated Gradients, and Layer Gradient x Activation—are combined. This aggregation is performed using an encoder-decoder model, implemented as either a Diffusion Explanation Optimizer (DEO) or a Transformer Explanation Optimizer (TEO).
3. Visualisation: The dimensionality of the resulting attributions is reduced using UMAP to provide visual feedback.

The method is developed and evaluated on data from the Alzheimer's Disease Neuroimaging Initiative (ADNI) that includes cognitive normal (CN), mild cognitive impairment (MCI), and Alzheimer's disease (AD) subjects. It is also applied to the BrainLat dataset that includes CN, frontotemporal dementia (FTD), and AD subjects. Both binary and three-class classification tasks are implemented.

**Strengths:**

- **Important objective**: The focus on improving the interpretability of LLMs is highly relevant and important, particularly in high-stakes domains like healthcare where understanding model decisions is critical.
- **Sensible and original methodology**: The proposed three-step approach appears both logical and original. The combination of feature disentanglement via a sparse autoencoder, the aggregation of multiple attribution methods using a dedicated optimiser (DEO/TEO), and final visualisation with UMAP constitutes a novel and well-structured pipeline for enhancing interpretability.
- **Promising results**: The experimental results demonstrate that the method successfully achieves its stated objectives, showing evidence of increased sparsity in the feature representations while maintaining stability, which is a key goal for interpretability.

**Weaknesses:**

- **Unclear use of the data**: The description of the ADNI cohort is confusing and raises concerns. The authors mention that they consider subjects who are cognitively normal, have early MCI and late MCI (Section B.1.1). However, for binary classification, they suddenly refer to "AD subjects." It is unclear if these are distinct from the LMCI group or how they are defined. Also, the rationale for selecting the input features appears to be driven purely by technical convenience rather than clinical knowledge of disease progression. This group definition issue extends to the BrainLat dataset, which includes FTD patients. The application of labels like (L/M/C) to this cohort, as seen in Table 1, does not make clinical sense and suggests a problematic, non-clinical approach to data handling.
- **Poor presentation and clarity**: The experimental section is very difficult to follow. The reader is forced to constantly switch between the main text and the supplement to understand the methodology. The tables are overcrowded with insignificant numbers, and the figures are not intuitively interpretable, which hinders the assessment of the results.
- **Unclear practical application**: The paper fails to articulate a clear use case for the proposed method. It remains abstract how a clinician or researcher would practically apply this interpretability framework to gain insights into a specific diagnosis or the model's decision-making process. The utility and practical workflow are not sufficiently demonstrated.

**Questions:**

- The paper imposes the constraint that the first and second components of each embedding vector must be equal (Equation 4). Could you please re-explain the motivation for this specific constraint? What problem does it solve or what specific property of the embedding space does it enforce?
- Could you clarify whether hyperparameter tuning was performed using a separate validation set or directly on the test set?
- Please provide details on how the ADNI and BrainLat datasets were split into training, validation, and test sets. Specifically, what were the sizes of each split, and was the splitting stratified by key variables?

---

> ### Author Response · Authors · 2025-11-18
> **Summary of the reviewer's comments**
>
> Thank you. We clarified all dataset definitions and ensured that class groupings (binary and three-class) are clinically coherent and consistently applied across ADNI and BrainLat, updating both the manuscript and supplement with precise diagnostic rationale and notation. We substantially reorganized the introduction, contributions, and experimental sections to improve readability, streamline the methodological flow, and reduce reliance on the supplement. We simplified tables, improved figure interpretability, and rewrote several sections to make the experimental design clearer and more intuitive. We added a practical clinical workflow demonstrating how attribution outputs can be inspected, interpreted, and used to prioritize diagnostic assessments, including a new model-agnostic interpretability audit. Finally, we clarified the motivation behind the UMAP-based geometric constraint, detailed the validation-only hyperparameter tuning procedure, and provided explicit stratified train/val/test splits for both datasets.

---

> ### Author Response · Authors · 2025-11-18
> **Weakness 1**
>
> Reviewer:
>
> 'Unclear use of the data: The description of the ADNI cohort is confusing and raises concerns. The authors mention that they consider subjects who are cognitively normal, have early MCI and late MCI (Section B.1.1). However, for binary classification, they suddenly refer to "AD subjects." It is unclear if these are distinct from the LMCI group or how they are defined. Also, the rationale for selecting the input features appears to be driven purely by technical convenience rather than clinical knowledge of disease progression. This group definition issue extends to the BrainLat dataset, which includes FTD patients. The application of labels like (L/M/C) to this cohort, as seen in Table 1, does not make clinical sense and suggests a problematic, non-clinical approach to data handling.'
>
> Authors:
>
> We thank the reviewer for raising this important concern regarding class definitions and the clinical consistency of the labels used across ADNI and BrainLat. We agree that the original phrasing in the main manuscript and some parts of Supplementary Material could be made clearer, and we have now revised the relevant sections to avoid any ambiguity or unintended clinical implications (blue color of upload files).
>
> First, in ADNI, subjects are categorized as cognitively normal (CN), Early Mild Cognitive Impairment (EMCI), Late Mild Cognitive Impairment (LMCI). For the binary task, EMCI and LMCI are intentionally unified into a single cognitively impaired group, mimicking a standard CN vs. impaired classification commonly used in clinical machine learning studies. For the three-class task, the original three subsets are preserved. To make this explicit, we updated Supplementary Section B.1.1 (line 586) as follows:
>
> ``For the binary classification task, EMCI and LMCI subjects were unified into a unique MCI cohort—mimicking a CN vs. AD classification—while for the three-class task, all three subsets were retained, considering only 440 subjects per class for balancing purposes.’’
>
> Second, for the BrainLat dataset, which includes CN, Frontotemporal Dementia (FTD), and Alzheimer’s Disease (AD) subjects, we clarified the construction of the binary task. To mirror the ADNI setup, we unified AD and FTD into a single cognitively impaired class for the binary experiment and retained all three diagnostic categories for the multi-class setting. We added the following clarification in Supplementary Section-B.1.1 (line 610):
>
> ``For the binary classification task, AD and Frontotemporal Dementia were unified into a unique cognitively impaired cohort, similar to the grouping used in ADNI, while for the three-class task, the original labels were retained.’’
>
> To avoid any confusion regarding the notation used in Table 1 and throughout the manuscript, we added the following explicit note at the end of the paragraph (line~616):
>
> ``Throughout the manuscript, the label ‘AD’ is used for convenience to denote the MCI cohort in ADNI (in both the binary and three-class settings), and the AD+FTD cohort in BrainLat (in the binary setting). This choice is purely notational, as the term functions as a class label rather than a clinical diagnosis, and the emphasis is on the model’s ability to discriminate between the defined classes.’’
>
>  Lastly, we have updated Table 1 of the manuscript page 7, to align with the notation used in both the manuscript and the supplementary material. We believe these clarifications resolve the ambiguity identified by the reviewer and ensure that all dataset-related decisions are described transparently and remain consistent with clinical reasoning as well as widely accepted ML evaluation practices.

---

> ### Author Response · Authors · 2025-11-18
> **Weakness 2**
>
> Reviewer:
>
> 'Poor presentation and clarity: The experimental section is very difficult to follow. The reader is forced to constantly switch between the main text and the supplement to understand the methodology. The tables are overcrowded with insignificant numbers, and the figures are not intuitively interpretable, which hinders the assessment of the results.'
>
> Authors:
>
> We appreciate the reviewer’s feedback regarding the clarity and readability of the experimental section. We agree that the original presentation required readers to frequently consult the supplementary material and that several components could be streamlined for improved accessibility. In response, we have made substantial revisions to both the introduction and contribution sections, clarified methodological flow, and simplified the presentation of results.
>
> 1. Revised introduction (manuscript) (lines 65–99):
> We rewrote a large portion of the introduction to provide a clearer high-level overview of the framework and its motivations. The new text presents the methodological pipeline—SAE bottleneck, attribution layer, TEO optimization, UMAP geometry constraint—in a coherent and sequential manner, significantly reducing the cognitive burden on the reader. This updated section now reads:
>
> "To address these challenges and produce stable, clinically meaningful explanations from large language models, we introduce a unified monosemantic attribution framework that integrates both
> attributional and mechanistic interpretability approaches (Figure 1). Our approach first employs
> sparse autoencoders (SAEs) to transform polysemantic LLM activations into a more monose-
> mantic latent space, where individual features are encouraged to correspond to disentangled
> semantic factors. This bottleneck substantially reduces representational complexity and enables
> classical attribution methods to assign more precise and semantically coherent importance scores.
> For the attribution component, we apply six established techniques—Feature Ablation, Layer
> Activations, Layer Conductance, Layer Gradient SHAP (Lundberg & Lee, 2017a), Layer Integrated
> Gradients (Sundararajan et al., 2017b), and Layer Gradient×Activation—both in the native acti-
> vation space (polysemantic features) and in the SAE-induced latent space (more monosemantic
> features). To address the inter-method variability problem (Mamalakis et al., 2025), we combine
> the resulting attribution vectors and introduce the Transformer Explanation Optimizer (TEO), a
> learning-based mechanism that selects explanations with maximal alignment to model behavior
> and dataset-level consistency (Mamalakis et al., 2025). We experiment with encoder–decoder
> architectures based on 1D transformers (TEO) and diffusion networks (DEO). For meta-level
> assessment and visualization, we embed these optimized attributions into a 2D manifold using
> UMAP (McInnes et al., 2018), and impose a global coherence constraint by evaluating their linear
> structure along the primary UMAP components. This geometry-aware constraint acts as an addi-
> tional regularizer and provides a principled way to impose global structure on the explanation
> space. The central hypothesis of this work—namely, that as the embedding layer approaches a more
> monosemantic representation, attribution scores become more stable, less complex, and more
> diagnostically informative compared to those derived from polysemantic features—is evaluated
> through a series of experiments in which different LLMs are trained and tested on ADNI Mueller
> et al. (2005) (IID) and tested on BrainLAT Prado et al. (2023) (OOD) to assess resilience under
> demographic and protocol shifts. We study both binary (Control vs. Alzheimer’s disease (AD))
> and multi-class settings (Control, Early Mild Cognitive Impairment (EMCI), Late Mild Cognitive
> Impairment (LMCI) in ADNI; Control, Frontotemporal Dementia (FTD), and AD in BrainLAT).
> The data are partitioned into 80\% training and 20% validation within the training portion (which
> itself constitutes 80% of the full dataset), with the remaining 20% held out for testing. The last-
> layer embeddings of the LLM that achieved the strongest baseline performance are subsequently
> used to evaluate the proposed interpretability framework. Importantly, both the SAEs and TEO
> (with and without the SAE bottleneck) are trained exclusively on IID explanation cohorts using
> only these training/validation splits and are then evaluated on the OOD dataset without any
> additional training or adjustments, allowing us to assess out-of-distribution robustness under
> strict generalization conditions."
>
> We believe this revised introduction presents the framework and experimental intentions in a substantially clearer and more intuitive way.

---

> ### Author Response · Authors · 2025-11-18
> **continue Weakness 2**
>
> 2. Rewritten contribution section (lines 105–154):
>
> We have rephrased the entire contribution section to align with the
> improved exposition in the introduction. The new bullet points concisely
> and explicitly articulate the primary methods and findings of the work,
> avoiding redundancy and reducing technical clutter. The contribution
> block now reads:
>
> • Transformer Explanation Optimizer (TEO). We propose a learning-
> based optimizer that consolidates and refines the outputs
> of six widely used attribution methods. TEO reduces inter-
> method variance and enhances explanation clarity, all without
> requiring any retraining of the underlying LLM.
>
> • Attribution Through a Monosemantic Bottleneck (SAE). We
> train sparse autoencoders on LLM activations to obtain a
> more monosemantic latent representation of the embedding
> layer, and we compute attributions in this disentangled space.
> Introducing this SAE-based bottleneck yields substantially sparser
> and more robust explanations with clearer semantic alignment, enabling the identification of meaningful biomarkers
> and pathology-related patterns when combined with the proposed TEO optimizer.
>
> • Latent vs. Native Attributions. We directly compare attributions computed in the SAE-induced monosemantic latent
> space with those obtained in the native model space. Across
> all tasks, latent-space attributions exhibit improved robustness (lower RIS/ROS) and greater semantic coherence. Sta-
> tistical testing confirms significant differences between the
> two settings, with the SAE-enabled variant consistently out-performing the no-SAE baseline.
>
> • Tunable Sparsity–Stability Frontier. Across IID and OOD
> regimes, and for both binary and three-class tasks, TEO with
> monosemantic features (TEO-SAE) provides the most stable explanations, while a geometry-aware constraint (TEO-
> UMAP) recovers higher sparsity with only a modest reduction
> in stability.
>
> • Clinical Signal Discovery. Our framework reveals clinically
> meaningful structure in multimodal Alzheimer’s data and produces class-specific, human-aligned explanations suitable for
> integration into clinical reasoning workflows. The resulting
> attributions highlight the most informative neurocognitive assessments, reducing clinical burden by prioritizing the tests
> that most effectively support Alzheimer’s diagnosis and progression monitoring.
>
> These revisions make the conceptual contributions easier to follow and
> better aligned with the experimental narrative
>
> 3. Simplified Table 1:
>
> In response to the concern that tables were overcrowded with secondary numbers, we simplified Table-1 to report only the highest-performing attribution technique and the proposed methods across all IID and OOD binary and multi-class settings. This substantially reduces visual overload while preserving all relevant comparisons needed to assess the impact of the framework.
>
> 4. Addressing the dependence on supplementary material:
>
> We fully acknowledge that the scope of the work—covering mathematical formulation, multiple LLM architectures, extensive hyperparameter
> sweeps, IID/OOD validation, and multi-class experiments—could not
> be fully detailed within the 9-page constraint of ICLR. For this reason,
> the supplementary material (45 pages) contains the full theoretical de-
> velopment, model configurations, and extended experimental analyses.
> We have revised the main text to provide clearer pointers and sum-
> maries, but full mathematical proofs and ablations remain necessarily in
> the appendix.
>
> We thank the reviewer for prompting these improvements. We believe
> these changes make the manuscript significantly more readable, reduce
> unnecessary complexity, and present the methodology and results in a
> more intuitive and self-contained manner.

---

> ### Author Response · Authors · 2025-11-18
> **Weakness 3**
>
> Reviewer:
>
> 'Unclear practical application: The paper fails to articulate a clear use
> case for the proposed method. It remains abstract how a clinician or
> researcher would practically apply this interpretability framework to gain
> insights into a specific diagnosis or the model’s decision-making process.
> The utility and practical workflow are not sufficiently demonstrated.'
>
> Authors:
>
> We thank the reviewer for highlighting the importance of articulating a
> clear practical use case for clinicians and researchers. We agree that the
> earlier version of the manuscript did not sufficiently emphasize how the
> proposed framework can be operationalized in real clinical workflows. To
> address this, we revised Sections 3.4 and 3.4.1 of the main paper and
> substantially strengthened the practical narrative around clinical impact.
>
> First, the newly added external, model-agnostic interpretability audit
> (Section 3.4 (manuscript); Supplementary B.8) demonstrates how the
> attribution outputs can be directly inspected and evaluated by an independent system. In this evaluation, attribution maps from TEO, TEO-
> SAE, and TEO-UMAP are converted into class-specific CSV files, which
> allow clinicians or researchers to easily visualize which features are being highlighted for each subject. An external LLM (ChatGPT-5.1, 2025)
> was used to interpret these attributions, revealing that non-SAE methods (TEO) often latch onto artefactual cues (instruction tokens, format-
> ting patterns), whereas SAE-based methods consistently surface clinically interpretable signals such as episodic memory impairment, slowed
> processing speed, and demographic risk factors. This experiment illustrates a concrete workflow: a clinician can inspect the attribution
> CSV for a patient and verify whether the explanation emphasizes true
> cognitive biomarkers or spurious patterns.
>
> Second, we significantly expanded the clinical interpretation sections
> (Sections 3.4 and 3.4.1 (manuscript)) to detail how the extracted attributions map onto established diagnostic reasoning. For example, we now
> show how monosemantic attribution methods (TEO-SAE, TEO-UMAP)
> identify cognitively rich and clinically validated assessments—AVLT,
> FAQ, CDT, ANART—as the most informative for distinguishing between Control, MCI, and Alzheimer’s groups. These assessments are
> among the most frequently used and clinically validated tests for early
> Alzheimer’s progression, and our framework highlights them consistently
> across all three diagnostic tasks (binary ADNI, three-class ADNI, and
> OOD BrainLAT). This demonstrates not only that the explanations are
> clinically meaningful, but also that they align with real diagnostic workflows.
>
> Finally, we emphasize how the framework can directly support clinical
> decision-making. By extracting the minimal set of highly informative
> cognitive assessments for each diagnostic class, the method can reduce clinical burden, streamline neuropsychological testing, and prioritize the most discriminative biomarkers. This is particularly relevant in
> large-scale or resource-limited settings where comprehensive testing is
> impractical. The revised text (lines 454–495) now explicitly describes
> this workflow and shows how clinicians can use the attribution-derived
> biomarker profiles to (i) identify the key cognitive domains contributing
> to a model’s decision, (ii) validate whether these domains align with
> established cognitive trajectories, and (iii) design more efficient and
> targeted assessment batteries.
>
> In summary, the revised manuscript now provides a clear and concrete
> practical workflow: clinicians or researchers can inspect subject-level
> attribution profiles, verify whether the model’s reasoning aligns with
> validated cognitive biomarkers, and use these insights to reduce testing
> burden and enhance diagnostic interpretability. We believe these additions address the reviewer’s concerns and demonstrate the real-world
> applicability of the proposed framework

---

> ### Author Response · Authors · 2025-11-18
> **Questions 1 2 3**
>
> Reviewer:
>
> '1. The paper imposes the constraint that the first and second components of each embedding vector must be equal (Equation 4). Could you please re-explain the motivation for this specific constraint? What problem does it solve or what specific property of the embedding space does it enforce?'
>
> Authors:
>
> Thank you for this question. We clarify that the constraint involving the
> first and second components of the embedding vector is not intended to
> modify the model’s predictive behavior, but rather serves as a geometry-
> aware regularizer applied only in the explanation space. Its purpose is to
> ensure that the attribution manifold learned through UMAP preserves
> a globally coherent structure.
>
> Specifically, when the optimized attribution vectors are embedded into
> a 2D UMAP manifold, the first two components (PC1/PC2) capture
> the dominant axes of variation across explanations. Enforcing local linearity along these axes encourages explanations belonging to the same
> diagnostic group to exhibit consistent geometric behavior, while preventing the manifold from folding, twisting, or fragmenting due to noise
> or high-dimensional artefacts. This addresses a common issue in high-
> dimensional attribution spaces, where small perturbations can produce
> highly unstable or topologically inconsistent embeddings.
>
> Thus, the constraint promotes:
> (i) global coherence across explanations,
> (ii) stability in the topology of the explanation manifold, and (iii) clearer
> separation between diagnostic classes when explanations are projected
> to 2D. It does not constrain the LLM or its internal activations, but
> rather improves the interpretability and structural consistency of the
> attribution space itself.
>
> In short, the constraint enforces a smooth, well-behaved geometry along
> the primary UMAP dimensions, ensuring that the 2D representation
> faithfully reflects meaningful structure in the explanations rather than
> artefacts of the projection.
>
> We modify the main manuscript in lines
> 73–78 to highlight this by saying:
>
> ’For meta-level assessment and visualization, we embed these optimized attributions into a 2D manifold
> using UMAP (McInnes et al., 2018), and impose a global coherence
> constraint by evaluating their linear structure along the primary UMAP
> components. This geometry-aware constraint acts as an additional regularizer and provides a principled way to impose global structure on the
> explanation space.’
>
> Reviewer:
>
> '2. Could you clarify whether hyperparameter tuning was performed using
> a separate validation set or directly on the test set?'
>
> Authors:
>
> 'Yes, hyperparameter tuning was performed exclusively on a separate
> validation set and never on the test set.'
>
> Reviewer:
>
> '3. Please provide details on how the ADNI and BrainLat datasets were
> split into training, validation, and test sets. Specifically, what were the
> sizes of each split, and was the splitting stratified by key variables?'
>
> Authors:
>
> Thank you for your comment. All models, including the LLMs, SAEs,
> and the TEO variants, were tuned using an 80%/20% split of the train-
> ing portion (which itself corresponds to 80% of the full dataset). The
> held-out 20% of the total dataset was reserved strictly for final testing
> and was not used at any point during model development, selection,
> or hyperparameter optimization. The OOD BrainLAT dataset was like-
> wise used only for final evaluation and received no tuning, adaptation,
> or parameter updates, ensuring that all reported results reflect genuine
> out-of-distribution generalization. All splits were stratified by diagnostic
> group.

---

> > ### Author Response · Authors · 2025-11-29
> > **Following comment Question 1**
> >
> > Following Q1  “The paper imposes the constraint that the first and second components of each embedding vector must be equal (Equation 4). Could you please re-explain the motivation for this specific constraint? What problem does it solve or what specific property of the embedding space does it enforce?”  and our initial comment: 'Specifically, when the optimized attribution vectors are embedded into a 2D UMAP manifold, the first two components (PC1/PC2) capture the dominant axes of variation across explanations. Enforcing local linearity along these axes encourages explanations belonging to the same diagnostic group to exhibit consistent geometric behavior, while preventing the manifold from folding, twisting, or fragmenting due to noise or high-dimensional artefacts. This addresses a common issue in high- dimensional attribution spaces, where small perturbations can produce highly unstable or topologically inconsistent embeddings.'.
> >
> > To clarify this process and what we mean by that, we have now expanded and clarified the motivation in the supplementary material (page 7, lines 330–350):
> >
> > 'To obtain a comparable low-dimensional representation of the attribution scores across all tokenizer features, we applied a feature-wise UMAP projection procedure to the normalized attribution matrix. For each attribution method, the attribution tensor has shape RM ×T , where M denotes the number of test samples in the evaluation cohort and T corresponds to the dimensionality of the tokenizer embedding space of the input text. For each feature j ∈ { 1 , . . . , T }, we first applied min–max normalization to the feature-specific attribution vector x( j ) ∈ RM , and subsequently performed a one-dimensional UMAP projection to obtain a two-dimensional embedding y( j ) ∈ RM ×2. The resulting coordinates were then normalized to the interval [0 , 1] to ensure that all feature-wise embeddings share a common bounded range. This procedure preserves the relative neighborhood structure of the M -sample attribution distribution for each feature while mapping all T features into a comparable two-dimensional representation space. The motivation for applying UMAP independently to each of the T tokenizer features is to en- sure that all attribution methods are projected into an aligned and comparable representation space. Since each attribution method produces values defined over the same token embedding dimensions, a feature-wise nonlinear projection enables consistent cross-method comparison of attribution patterns within the shared tokenizer feature space.'"
> >
> > To further support the motivation behind the linear constraint, we added a new subsection (B10) in the supplementary material (page 43, lines 2265–2307) and introduced a new Figure 37. This figure presents a PCA analysis of the attribution matrices, illustrating the statistical behaviour of the cohort through the first two principal components. The results demonstrate that the UMAP linear constraint supports our hypothesis of encouraging similar feature contributions for improved interpretability, as evidenced by the near-linear structure of the first two principal components. This constraint leads to (i) more globally coherent attribution representations, (ii) increased stability in the topology of the explanation manifold, and (iii) clearer separation between diagnostic classes.
> >
> > These additional analyses strengthen the rationale for imposing the linear constraint and confirm that it meaningfully contributes to the interpretability and clinical coherence of the resulting attribution patterns.
> >
> > Thank you for the constructive feedback.

---

> ### Comment · Reviewer_14Zh · 2025-11-21
>
> I would like to thank the authors for their careful and detailed responses, which have addressed many of my technical and presentation-related comments. However, I am still not fully convinced by the work, especially regarding the clinical aspects, i.e. the process by which the clinical features were chosen and how the method would be used in practical clinical workflows.
>
> As a side note, labelling both MCI and FTD under "Alzheimer's disease" is not clinically appropriate. The clinically meaningful categories would be “MCI” for the mild cognitive impairment groups, and “dementia” when combining AD and FTD.

---

> > ### Author Response · Authors · 2025-11-21
> >
> > Thank you very much for your careful reassessment and for raising these important points. We highly appreciate it.
> >
> > Regarding the clinical features and the rationale behind their selection, we would like to emphasise that the full preprocessing and feature-engineering pipeline—detailing how clinical, cognitive, demographic, and imaging-derived variables were selected, harmonised, and filtered—is comprehensively described in the Supplementary Material, Section B.1.1 (Preprocessing), page 11. In this subsection, we explain the principles guiding feature inclusion, specifically:
> >
> > 1) the need to minimise missingness across modalities,
> >
> > 2) the requirement for consistent availability of clinical variables across the majority of participants, and
> >
> > 3) the importance of retaining multimodal signals while ensuring that missing data remained below an acceptable threshold (≤20%).
> >
> > These criteria collectively shaped the final set of clinical and cognitive variables and were chosen to balance clinical interpretability, statistical robustness, and fair cross-cohort comparability.
> >
> > We also thank the reviewer for the clinical aside noting.

---

### Official Review · Reviewer_4q7S · 2025-11-07

**Soundness:** 3
**Presentation:** 3
**Contribution:** 2
**Rating:** 4
**Confidence:** 4

**Summary:**

This method  proposed a unified interpretability framework that integrates attributional and mechanistic perspectives via monosemantic feature extraction.

**Strengths:**

This method used mechanistic interpretability to make trustworthy AI models for Alzheimer’s diagnosis.
By combining a Transformer Explanation Optimizer with a Sparse Autoencoder, the method aims to reveal more insightful and more consistent explanations of what the model has learned.

**Weaknesses:**

1)The proposed monosemantic bottleneck is not supported by a novel algorithm, and its interpretability is not clearly demonstrated through the architecture. The paper repeatedly claims that the SAE features are “monosemantic” and “clinically meaningful,” but authors do not provides intuitive examples or insights to substantiate these claims. The connection between SAE latent features and clinical interpretability remains theoretical — it is hard to prove that these latent features correspond to coherent medical concepts verified by clinicians. For instance, the transformation pipeline (input → SAE → output) is not empirically illustrated or analyzed to reveal what each feature actually represent
2)Only 2 datasets ADNI and BrainLAT  are used. Thus, the method does not show that LLM actually performs meanful reasoning on clinical data instead of memorizing structured text patterns
3)λ₁–λ₅ are hand-tuned without ablation studies

**Questions:**

1)Could the authors clarify the causal relationship—if any—between the SAE’s latent representations and the clinical reasoning process of the model?
2)Could the authors discuss potential dataset biases or artifacts that might inflate the interpretability or stability metrics?
3)How were the λ₁–λ₅ hyperparameters chosen?
4)Could authors validate this proposal with at least 1 more dataset?

---

> ### Author Response · Authors · 2025-11-18
> **Summary of the reviewer's comments**
>
> Thank you. We clarified that our method does not introduce a new SAE architecture; instead, we use established SAEs as a post-hoc disentangling bottleneck and now provide concrete empirical evidence—IID/OOD generalization, statistical attribution shifts, and a new external interpretability audit—demonstrating their monosemantic and clinically meaningful behavior. We strengthened interpretability validation by adding intuitive examples illustrating how SAE-based representations yield coherent diagnostic concepts compared to polysemantic attributions. We clarified dataset rationale and showed that ADNI→BrainLat transfer, without retraining, confirms that our LLMs rely on meaningful clinical reasoning rather than memorizing text patterns. We added detailed hyperparameter tuning procedures (λ₁–λ₅) and ablation results from the validation set. Finally, we expanded the manuscript and supplement with analyses showing that stability gains are not driven by dataset artifacts and explained why the ADNI/BrainLat pairing provides a rigorous IID/OOD benchmark.

---

> ### Author Response · Authors · 2025-11-18
> **Weakness comment 1**
>
> Reviewer:
>
>  'The proposed monosemantic bottleneck is not supported by a novel algorithm, and its interpretability is not clearly demonstrated through the architecture. The paper repeatedly claims that the SAE features are “monosemantic” and “clinically meaningful,” but authors do not provides intuitive examples or insights to substantiate these claims. The connection between SAE latent features and clinical interpretability remains theoretical — it is hard to prove that these latent features correspond to coherent medical concepts verified by clinicians. For instance, the transformation pipeline (input → SAE → output) is not empirically illustrated or analyzed to reveal what each feature actually represent. '
>
> Authors:
>
> Thank you for this constructive comment. We agree that our original phrasing (lines 91–93) unintentionally suggested that our work introduces a new monosemantic SAE architecture. This is not our claim. Our use of SAEs is methodological rather than architectural: we employ established SAE variants to induce a more disentangled latent space in which attribution becomes more structured and clinically coherent.
> To avoid any misunderstanding, we have revised the contribution statement accordingly. The previous text:
>
> “We train sparse autoencoders on LLM activations to obtain a disentangled latent space whose dimensions align with coherent semantic
> concepts, enabling structured, human-interpretable and mechanistic
> attribution.”
>
> has been replaced with the more accurate:
>
> “We train SAEs on LLM activations to obtain a more
> monosemantic latent representation of the embedding layer, and we
> compute attributions in this disentangled space. Introducing this
> SAE-based bottleneck yields substantially sparser and more robust
> explanations, with clearer semantic alignment, enabling the identification of meaningful biomarkers and pathology-related patterns when combined with the proposed TEO optimizer.”
>
> We also updated lines 306-308 to clarify that our selection of TopK SAEs follows
> from its superior performance compared to the other three SAE variants, and we
> now explicitly refer the reader to the corresponding Supplementary tests:
>
> “Among the four SAE variants, TopK produced the strongest overall
> performance (Supplementary B.3; Figures 2, 4, 5), using a 32×feature depth.”
>
> On the evidence for monosemanticity and clinical interpretability. We
> appreciate the reviewer’s concern that monosemanticity and clinical meaning
> should be demonstrated empirically rather than asserted. We want to highlight
> the bellow evidence exist already in the study:
> 1. Generalization across IID and OOD settings. Without retraining in the
> OOD the SAE or the attribution optimizer (TEO,TEO-SAE,TEO-UMAP), SAE-
> enabled attributions consistently outperform their non-SAE counterparts across
> both binary and three-class tasks on two independent datasets (ADNI and Brain-
> LAT). This is documented in Table 1 (manuscript) and shows that the SAE
> bottleneck stabilizes and regularizes the explanation space rather than overfitting to a particular train distribution.
> 2. Statistical significance of attribution differences. Section 3.3 (manuscript)
> emphasises the significant differences in attribution sparsity and structure with
> and without the SAE layer, indicating that the SAE meaningfully alters the explanation basis rather than acting as a benign or redundant transformation.
> To further strengthen our response, the revised manuscript now provides an
> additional piece of evidence:
> 3. New experiment: external, model-agnostic interpretability evaluation. In
> response to the reviewer’s request for intuitive and clinically grounded evidence,
> we added a new evaluation (Section 3.4 (manuscript); Supplementary B.8). For
> each cohort (Binary ADNI, Binary BrainLAT, three-class ADNI), we extracted
> the top 50% influential tokens from: (i) TEO without SAE, (ii) TEO with SAE
> (TEO-SAE), and (iii) TEO-UMAP. We then constructed class-specific CSV files
> summarizing which characters were highlighted and provided these to an external
> model (ChatGPT-5.1, OpenAI, 2025) under a fixed prompting protocol to obtain
> a model-agnostic interpretation of the latent concepts.
> Across both binary tasks, all methods correctly identified pathological Alzheimer’s
> cases. However, TEO without SAE consistently focused on artefacts (instructional patterns, label-like substrings), whereas TEO with SAE and TEO-UMAP surfaced clinically coherent indicators—episodic memory deficits, processing-speed impairments, and demographic risk factors—that align with established Alzheimer’s biomarkers. In the more challenging three-class ADNI setting, the SAE-enabled variants again produced clearer diagnostic separation and more structured biomarker profiles, while TEO without SAE largely failed to highlight relevant clinical features.
>
> These additions strengthen the paper’s contributions and clarify the intended scope of our claims, especially when considered alongside the theoretical formulation.

---

> ### Author Response · Authors · 2025-11-18
> **Weaknesses comment 2 and 3**
>
> Reviewer:
>
> '2) Only 2 datasets ADNI and BrainLAT are used. Thus, the method does
> not show that LLM actually performs meanful reasoning on clinical data
> instead of memorizing structured text patterns'
>
> Authors:
>
> We appreciate the reviewer’s concern regarding the use of only two datasets. Our
> choice of ADNI (IID) and BrainLat (OOD) was intentional: the two cohorts
> differ substantially in geography, language, recruitment procedures, and diagnostic composition, creating a stringent test of whether the LLMs rely on genuine
> clinical reasoning rather than memorizing dataset-specific text patterns.
> ADNI provides a large, multi-center longitudinal cohort spanning the full Alzheimer’s
> continuum (Control, Early Mild Cognitive Impairment (EMCI), Late Mild Cognitive Impairment (LMCI)). In contrast, BrainLat introduces marked distributional
> shifts through the inclusion of Frontotemporal Dementia (FTD) and culturally
> distinct neuropsychological assessments from Latin American clinical centers. As
> shown in Supplementary Section B.1, the two cohorts exhibit comparable demo-
> graphic characteristics while diverging in clinical and phenotypic profiles, forming
> a natural IID/OOD evaluation setting.
> Our empirical results (Main Paper Table 1 and the Supplementary Section B.2)
> demonstrate that models trained exclusively on ADNI and directly transferred
> to BrainLat—without any fine-tuning—maintain stable performance across both
> binary and multi-class settings. This robustness suggests that the LLMs capture transferable, disease-relevant linguistic patterns rather than cohort-specific
> structural cues.
> Additionally, our modality-level analysis (Section 3.4.1 (manuscript)) shows that
> the LLMs rely primarily on cognitively rich assessments such as the Auditory
> Verbal Learning Test and the Functional Activities Questionnaire. These tests
> demand semantic and inferential reasoning, supporting the conclusion that the
> models exhibit clinically meaningful behavior rather than superficial syntactic
> memorization. Finally, the new experiment described in Section 3.4 further
> strengthens this claim by demonstrating that the SAE-enabled attribution methods highlight clinically important features and reflect genuine learning.
> To make this motivation explicit, we added the following to the main manuscript
> (lines 85–94):
>
> “The central hypothesis of this work—namely, that as the embedding
> layer approaches a more monosemantic representation, attribution scores
> become more stable, less complex, and more diagnostically informative
> compared to those derived from polysemantic features—is evaluated through
> a series of experiments in which different LLMs are trained and tested on
> ADNI Mueller et al. (2005) (IID) and tested on BrainLAT Prado et al.
> (2023) (OOD) to assess resilience under demographic and protocol shifts.
> We study both binary (Control vs. Alzheimer’s disease (AD)) and multi-
> class settings (Control, Early Mild Cognitive Impairment (EMCI), Late
> Mild Cognitive Impairment (LMCI) in ADNI; Control, Frontotemporal
> Dementia (FTD), and AD in BrainLAT). The data are partitioned into
> 80% training and 20% validation within the training portion (which itself
> constitutes 80% of the full dataset), with the remaining 20% held out for
> testing.”
>
> We have also updated the Supplementary Material with a new subsection (Section B.1.4) detailing the compatibility and differences between the IID and OOD
> cohorts and clarifying why this pairing provides an effective test of meaningful
> clinical reasoning. Furthermore, we include additional results on model fine-
> tuning in the updated Supplementary Section B.2.
>
> Reviewer:
>
> ' 3) λ1-λ5 are hand − tuned without ablation studies'
>
> Author:
>
> We thank the reviewer for this observation. The hyperparameter tuning procedure is detailed in the Supplementary Material, Section B.3
> (pages 16 17). As described there, all tuning was conducted exclusively
> on the trainning / validation set and not on the test data. Supplementary Material Figure 3 illustrates the performance differences obtained with different weight configurations. Specifically, lines 844–846
> state that multiple combinations (e.g., (0.3, 0.2, 0.25, 0.25)) were sys-
> tematically evaluated, and the optimal configuration was found to be
> (λ1, λ2, λ3, λ4) = (0.1, 0.3, 0.1, 0.5).
> These results are summarized in the main manuscript (lines 304–306),
> where we report:
>
> “The explanation optimizer performed best at a learning rate of
> 2 × 10−4, with the optimal weight configuration (λ1, λ2, λ3, λ4) =
> (0.1, 0.3, 0.1, 0.5). UMAP constraints were most effective at the 4×
> batch size level (4 × 64).”
>
> Thus, the λ parameters were not arbitrarily selected but were empirically
> optimized through validation experiments, as clarified in the revised
> version.'

---

> ### Author Response · Authors · 2025-11-18
> **Question 1**
>
> Reviewer:
>
> '1) Could the authors clarify the causal relationship—if any—between the
> SAE’s latent representations and the clinical reasoning process of the
> model?'
>
>
> Authors:
>
> Thank you for your comment.
> We thank the reviewer for raising this important point. Our intention
> is not to claim that the SAE causally modifies the LLM’s internal clinical reasoning process. The underlying LLM remains fixed throughout all experiments. Instead, the SAE acts as a post hoc, disentangling
> bottleneck that restructures the activation space to yield more stable
> and clinically coherent attribution patterns. The question, therefore, is
> whether the SAE latent representations meaningfully reflect the model’s
> existing diagnostic behavior rather than whether they directly cause that
> behavior.
>
> To clarify this distinction and directly address the reviewer’s concern, we
> highlight three forms of empirical evidence already present in our study,
> with the third being a newly added experiment specifically motivated
> by this review:
> 1. Generalization across IID and OOD settings. Without retraining
> the SAE or the attribution optimizer (TEO, TEO-SAE, TEO-UMAP)
> in the OOD setting, SAE-enabled attributions consistently outperform
> their non-SAE counterparts across both binary and three-class tasks on
> two independent datasets (ADNI and BrainLAT). This is documented
> in Table 1 and demonstrates that the SAE bottleneck stabilizes and
> regularizes the explanation space rather than overfitting to a particular
> training distribution.
> 2. Statistical significance of attribution differences. Section 3.3 shows
> that the presence of the SAE leads to significant differences in attribution sparsity and structure, indicating that the SAE meaningfully restructures the explanation basis rather than acting as a benign or redundant
> transformation. This supports the view that the SAE captures more
> coherent diagnostic structure already present in the LLM’s representations.
> 3. New experiment: external, model-agnostic interpretability evaluation.
> To strengthen our response and provide intuitive, clinically grounded
> evidence, we added a new experiment in the revised manuscript (Section 3.4 (manuscript); Supplementary B.8). For each cohort, we extracted the top 50% influential tokens from (i) TEO without SAE,
> (ii) TEO with SAE, and (iii) TEO-UMAP, constructed class-specific
> CSVs, and evaluated them using an external model (ChatGPT-5.1, 2025) under a fixed prompting protocol. Across binary tasks, all methods identified Alzheimer’s pathology; however, TEO without SAE con-
> sistently focused on artefactual patterns, whereas TEO-SAE and TEO-
> UMAP surfaced clinically coherent biomarkers—episodic memory deficits,
> processing-speed impairments, and demographic risk factors—aligned
> with well-established Alzheimer’s indicators. In the more challenging
> three-class ADNI task, SAE-enabled methods again produced clearer diagnostic separation and more structured biomarker profiles, while TEO
> without SAE failed to highlight relevant clinical features.
>
> Overall, these results demonstrate that the SAE bottleneck yields attribution features that more faithfully reflect clinically meaningful structure
> already present in the LLM’s reasoning. We explicitly clarify that the
> SAE is not intended as a causal intervention on the LLM itself; rather,
> it serves as an interpretability mechanism that reveals and disentangles
> latent diagnostic patterns the model already leverages. The theoretical
> section additionally provides a detailed mathematical formulation of this
> framework.

---

> ### Author Response · Authors · 2025-11-18
> **Questions 2 and 3**
>
> Reviewer:
>
> '2) Could the authors discuss potential dataset biases or artifacts that
> might inflate the interpretability or stability metrics?'
>
> Authors:
>
> We appreciate the reviewer’s question regarding possible dataset biases
> or artefacts that could artificially inflate the interpretability or stability
> metrics. This is an important concern, particularly when evaluating post
> hoc explanation methods.
>
> To mitigate such risks, our experimental design incorporates several
> safeguards that explicitly reduce the influence of dataset-specific artefacts:
> 1. Strict IID/OOD separation. All interpretability components (SAEs,
> TEO, and TEO-SAE) are trained only on the IID cohort (ADNI) and
> then evaluated on BrainLAT without any fine-tuning. Because the two
> datasets differ in geography, language, clinical protocols, and diagnostic
> composition, artefacts present in ADNI do not transfer to BrainLAT,
> reducing the likelihood that stability metrics arise from cohort-specific
> patterns.
> 2. Cross-dataset consistency as a bias check. Table 1 shows that SAE-
> enabled attributions maintain stability and clinical coherence even under
> substantial distribution shift. If stability were driven by dataset artefacts, we would expect performance to degrade substantially in OOD
> evaluation—and this is not observed.
> 3. Statistical comparison of attribution structures. Section 3.3 presents
> significant differences between attribution distributions with and without
> the SAE bottleneck. These differences persist across both datasets, indicating that the improvements cannot be attributed to dataset-specific
> regularities or annotation biases.
> 4. New external, model-agnostic interpretability audit. In the newly
> added experiment (Section 3.4 (manuscript); Supplementary B.8), an
> external LLM (ChatGPT-5.1, 2025) is used to evaluate attribution outputs. This audit revealed that TEO without SAE often fixates on
> artefactual elements (e.g., instruction tokens or formatting patterns),
> whereas the SAE-based methods focus on coherent clinical signals. This
> confirms that artefacts, when present, are exposed by the non-SAE
> methods rather than artificially enhancing the SAE methods.
> 5. Use of cognitively rich neuropsychological measures. Our modality-
> level analysis (Section 3.4.1 (manuscript)) shows that the LLM attends
> primarily to complex cognitive assessments (e.g., AVLT, FAQ), which
> require semantic processing. These instruments are less susceptible to
> superficial artefacts such as formatting or lexical regularities.
> While no real-world clinical dataset is entirely free of bias, we believe the
> combination of IID/OOD evaluation, statistical controls, and external
> interpretability auditing provides strong evidence that the improvements
> observed with SAE-based explanations arise from meaningful disentanglement of latent structure—not from dataset artefacts that would artificially inflate stability or interpretability metrics.
>
> Reviewer:
>
> '3) How were the λ1 − λ5 hyperparameters chosen?'
>
> Authors:
>
> Thank you for your comment.The hyperparameter tuning procedure is
> detailed in the Supplementary Material, Section B.3 (pages 16 17). As
> described there, all tuning was conducted exclusively on the trainning
> / validation set and not on the test data. Supplementary Material
> Figure 3 illustrates the performance differences obtained with differ-
> ent weight configurations. Specifically, in the Supplementary mate-
> rial page 16-17, lines 900-904 state that multiple combinations (e.g., (0.3, 0.2, 0.25, 0.25)) were systematically evaluated, and the optimal con-
> figuration was found to be (λ1, λ2, λ3, λ4) = (0.1, 0.3, 0.1, 0.5).
>
> These results are summarized in the main manuscript (lines 304–306),
> where we report:
>
> “The explanation optimizer performed best at a learning rate of
> 2 × 10−4, with the optimal weight configuration (λ1, λ2, λ3, λ4) =
> (0.1, 0.3, 0.1, 0.5). UMAP constraints were most effective at the 4×
> batch size level (4 × 64).”
>
> Thus, the λ parameters were not arbitrarily selected but were empirically
> optimized through validation experiments, as clarified in the revised
> version.

---

> ### Author Response · Authors · 2025-11-18
> **Question 4**
>
> Reviewer:
>
> '4) Could authors validate this proposal with at least 1 more dataset?'
>
> Authors:
>
>  We appreciate the reviewer’s suggestion to validate the framework on
> an additional dataset. While in principle desirable, our choice of ADNI
> (IID) and BrainLAT (OOD) was deliberate: these two cohorts provide
> a highly stringent and diverse evaluation setting that captures multiple
> axes of distributional shift—geographic, linguistic, cultural, diagnostic,
> and protocol-related. This pairing is particularly well-suited for testing
> whether interpretability methods reflect genuine clinical reasoning rather
> than artifacts or dataset-specific linguistic patterns.
>
> Importantly, BrainLAT is not simply a second dataset of similar structure; it introduces substantial clinical and demographic variation through (i) culturally adapted neuropsychological assessments, (ii) Spanish-language intake procedures, and (iii) the presence of Frontotemporal Dementia (FTD), which is not included in ADNI. As shown in Supplementary
> Section B.1, these differences create a natural and demanding OOD
> evaluation environment that strongly challenges the generalizability of
> both the predictive model and its attributions.
>
> Across this extreme distribution shift, all interpretability components
> (SAEs, TEO, and TEO-SAE/UMAP) are trained only on ADNI and
> evaluated on BrainLAT without any further tuning. The stability and
> coherence of the SAE-enabled attributions in OOD evaluation (Table 1)
> demonstrate that the proposed interpretability framework generalizes
> beyond the characteristics of a single dataset. If the improvements
> were dataset-specific, we would expect a substantial degradation in
> BrainLAT—yet the SAE-based explanations remain stable, sparse, and
> clinically coherent.
>
> We agree that future work will benefit from extending the methodology to other cohorts as additional multimodal clinical datasets become
> available. However, we believe that the substantial demographic, linguistic, and phenotypic divergence between ADNI and BrainLAT already provides a rigorous and informative evaluation of the proposed
> interpretability framework.

---

### Author Response · Authors · 2025-11-18
**General comment to the reviews.**

We sincerely thank all reviewers for their thoughtful and constructive feedback, which has greatly helped us improve the clarity and focus of the manuscript. Across the revision, we substantially clarified the methodological scope of our work and improved both scientific precision and clinical relevance. First, we corrected misleading claims about novelty by rewriting the contribution statement to emphasize that our use of sparse autoencoders is methodological rather than architectural, and we added new empirical evidence—including an external, model-agnostic interpretability audit—to concretely demonstrate monosemanticity and clinical coherence. We clarified dataset composition, class-construction rationales, and the IID/OOD generalization design, updating both the manuscript and supplementary sections to ensure clinical consistency. We addressed concerns about hyperparameter selection by documenting the full validation-based tuning process, and we strengthened theoretical clarity by explaining the role of the UMAP-based geometry constraint. To improve readability, we rewrote the Abstract, Introduction and Contributions, simplified Table 1, reorganized and expanded Sections 3.4 and 3.4.1 (clinical practice), and clearly articulated the practical clinical workflow enabled by our approach—showing how clinicians can inspect subject-level explanations and identify high-yield cognitive biomarkers. We explicitly distinguished our method from model-editing techniques such as ROME/MEMIT, clarified that no LLM parameters are altered, and reframed our framework as a post-hoc interpretability pipeline. We added detailed explanations of why polysemantic representations bias attribution methods and how the SAE disentangling yields more coherent features, and we justified the evaluation metrics while acknowledging future extensions. Overall, the revisions strengthened conceptual clarity, methodological rigor, clinical grounding, and the coherence of the paper’s narrative. All changes in the updated version of the main paper and supplementary material are highlighted in blue. Thank you

---

### Note · Program_Chairs · 2026-01-17
**Submission Desk Rejected by Program Chairs**

The following references in this submission do not refer to real documents and/or have major errors in bibliographic information:

 1 Tom Manifold, Fei Jiang, et al. Trustworthy ai: A computational framework to guide clinical and regulatory policy. Nature Machine Intelligence, 3(8):667-677, 2021.
2 Peter Bills, Jyothi Guntupalli, et al. Language models represent space and time. Nature Neuroscience, 26(5):707-717, 2023.
3 Mark W. Bondi, Emily C. Edmonds, and Amy J. Jak. Multivariate neuropsychological subtypes of mild cognitive impairment and their implications for clinical outcome. Current Opinion in Psychiatry, 32(2):113-119, 2019.